# Combinatorial single-cell profiling of major chromatin types with MAbID

Silke J. A. Lochs ®[1,2,5], Robin H. van der Weide ®[1,2,5], Kim L. de Luca ®[1,2],
Tessy Korthout[1,2], Ramada E. van Beek[1,2], Hiroshi Kimura ®[3] & Jop Kind ®[1,2,4] ✉

Gene expression programs result from the collective activity of numerous regulatory factors. Studying their cooperative mode of action is imperative to understand gene regulation, but simultaneously measuring these factors within one sample has been challenging. Here we introduce Multiplexing Antibodies by barcode Identification (MAbID), a method for combinatorial genomic profiling of histone modifications and chromatin-binding proteins. MAbID employs antibody–DNA conjugates to integrate barcodes at the genomic location of the epitope, enabling combined incubation of multiple antibodies to reveal the distributions of many epigenetic markers simultaneously. We used MAbID to profile major chromatin types and multiplexed measurements without loss of individual data quality. Moreover, we obtained joint measurements of six epitopes in single cells of mouse bone marrow and during mouse in vitro differentiation, capturing associated changes in multifactorial chromatin states. Thus, MAbID holds the potential to gain unique insights into the interplay between gene regulatory mechanisms, especially for low-input samples and in single cells.

Gene regulation involves the coordinated activity of many factors at different genomic scales. At a large scale, chromosomes reside in distinct nuclear territories[1,2] that are further partitioned into compartments of similar chromatin states[3,4]. At a local scale, DNA methylation[5], histone post-translational modifications (PTMs)[6] and chromatin remodeling complexes[7] synergistically modulate transcriptional activity. The interplay between these factors ultimately determines cellular identity and function. To understand the mechanisms governing gene expression, technologies capable of simultaneously measuring multiple gene regulatory states are required.

Many powerful methods have been developed that enable single-cell profiling of various aspects of gene regulation. Most prominent are approaches linking transcriptional heterogeneity to variations in DNA methylation[8–10], chromatin accessibility[9,11,12], protein–DNA binding[13], nuclear architecture[13–15] and histone PTMs[16–20]. However, techniques to simultaneously profile many different modalities in the same cell are still limited.

Recently, methods have been developed to profile up to three histone PTMs in the same cell[21–26]. Such multifactorial strategies hold great potential for dissecting the underlying coordination and mechanistic basis that govern gene regulation. These methodologies generally build on antibody detection followed by Tn5-mediated tagmentation[27] and have thus far been implemented to measure three epitopes per cell, all residing in active chromatin or facultative heterochromatin[21–26]. Whether this strategy can be extended to profile increasingly complex sets of epitopes remains unresolved, especially including those of constitutive and inaccessible heterochromatin.

Here we present Multiplexing Antibodies by barcode Identification (MAbID), a method that employs standard restriction-digestion and ligation steps. With MAbID, combined measurements of epitopes across all major chromatin types can be obtained, including facultative and constitutive heterochromatin. We used secondary or primary antibody–DNA conjugates to generate low-input and single-cell readouts for up to six epitopes simultaneously. We demonstrate that with MAbID

[1]Hubrecht Institute, Royal Netherlands Academy of Arts and Sciences (KNAW) and University Medical Center Utrecht, Utrecht, the Netherlands. [2]Oncode Institute, Utrecht, the Netherlands. [3]Cell Biology Center, Institute of Innovative Research, Tokyo Institute of Technology, Yokohama, Japan. [4]Department of Molecular Biology, Faculty of Science, Radboud Institute for Molecular Life Sciences, Radboud University, Nijmegen, the Netherlands. [5]These authors contributed equally: Silke J. A. Lochs, Robin H. van der Weide. ✉e-mail: j.kind@hubrecht.eu

differences in single-cell chromatin states between closely related cell types can be detected and that the method is suitable to classify different cell types obtained from primary mouse bone marrow (BM). We anticipate that MAbID will provide an approach to obtain insights into the multifaceted regulation of gene expression in dynamic and complex biological systems.

## Results

### MAbID enables genomic profiling of diverse chromatin states

To multiplex measurements of several chromatin states within one sample, we devised a strategy to uniquely barcode antibodies and map epitope positions on chromatin through specific restriction-ligation steps. To this end, we covalently conjugated antibodies to DNA adapters using click-chemistry[28–30] (Extended Data Fig. 1a,b). The MAbID protocol (Fig. 1a) involves: (1) collection of ~250,000 cells, nuclei isolation and mild fixation; (2) incubation with primary antibodies followed by incubation with barcoded secondary antibody–DNA conjugates; (3) fluorescence-activated cell sorting (FACS); (4) genomic digestion with MseI, recognizing TTAA sequence motifs; (5) dephosphorylation of the digested genome to prevent self-ligation of the genome and enhance integration of the antibody-adapter; (6) NdeI digestion of the antibody-adapter, leaving an MseI-compatible overhang with a 5′ phosphate; and (7) proximity ligation of the antibody-adapter to the genome to mark the genomic position of the epitope. The protocol continues with (8) lysis and protein degradation followed by (9) NotI digestion to enable subsequent ligation of a sample-adapter with a unique sample barcode (Extended Data Fig. 1b). The sample-adapter enables pooling of multiple samples for linear amplification and subsequent Illumina library preparation[13].

We first benchmarked MAbID against public datasets by using biological replicates of 1,000 K562 nuclei and performing individual measurements of several histone PTMs, RNA Polymerase II and the Lamin B1 protein. We generated secondary antibody–DNA conjugates and employed these in combination with different primary antibodies (Supplementary Table 1) to compare the quality of multiple genomic profiles in parallel. On average, 78.9% of the reads contained the correct sequence structure consisting of antibody and sample barcodes, and 97.7% of the uniquely aligned reads are located at expected TTAA sequence motifs (Extended Data Fig. 1c). A control sample in which the primary antibody was omitted during the first incubation step serves as an input (mock immunoprecipitation, IP) dataset for normalization. The control signal largely mirrors chromatin accessibility, as compared with publicly available assay for transposase-accessible chromatin with sequencing (ATAC-seq) data (Extended Data Fig. 1d). This necessitates normalization of the MAbID profiles to the control to effectively remove the accessibility component (Extended Data Fig. 1d–f).

Visualization of the normalized data by uniform manifold approximation and projection (UMAP) shows good concordance between biological replicates and consistent grouping of the 1,000-cell MAbID samples with corresponding chromatin immunoprecipitation followed by sequencing (ChIP–seq) datasets obtained from millions of cells

(Fig. 1b and Extended Data Fig. 2a). Genome-wide MAbID signal correlates best with publicly available ChIP–seq and CUT&Tag data of matching histone PTMs, with mean Pearson's correlation coefficients ranging from 0.24 to 0.50 for active chromatin types and 0.24 to 0.46 for heterochromatin types (Fig. 1c and Extended Data Fig. 2b). On a local scale, MAbID profiles show expected patterns of enrichment and similarity to ChIP–seq or DamID datasets (Fig. 1d and Extended Data Fig. 2c).

To further explore the specificity of MAbID, we calculated FRiP (Fraction of Reads in Peaks) scores over ChromHMM domains or lamina-associated domains (LADs). For non-normalized MAbID data, FRiP scores are lower with respect to public data from other approaches (Extended Data Fig. 2d). However, the enrichment of normalized data (Signal Enrichment in Peaks, SEiP score) is in a similar range to measurements obtained with orthogonal methods (Extended Data Fig. 2d). All MAbID samples display signal enrichment over the expected genomic regions, irrespective of chromatin type (Fig. 1e). Moreover, for histone PTMs associated with active gene expression, the signal scales according to the transcriptional activity of genes (Fig. 1f).

Next, we determined the resolution of MAbID by quantifying the signal distribution of H3K4me3 over transcription start sites (TSSs) and H3K27me3 over Polycomb-group domains (ChromHMM) compared with corresponding ChIP–seq datasets (Extended Data Fig. 2e). H3K4me3 signal decays to 50% (compared with 100% at the TSS) at 3–4-kilobase (kb) distance from the top of the peak. For H3K27me3, this distance corresponds to 7–8 kb around the domain border. Compared with ChIP–seq, MAbID signal thus generally extends an additional 1–2 kb in either direction. Finally, we investigated the compatibility of MAbID with profiling of other chromatin-binding proteins. We focused on the zinc-finger transcription factor CTCF, and SUZ12, a subunit of the Polycomb Repressive Complex 2. Both proteins display the expected enrichment of signal over publicly available ChIP–seq peaks or ChromHMM domains (Extended Data Fig. 2f). In summary, these results show that MAbID can accurately profile diverse epigenetic modifications and chromatin-binding proteins in as little as 1,000 cells.

### Multiplexing MAbID measurements in one sample

MAbID is designed to multiplex several antibodies in the same sample to profile many epigenetic landscapes simultaneously. To test this, we performed MAbID with four antibodies of different host-origin, along with uniquely barcoded secondary antibody–DNA conjugates specific for each host (Fig. 2a and Supplementary Table 1). We first assessed differences in data quality between individual and multiplexed measurements. All samples group on epitope, with high concordance between biological replicates (Fig. 2b). Importantly, the yield and read-statistics are comparable between samples incubated with a single antibody or all four antibodies simultaneously (Extended Data Fig. 3a,b). Moreover, genome-wide mean Pearson's correlation coefficients with public ChIP–seq data are generally independent of the number of multiplexed antibodies (Fig. 2c and Extended Data Fig. 3c). Finally, we assessed the local distribution of signal for single and combined samples, which

**Fig. 1 | Genomic profiling of a broad range of epigenetic markers with MAbID. a**, Schematic representation of the MAbID procedure. **b**, UMAP embedding of MAbID, ChIP–seq and DamID samples, colored on epitope with encircled chromatin types. One reference dataset is included per chromatin type. Selected Ab indicates the primary antibody used hereafter; alternative Ab represents a different primary antibody against the same epitope. **c**, UMAP as in **b**, colored by correlation with ChIP–seq samples of H3K36me3, H3K27me3 and H3K9me3. Color-scale represents the Pearson's *r* correlation coefficient of MAbID with ChIP–seq samples. **d**, Genome browser tracks of MAbID, ChIP–seq or DamID samples. Genes and ENCODE domain calls (LAD, lamina-associated domain; EnhA1, Active enhancer 1; ReprPC, repressed Polycomb) are indicated. The *y* axis reflects positive log$_2$(counts/control) for MAbID and DamID and fold change (IP/input) for ChIP–seq. **e**, MAbID signal enrichment of Lamin B1 around LADs, H3K27me3 around Polycomb-group domains, as well as H3K4me3 and H3K4me1 around respective ChIP–seq peaks. Top, average enrichment; bottom, signal per genomic region (sorted on MAbID signal). The number *n* represents the number of genomic regions included per heatmap and the data range is indicated underneath. **f**, MAbID signal enrichment of H3K4me3 and H3K36me3 around active genes. Genes were stratified on expression level and categorized in percentiles. Top, average enrichment per percentile-group; bottom, signal per set of genes, ordered from high to low. *n* = 7,553 genes included per heatmap; the data range is indicated underneath. Ab, antibody; gDNA, genomic DNA; IP, immunoprecipitation; Mb, megabases; TES, transcription end site.

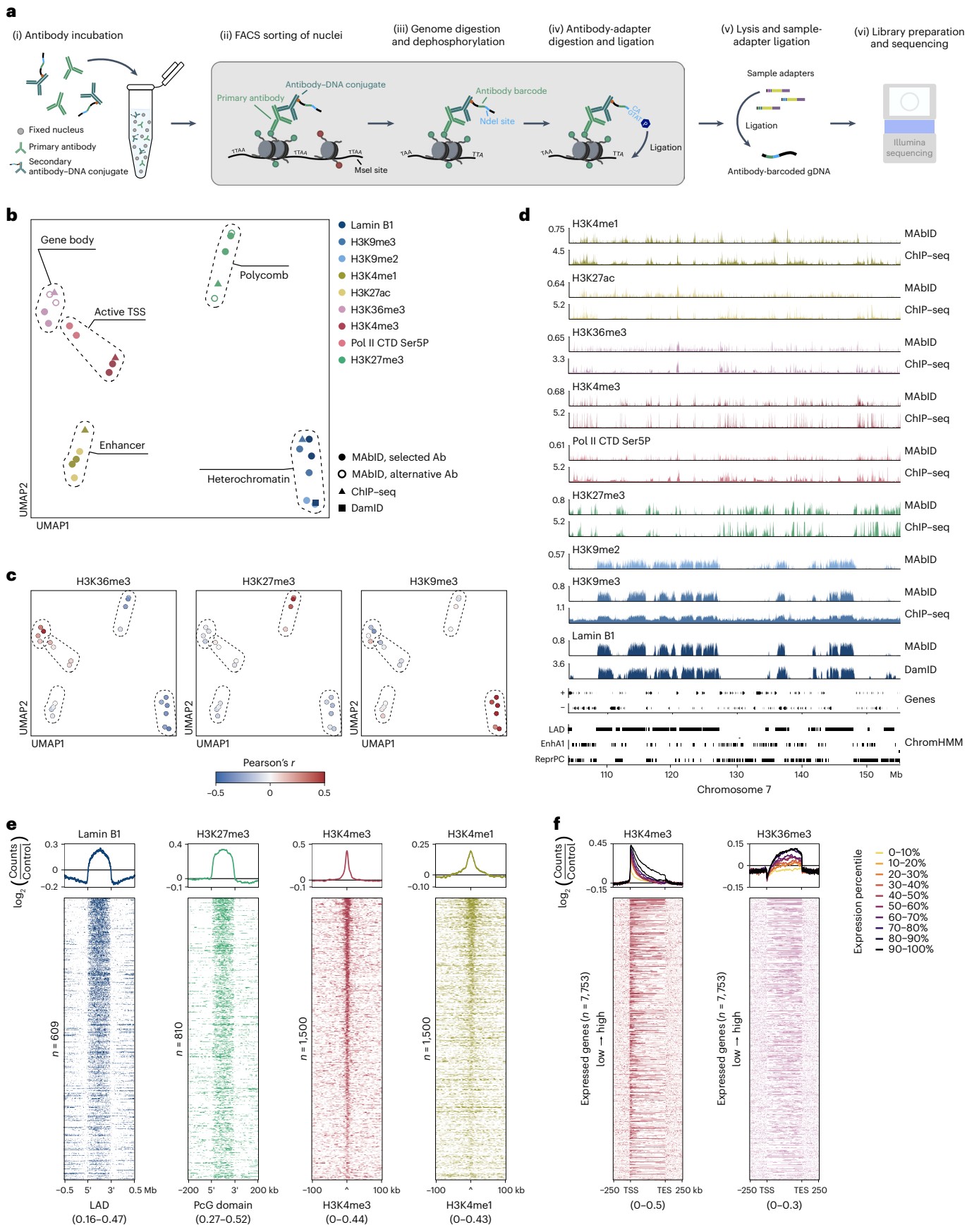

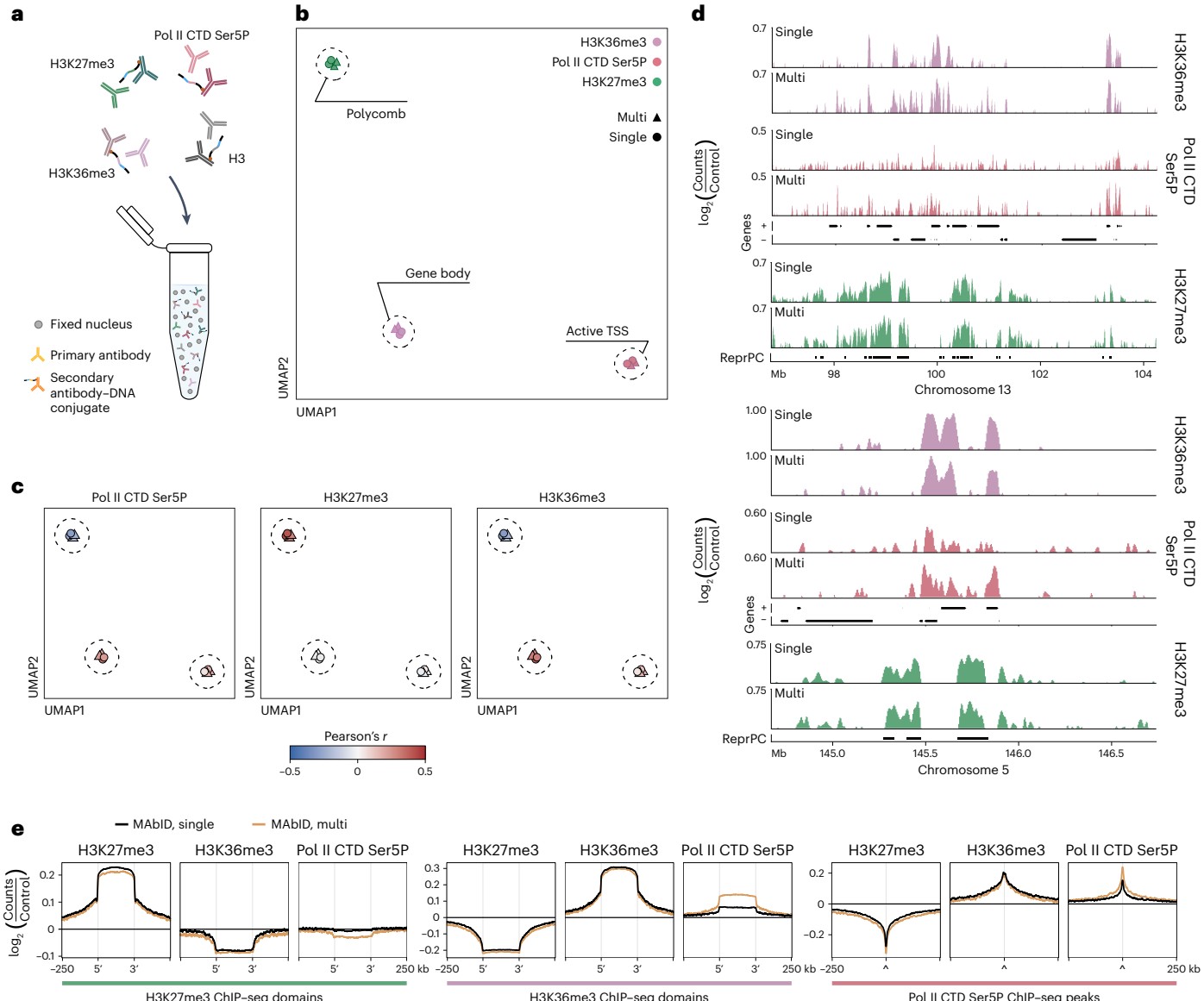

**Fig. 2 | MAbID enables multiplexing of several antibodies in one sample.**
**a**, Schematic showing the multiplexing of primary antibodies from different
species-of-origin with several species-specific secondary antibody–DNA
conjugates. **b**, UMAP of MAbID replicates from combined (multi) or individual
(single) measurements. Coloring is based on the epitope; chromatin types are
encircled. **c**, UMAP as in **b**, colored by correlation with ChIP–seq samples of Pol II
CTD Ser5P, H3K27me3 and H3K36me3. Color-scale represents the Pearson's *r*
correlation coefficient of MAbID with ChIP–seq samples. **d**, Genome browser
tracks at a broad (top) or narrow (bottom) genomic scale comparing MAbID
samples from combined (multi) or individual (single) measurements. Genes
(+, forward; −, reverse) and ChromHMM domain calls are indicated. The *y* axis
reflects positive log₂(counts/control). **e**, Average MAbID signal enrichment of
H3K27me3, H3K36me3 and Pol II CTD Ser5P around the same domains/peaks
called on ChIP–seq data, comparing MAbID samples from combined (multi) or
individual (single) measurements.

show matching profiles on both broad and more narrow genomic scales
(Fig. 2d). Compared with publicly available ChIP–seq data, the signal
enrichments are highly similar and located at the expected genomic
regions (Fig. 2e and Extended Data Fig. 3d).

In combined experiments, we unexpectedly observed rela-
tively high correlation between histone H3 and H3K27me3 ChIP–seq
data (Extended Data Fig. 3c). This similarity is also apparent upon
comparison of H3 genomic profiles for individual and combined
measurements (Extended Data Fig. 3d). We anticipate that this is caused
by cross-reactivity of the anti-sheep secondary antibody–DNA conju-
gate with the primary rabbit H3K27me3 IgG and therefore excluded H3
from subsequent analysis. To rule out the possibility of cross-reactivity
between other secondary antibody–DNA conjugates, we performed

multiplexed MAbID experiments, with combinations in which one of
the three antibodies was excluded. The data are comparable between
the different combinations, verifying that the signal is indeed specific
to the corresponding IgG (Extended Data Fig. 3e).

We also noted that MAbID signal with the Polymerase II CTD Ser5P
antibody is unexpectedly enriched over the gene body, in addition to
the expected enrichment at the TSS[31,32] (Extended Data Fig. 3f). This
may indicate that this antibody has broader affinity for other CTD
phospho-modifications. Regardless, the signal scales with gene expres-
sion output, indicating that the antibody marks transcriptional activ-
ity (Extended Data Fig. 3f). Together, these results show that MAbID
enables robust identification of the genomic localization of several
epitopes in a multiplexed assay.

**Genomic barcoding can be tailored to the epitope of interest**

Next, we sought to increase the modularity of MAbID by including another pair of restriction enzymes in addition to MseI and NdeI. This allows tailoring to the epitope of interest by increasing the theoretical resolution and potentially enhancing signal. We paired MboI with BglII to target GATC sequence motifs, because of (1) the ability of MboI to digest cross-linked chromatin[33,34], (2) the difference in genomic distribution to the TTAA motif (Extended Data Fig. 4a) and (3) the high fraction of mappable genome across chromatin types (Extended Data Fig. 4b). This extension of the method enables mixing of secondary antibody–DNA conjugates targeting both motifs in a single reaction.

We tested this strategy with a combination of three antibodies in single or combined (each with TTAA or GATC) measurements. Based on motif enrichment, secondary antibody–DNA conjugates were created with NdeI-compatible adapters (TTAA) for H3K36me3 and BglII-compatible adapters (GATC) for H3K27me3 and Pol II CTD Ser5P (Extended Data Fig. 4a). Additionally, a sample was included in which each epitope was targeted by both types of secondary antibody–DNA conjugate (both TTAA and GATC), to increase the theoretical number of potential ligation events. Samples consistently group on epitope and display the expected enrichment of signal, regardless of the motif or experimental setting (Extended Data Fig. 4c,d). The signal resolution increases 1–2 kb in the combined (TTAA and GATC) sample compared with the individual samples, as measured by the decay of H3K27me3 signal over Polycomb-group ChromHMM domain borders (Extended Data Fig. 4e). Moreover, the overall complexity and read numbers are similar across samples (Extended Data Fig. 4f,g). These outcomes underscore the robustness and flexibility of MAbID, while the modular design creates the opportunity to tailor experiments to the genomic distribution of the target. In the following experiments, we match restriction enzyme pairs with the epitope of interest.

**Primary antibody conjugates increase multiplexing potential**

To increase the number of multiplexed measurements, we explored the potential of directly conjugating the antibody-adapter to primary antibodies. This is more challenging, because (1) only one primary antibody–DNA conjugate can bind per epitope and (2) conjugation could potentially affect epitope binding of monoclonal primary antibodies. Nevertheless, implementation of primary antibody–DNA conjugates eliminates the dependency on antibody host-origins and vastly increases the potential number of multiplexed measurements. We selected antibodies against a variety of chromatin types and conjugated each to a uniquely barcoded antibody-adapter, with slight modifications to account for differences in storage buffer compositions (Extended Data Fig. 5a).

We performed MAbID in biological replicates of 1,000 K562 nuclei (Fig. 3a). Genomic profiles largely overlap with those from publicly available ChIP–seq data and display overall similarity to corresponding MAbID samples obtained with secondary antibody–DNA conjugates (Fig. 3b,c). The MAbID signal amplitudes and signal-to-noise ratios are somewhat lower compared with MAbID performed with secondary antibodies. This is most apparent for H3K4me3 and H3K36me3, presumably relating to the narrow genomic windows of enrichment for these epitopes. Nevertheless, UMAP embedding shows all samples clustering on chromatin type and their respective secondary antibody–DNA conjugate sample (Fig. 3d). Moreover, the yield and read-statistics are as expected and the biological replicates correlate well (Extended Data Fig. 5b,c). We examined the epitope-specificity of the primary antibody–DNA conjugates further by comparing the signal enrichment with secondary antibody–DNA conjugate samples (Fig. 3e). The signal distribution over respective ChromHMM or public ChIP–seq domains is highly comparable, validating that the antibodies maintain specificity after conjugation. Collectively, these results confirm that

MAbID performed with primary antibody–DNA conjugates generates specific genomic profiles for different chromatin types.

Finally, we tested the performance of primary antibody–DNA conjugates in a multiplexed setting, by selecting six epitopes encompassing a comprehensive set of chromatin types. K562 cells were incubated with the mix of primary antibody–DNA conjugates and sorted as samples of 100 nuclei in 384-well plates. The multiplexed MAbID samples group with the previously generated 1,000-cell individual samples, verifying the similarity between the sample types (Extended Data Fig. 5d). Genome-wide correlations with publicly available ChIP–seq data are comparable between individual and combined measurements (Extended Data Fig. 5e). Thus, MAbID with primary antibody–DNA conjugates potentiates profiling of an increasingly complex set of histone PTMs and chromatin-binding proteins.

**scMAbID measures epigenetic states at single-cell resolution**

We previously optimized single-cell genomic profiling methods using 384-well plates and liquid-handling robots. We therefore integrated these protocols with MAbID to generate multiplexed epigenomic measurements in single cells (scMAbID). To investigate the ability of scMAbID to discern chromatin states between different cell types, we differentiated mouse embryonic stem cells (mESCs) towards early neural progenitor cells (early NPCs)[35]. Both cell types were incubated with the mix of six primary antibody–DNA conjugates targeting a range of chromatin types (Fig. 4a). In parallel, human K562 cells were processed in the same plates to benchmark scMAbID against the bulk datasets (Fig. 4a). The human or mouse origin was assigned in silico with a median number of misannotated reads below 0.4%, indicating that the cell of origin can be robustly assigned (Extended Data Fig. 6a).

A total of 1,956 K562 cells, 1,424 mESCs and 1,424 early NPCs were sequenced and 1,248, 674 and 849 cells passed the quality thresholds, respectively (Extended Data Fig. 6b). The median number of unique counts per cell after filtering is 2,715 for K562 cells, 2,281 for mESCs and 2,842 for early NPCs, and the median count per epitope ranges between 119 and 706 per cell (Fig. 4b). These numbers are in a similar range to those reported by other methods measuring two or three histone PTMs simultaneously[21,23,24,26] (Extended Data Fig. 6c,d). nano-CUT&Tag[23] is the notable exception, substantially outperforming all other methods in terms of read counts (Extended Data Fig. 6c,d). scMAbID is the only plate-based protocol, resulting in lower throughput (Extended Data Fig. 6d). However, the recovery of cells after sequencing is equal to the other approaches and the combination with FACS provides opportunities for selecting cells of interest (Extended Data Fig. 6d).

To verify the specificity of scMAbID data, the unique reads of K562 cells were combined to generate in silico populations (ISPs). scMAbID ISP profiles display a comparable distribution to matching bulk MAbID, ChIP–seq or DamID datasets (Fig. 4c). The correspondence between scMAbID ISP and public reference datasets is moderately lower than observed for bulk MAbID, which is possibly related to the lower read numbers obtained in single-cell measurements. Nevertheless, UMAP visualization shows grouping of scMAbID ISP samples with their respective 1,000-cell counterparts, illustrating the genome-wide similarity between these datasets (Fig. 4d).

Next, we calculated FRiP scores for each epitope in single cells using public reference domains. High FRiP scores are observed for all epitopes over their corresponding domain, while these are considerably lower for unrelated chromatin types (Fig. 4e). We also compared scMAbID FRiP scores with those calculated with publicly available MulTI-Tag[26] and NTT-seq[24] datasets obtained in K562 cells (Extended Data Fig. 6e). NTT-seq moderately outperforms scMAbID, especially on active chromatin types, but overall FRiP scores are on a comparable scale (Extended Data Fig. 6e). scMAbID FRiP scores for H3K9me3 and Lamin B1 in LADs are markedly higher than expected for a random distribution (Extended Data Fig. 6e). Since these epitopes were not

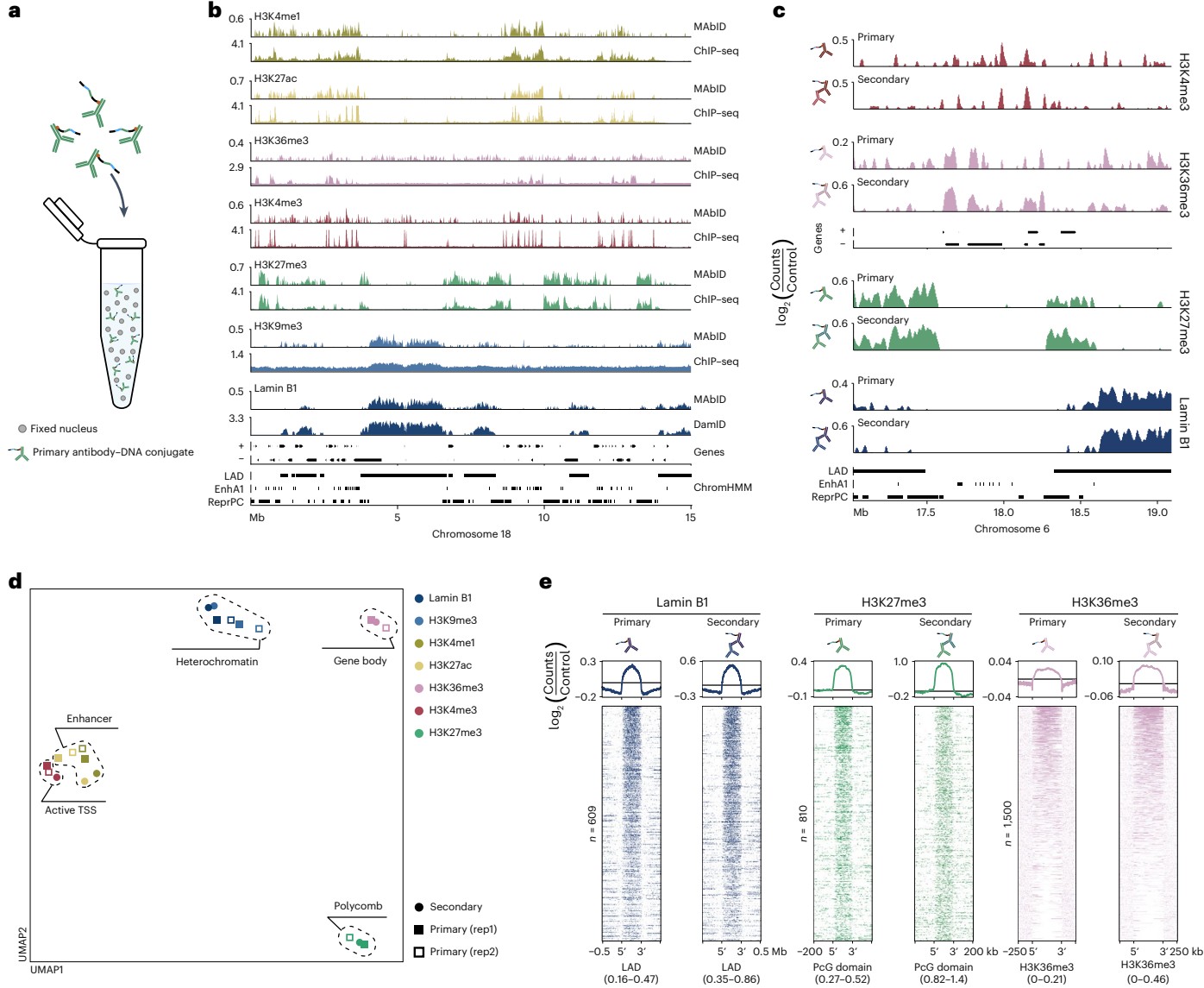

**Fig. 3 | Expanding MAbID with primary antibody–DNA conjugates.**
**a**, Schematic showing nuclei incubation with primary antibody–DNA conjugates. **b**, Genome browser tracks comparing MAbID samples using primary antibody–DNA conjugates with ChIP–seq or DamID samples. Genes (+, forward; −, reverse) and ENCODE/ChromHMM domain calls are indicated. The y axis reflects positive $\log_2$(counts/control) for MAbID and DamID and fold change (IP/input) for ChIP–seq. **c**, Genome browser tracks comparing MAbID samples using primary antibody–DNA conjugates or secondary antibody–DNA conjugates (in combination with a primary antibody). Genes and ENCODE/ChromHMM domain calls are indicated. Scaling is based on the minimum and maximum value per sample and the y axis reflects positive $\log_2$(counts/control) values. **d**, UMAP of

MAbID replicates using primary antibody–DNA conjugates and a MAbID sample of merged replicates using secondary antibody–DNA conjugates (in combination with a primary antibody). Coloring is based on the epitope; chromatin types are encircled. **e**, MAbID signal enrichment of Lamin B1, H3K27me3 and H3K36me3 around the same domains/peaks from ChromHMM or ChIP–seq data, comparing MAbID samples using primary or secondary antibody–DNA conjugates. Top, average enrichment; bottom, signal per genomic region (sorted on MAbID signal). The number n represents the number of genomic regions included per heatmap and the data range is indicated underneath. LAD regions, 4D Nucleome; Polycomb-group domains, ChromHMM; H3K36me3, ChIP–seq domain calls.

measured with the other multifactorial approaches, direct comparisons in constitutive heterochromatin were unattainable.

Finally, we sought to determine if the epitope-specific information from each individual cell enables separation of samples by chromatin state. We took all epitope measurements passing a threshold of 150 unique counts per cell (n = 6,729) and embedded these within UMAP space. Cells consistently separate based on chromatin type, with similar types mixing together (Fig. 4f). This is independent of read depth per cell or epitope (Extended Data Fig. 6f). Collectively, these results validate that scMAbID generates specific epigenetic profiles at single-cell resolution in multiplexed experiments of six epitopes.

**Integrating measurements of multifactorial chromatin states**
Next, we explored the ability of scMAbID to discern between mESCs and early NPCs based on multiplexed epigenomic profiles. mESC scMAbID ISP samples display similar genomic distributions to those observed with ChIP–seq or bulk MAbID (Extended Data Fig. 7a). For both cell types, FRiP scores of single-cell epitope measurements are higher for the corresponding chromatin types with respect to other regions (Extended Data Fig. 7b). Moreover, the single-cell epitope measurements with the highest depth (mESCs, n = 1,800; early NPCs, n = 1,800) mainly separate on their respective chromatin type (Extended Data Fig. 7c). Overall, these results validate the epitope-specific scMAbID measurements in mESCs and early NPCs.

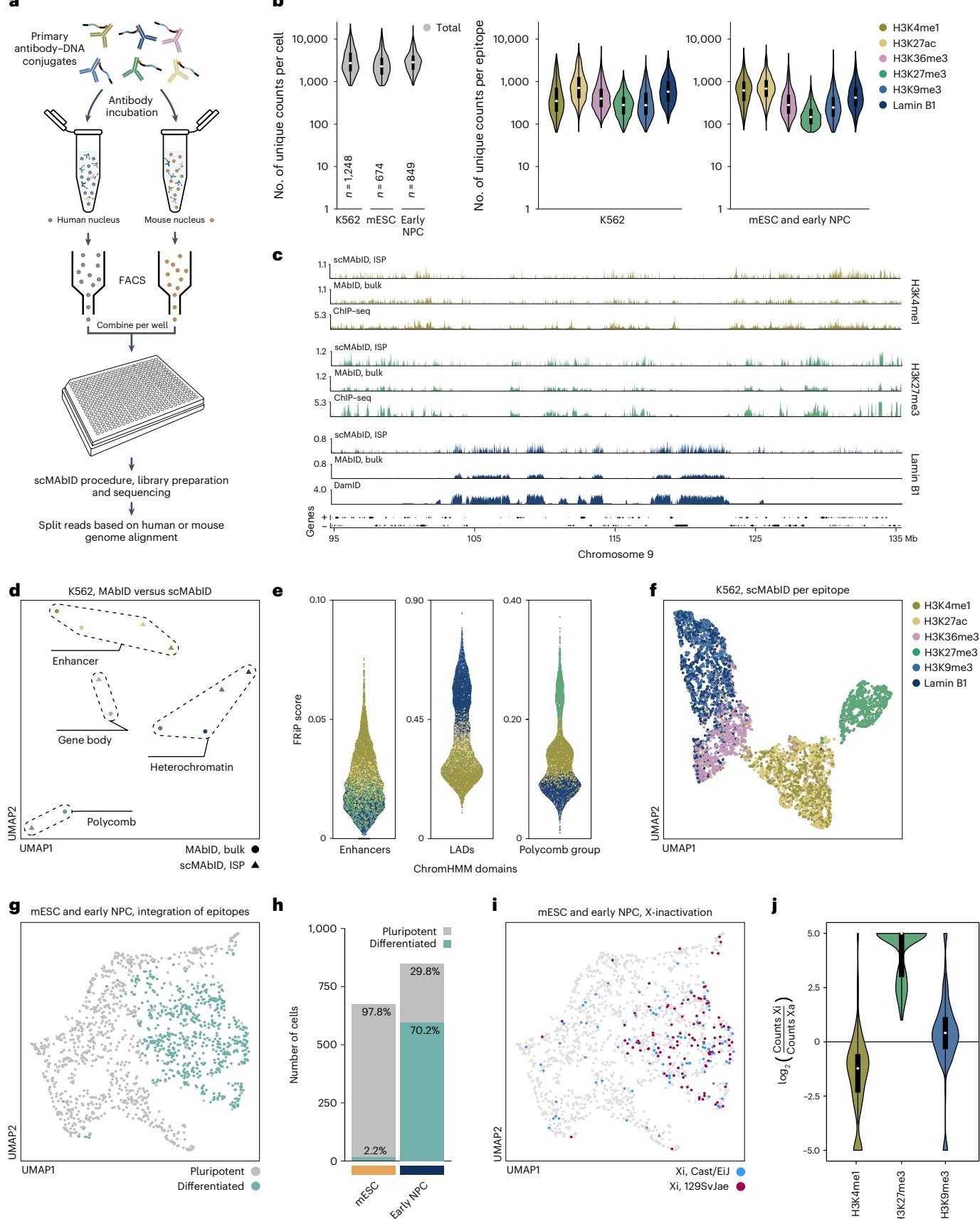

We noticed that while UMAP embedding is mostly driven by chromatin signature, mESCs and early NPCs do separate within the same chromatin type (Extended Data Fig. 7c). We wondered whether integration of all multiplexed measurements would improve separation.

Therefore, we computed the combined epigenomic information for each cell by calculating cell-similarity (Jaccard) matrices per epitope and performing dimensionality reduction on the summed matrices[16]. Upon cluster assignment, 97.8% of the mESCs and 70.2% of early NPCs

**Fig. 4 | Integration of six multiplexed MAbID measurements in single cells.**
**a**, Human (K562) and mouse (mESC, early NPC) cells are incubated with a combination of six primary antibody–DNA conjugates for scMAbID processing. **b**, Violin plots showing the number of unique scMAbID counts per cell and per epitope. K562, $n = 1,248$; mESC, $n = 674$; early NPC, $n = 849$ cells. Boxplots inside violin plots show minima, maxima, interquartile range (box bounds) and median (white dot). **c**, Genome browser tracks comparing K562 scMAbID ISPs ($n = 1,248$) and bulk K562 MAbID samples of H3K4me1, H3K27me3 and Lamin B1 with ChIP–seq or DamID samples. Genes (+, forward; −, reverse) are indicated. The $y$ axis reflects positive $\log_2$(counts/control) for MAbID and DamID and fold change (IP/input) for ChIP–seq. **d**, UMAP of K562 scMAbID ISPs and MAbID samples. Coloring is based on epitope; chromatin types are encircled. **e**, FRiP scores per scMAbID epitope measurement per K562 cell across ChromHMM domains−

Enhancers (EnhA1), LADs and Polycomb-group (ReprPC). Plotting order from back to front: H3K9me3, H3K36me3, Lamin B1, H3K27me3, H3K27ac, H3K4me1. **f**, UMAP of K562 scMAbID samples. Each dot represents one epitope measurement, colored on epitope. Samples with at least 150 unique counts per epitope were included ($n = 6,729$). **g**, UMAP of integrated mouse scMAbID samples (mESC, $n = 674$; early NPC, $n = 849$). Each dot represents one cell, colored on Leiden algorithm cluster assignments ('pluripotent' or 'differentiated'). **h**, Barplot of percentages of mESCs or early NPCs assigned to each cluster of **g**. **i**, UMAP as **g**, colored on inactive X-chromosome allele (Xi) based on the allelic H3K27me3 count-ratio. **j**, Violin plots showing the unique count-ratio of the Xi versus the active X-allele (Xa) per cell for H3K4me1, H3K27me3 and H3K9me3 ($n = 161$ identified cells). The $y$ axis reflects $\log_2$(counts Xi/counts Xa). ISP, in silico population.

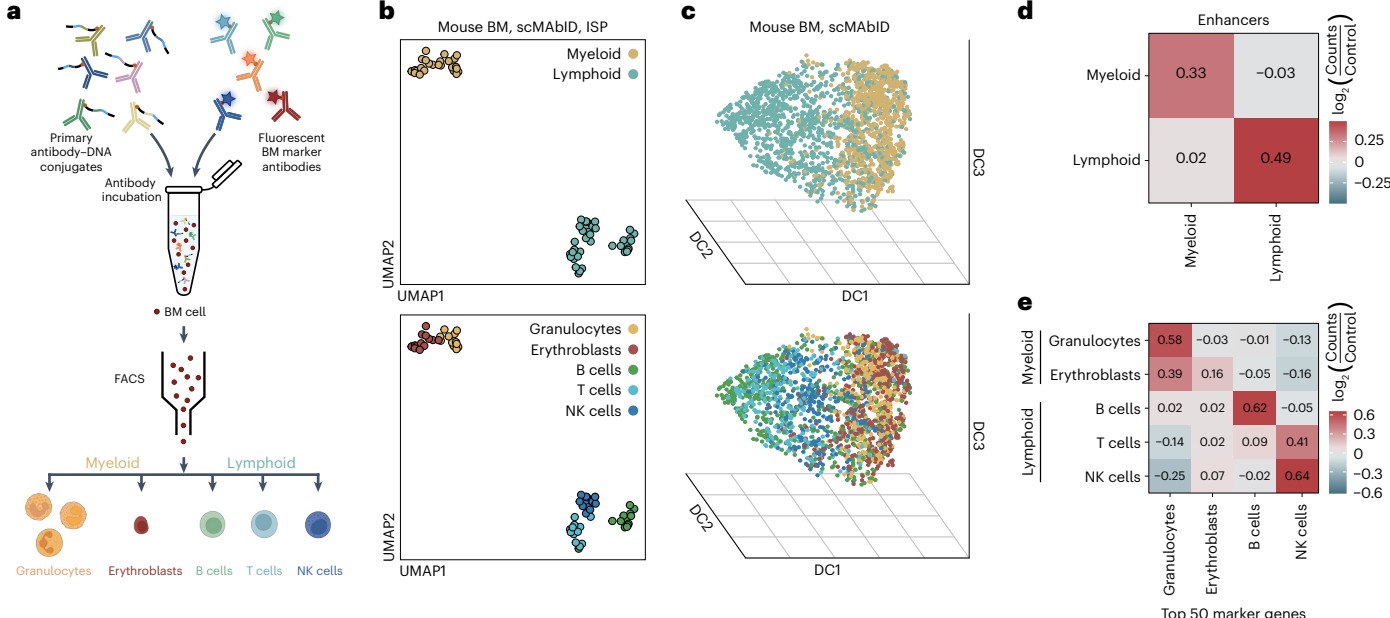

**Fig. 5 | Multifactorial chromatin profiling in primary mouse BM distinguishes cell type-specific gene expression programs. a**, Schematic representation of the scMAbID experiment, incorporating BM cell-surface marker stainings. Cell images were created with BioRender.com. **b**, UMAP of BM scMAbID ISP samples, with integrated-epitope measurements summed per cell type, in which each dot represents the combined samples of one plate (plates, $n = 13$). Coloring is based on the lineage (top) or cell type (bottom). **c**, Three-dimensional diffusion maps of single-cell scMAbID samples, with integrated-epitope measurements for each cell ($n = 3,433$). Coloring is based on the lineage (top) or cell type (bottom). DC, diffusion component. **d**, Matrix visualizing the counts of scMAbID ISP enhancer

(H3K4me1 + H3K27ac) samples per lineage over the top 50 most differentially expressed marker genes per lineage. scMAbID H3K4me1 and H3K27ac counts from each lineage were summed over all marker genes per set and normalized for the control dataset. Values reflect $\log_2$(counts/control). **e**, Matrix visualizing the counts of scMAbID ISP enhancer (H3K4me1 + H3K27ac) samples per cell type over the top 50 most differentially expressed marker genes per cell type. scMAbID H3K4me1 and H3K27ac counts from each cell type were summed over all marker genes per set and normalized for the control dataset. Values reflect $\log_2$(counts/control). NK, natural killer.

are assigned to their cellular origin based on integrated chromatin signatures (Fig. 4g,h and Extended Data Fig. 7d). This confirms that multifactorial chromatin profiles contain sufficient information to accurately separate closely related cell types. Interestingly, 29.8% of early NPCs are annotated as mESCs, presumably because these cells failed to differentiate or maintained a more pluripotent state (Fig. 4h). We subsequently labeled the clusters as 'pluripotent' and 'differentiated'.

We next assessed the contribution of each modality to the assignment of cellular states. To examine this, we used the information gain metric[36] to systematically determine the accuracy of cluster assignments with reduced sets of epitopes. The information gain metric increases with the inclusion of additional epitopes, underlining the added value of multiplexing measurements (Extended Data Fig. 7e). Unsurprisingly, H3K27me3, H3K4me1 and H3K27ac contributed most to cluster assignment, as these are reported to be valuable predictors of cell type and developmental stage (Extended Data Fig. 7f)[16,20].

## scMAbID captures epigenetic transitions upon X inactivation
Female mESCs undergo X-chromosome inactivation (XCI)[37] upon differentiation. This involves major changes in the distribution of several histone PTMs. Our female mESCs are of hybrid origin (Cast/EiJ × 129SvJae), which enables the separation of parental alleles based on single-nucleotide polymorphisms. We therefore addressed whether multiplexed scMAbID data could be utilized to assign cells and alleles that underwent XCI and identify their multifactorial epigenetic signatures.

The inactive X-allele (Xi) is associated with a marked increase in H3K27me3 levels[38,39]. Therefore, we calculated the allelic ratio of unique H3K27me3 counts on the X-chromosome to identify cells that underwent XCI and their respective Xi-allele. As expected, XCI cells predominantly reside in the differentiated cluster (Fig. 4i and Extended Data Fig. 7g). Overall, cells of the differentiated cluster display increased H3K27me3 levels across the X-chromosome and over Hox gene clusters (Extended Data Fig. 7h). Interestingly, only 4.0% of early NPCs labeled

as pluripotent have undergone XCI, compared with 27.0% of early NPCs assigned to the differentiated cluster, implying that these cells indeed failed to exit pluripotency.

Lastly, we determined the occupancy of H3K4me1 and H3K9me3 on the Xi compared with the active X-allele. The levels of H3K4me1, marking enhancer regions, are decreased on the Xi as expected during early stages of XCI[40,41] (Fig. 4j). H3K9me3 levels on the Xi are increased, in line with previous reports showing that this mark increases coincidently with the accumulation of *Xist* RNA on the inactive X-allele[40,42] (Fig. 4j). These results highlight the potential of MAbID to capture single-cell multifactorial dynamics in chromatin states along differentiation trajectories.

### Identifying gene expression signatures in mouse BM

As a further application, we investigated the performance of scMAbID on primary tissue. To this end, we isolated primary BM cells from mice and performed ethanol fixations to preserve overall cell structure. Cells were incubated with a combination of six primary antibody–DNA conjugates, with each antibody conjugated to both TTAA- and GATC-compatible antibody-adapters to maximize genomic coverage. Subsequently, cells were stained with five fluorescently labeled antibodies against BM cell-surface markers to isolate cell types by FACS, including granulocytes and erythroblasts from the myeloid lineage as well as B cells, T cells and natural killer cells from the lymphoid lineage (Fig. 5a and Extended Data Fig. 8a). In parallel, K562 cells are processed in each well as an internal control. The unique counts per cell and epitope are similar across cell types, even though these are lower compared with our previous observations for scMAbID experiments (Extended Data Fig. 8b). We retained 3,433 BM cells (of 4,862 sequenced cells, 70.6% recovery) after quality control, ranging from 471 to 969 cells per cell type (Extended Data Fig. 8b).

To assess the biological information in the BM scMAbID dataset, we integrated all epitope measurements per cell, as done before[16]. First, BM scMAbID ISP samples were generated by summing integrated-epitope matrices per 384-well plate for each cell type. UMAP visualizations of these samples show consistent separation between the myeloid and lymphoid lineages, as well as grouping on cell type (Fig. 5b). For single-cell integrated-epitope samples, we used three-dimensional diffusion map embedding[43,44] to preserve global structures (Fig. 5c). The diffusion components are primarily driven by lineage and although grouping on cell types is evident, it is less strong than in the ISP-based analysis (Fig. 5c and Extended Data Fig. 8c). Nonetheless, these results affirm that scMAbID can be used to obtain specific multifactorial chromatin states from primary cells.

Next, we intersected scMAbID measurements with cell type-specific gene expression programs. To achieve this, we identified genes unique to (1) each lineage and (2) each cell type based on public sortChIC data[45]. The specificity of these marker gene sets was validated by Gene Ontology (GO) enrichment analysis (Extended Data Fig. 8d). Because T cells were not part of the reference data, we compared these with the closely related natural killer cells. We then calculated scMAbID signal over the top 50 marker genes of each lineage or cell type. We focused on enhancer epitopes H3K4me1 and H3K27ac because of the relative high epitope counts. The scMAbID ISP signals for these epitopes are strongly enriched over lineage- and cell type-specific genes (Fig. 5d,e). The lower enrichment for erythroblasts is likely related to the more lenient gating strategy for this cell type, resulting in a less pure population (Fig. 5e and Extended Data Fig. 8a). T cells show a high signal enrichment over natural killer cell marker genes, as expected (Fig. 5e). Moreover, average single-cell epitope counts across marker genes show lineage and cell type specificity, generally being higher for active chromatin types compared with inactive chromatin types (Extended Data Fig. 8e,f).

Finally, we examined the potential of scMAbID data for unbiased identification of differentially expressed genes between BM cell types.

We conducted Wilcoxon rank-sum tests on the single-cell scMAbID enhancer epitope counts over all genes, by combining count numbers for H3K4me1 and H3K27ac. As a result, we obtained a small selection of significant genes, many of which are reportedly expressed in specific BM cell types (Supplementary Table 2). A few illustrative examples include *Trem1* (refs. 46,47), *Bank1* (refs. 48,49), *Pik3cd* (refs. 50,51) and *Ccr2* (refs. 52,53), which were also identified in the marker gene sets of the reference data (Extended Data Fig. 8g). Even though these genes are only detected in a small fraction of cells, the enhancer epitope-count numbers are evidently highest in the expected cell type (Extended Data Fig. 8g). Together, these results show the promise of scMAbID to study cell type-specific gene expression programs in primary tissues.

## Discussion

Recent advancements of single-cell multi-omics strategies empower deeper analyses of gene regulation, but multiplexing measurements of several epigenetic modifications remains challenging[9–11,16,19]. We introduced MAbID, a method for combined single-cell profiling of histone PTMs and chromatin-binding proteins, to obtain joint readouts of an unprecedented six epitopes encompassing all major chromatin types.

Other recent methods generating combined measurements employ Tn5 transposase to map epitope positions on chromatin[21,23–26]. While this strategy yields high-quality single-cell profiles across different chromatin types[16,54], it remains unclear how the intrinsic affinity for open chromatin[55] will affect combined measurements, especially in profiling constitutive heterochromatin. With Tn5-independent MAbID, we successfully performed multifactorial profiling of such epitopes located at inaccessible chromatin in combination with epitopes residing in active chromatin.

Further improvement of MAbID can be achieved by reducing background signal in accessible chromatin regions (Extended Data Fig. 1d). While the background can be corrected for by normalization, it would be desirable to further reduce this through optimization of blocking reagents, antibody titrations or improved restriction-digestion and ligation steps. Since MAbID employs restriction-ligation reactions, the signal distribution and resolution is limited to certain sequence motifs. To enhance this, additional sets of restriction enzymes could be included, such as the presented MboI/BglII restriction pair, or more sequence-unbiased genome-digestion enzymes could be incorporated, such as MNase[56].

To our knowledge, MAbID is the first method to profile a combination of more than three epitopes, even though there is no theoretical or technical limitation towards combining more measurements for Tn5-based multifactorial approaches. As multiplexing measurements did not influence MAbID data quality, we expect that increasing the number of epitopes above six should be feasible. In this regard, the efficiency of the conjugation procedure is critical in obtaining high-quality data. Especially for monoclonal antibodies, it is imperative to validate the binding potency and specificity towards the epitope after conjugation.

MAbID's plate-based protocol provides the opportunity to select specific cells from a larger population by FACS, which can be a powerful strategy to enrich for rare cell types. We validated this by sorting five discrete cell types from mouse BM, using fluorescently labeled antibodies against cell-surface markers. This approach can not only reduce sequencing costs, but also allows for the addition of labeled control cells during antibody incubation, improving the efficiency and thereby enabling the protocol to work with increasingly low cell numbers. On the other hand, plate-based assays have limited throughput, which can be resolved through future implementation of combinatorial-indexing strategies[57].

A common challenge for all current multifactorial methods, including MAbID, is the low coverage obtained from single cells[21,23,24,26]. This sparsity hampers studying relationships between epitopes, such as investigating co-occupancy. A recent study by Gopalan et al.[21] tackled this in bulk samples by capturing reads containing two epitope-specific

barcodes. Such a strategy could be incorporated in MAbID through a few modifications of the protocol, such as PCR-based amplification.

We anticipate that MAbID, as an orthogonal method to the existing tagmentation-based approaches, will contribute to the advancement of the single-cell multi-omics field to study the combined epigenetic landscapes of complex biological systems in integrated experiments.

## Online content

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

## Methods

### Cell culture

Cell lines were grown in a humidified chamber at 37 °C in 5% $CO_2$, and were routinely tested for mycoplasma. K562 cells (gift from the van Steensel laboratory, NKI, the Netherlands) were cultured in suspension in RPMI 1640 (Gibco, 61870010) supplemented with 10% FBS (Sigma, F7524, lot BCBW6329) and 1% Pen/Strep (Gibco, 15140122). Cells were passaged every 2–3 d. Mouse F1 hybrid Cast/EiJ (paternal) × 129SvJae (maternal) embryonic stem cells (mESCs; gift from the Joost Gribnau laboratory, Erasmus MC, the Netherlands) were cultured on irradiated primary mouse embryonic fibroblasts in mESC culture medium defined as: Glasgow's MEM (G-MEM, Gibco, 11710035) supplemented with 10% FBS, 1% Pen/Strep, 1 × GlutaMAX (Gibco, 35050061), 1 × MEM nonessential amino acids (Gibco, 11140050), 1 mM sodium pyruvate (Gibco, 11360070), 0.1 mM β-mercaptoethanol (Sigma, M3148), 1,000 U ml$^{-1}$ ESGROmLIF (EMD Millipore, ESG1107). mESCs were alternatively cultured in feeder-free conditions on gelatin-coated plates (0.1% gelatin, in-house) in 60%-BRL medium, a mix of 40% culture medium and 60% conditioned culture medium (incubated for 1 week on Buffalo Rat Liver cells), supplemented with 10% FBS, 1% Pen/Strep, 1 × GlutaMAX, 1 × MEM nonessential amino acids, 0.1 mM β-mercaptoethanol, 1,000 U ml$^{-1}$ ESGROmLIF. Cells were split every 2–3 d and medium was changed every 1–2 d. For collection, cells were washed with PBS (in-house) and incubated with TrypLE Express Enzyme (Gibco, 12605010) for 3 min at 37 °C. Cells were dissociated by pipetting and TrypLE was inactivated by diluting cells fivefold in culture medium, before proceeding with fixation and permeabilization as described in 'Cell collection and fixation'.

### Neural differentiation

For differentiation towards the neural lineage (largely following a standard in vitro differentiation protocol[35]), mESCs were taken in culture on mouse embryonic fibroblasts and passaged three times in feeder-free conditions in 60%-BRL medium. On day 0, mESCs were plated on gelatin-coated six-well plates (0.15% gelatin, Sigma, G1890) at $2.5 × 10^4$ cells per cm$^2$ in N2B27 medium defined as: 0.5 × DMEM-F12 (Gibco, 11320033), 0.5 × Neurobasal medium (Gibco, 21103049), 15 mM HEPES (Gibco, 15630080), 0.5 × N-2 supplement (Gibco, 17502048), 0.5 × B-27 serum-free supplement (Gibco, 17504044), 0.1 mM β-mercaptoethanol. From day 3 onwards, cells were washed daily with DMEM-F12 medium and refreshed with N2B27 medium. For collection on day 5, cells were washed with DMEM-F12 and incubated for 1 min at room temperature with Accutase Enzyme Detachment Medium (Invitrogen, 00-4555-56). Cells were dissociated by pipetting and Accutase was inactivated by diluting cells tenfold in DMEM-F12, before proceeding with fixation and permeabilization as described in 'Cell collection and fixation'.

### Mouse BM isolation and ethanol fixation

All mice used in this study were bred and maintained in the Hubrecht Institute Animal Facility. Experimental procedures were approved by the Animal Experimentation Committee of the Royal Netherlands Academy of Arts and Sciences and performed according to guidelines. C57BL/6NCrl genotype mice were used for BM isolations, four female littermates of 9 weeks old. To isolate BM cells, tibia and femur bones from the hindlegs were removed. The top of the bone was removed and marrow was flushed out using a syringe with HBSS buffer (Gibco, 14025092). Cells were isolated from the marrow by pipetting and poured through a 70-μm cell strainer (Greiner, 542070) before diluting in 25 ml of HBSS buffer, followed by centrifuging for 10 min at 300$g$ at 4 °C. Supernatant was removed, and cells were resuspended in 10 ml of PBS and centrifuged at 500$g$ for 5 min. After removing supernatant, cells were counted using a TC20 Automated Cell Counter (BioRad, 1450102) and diluted in 300 μl of PBS per $1 × 10^6$ cells. Per 300 μl of PBS, 700 μl of ice-cold ethanol (100%, Boom, 84028185) was added dropwise while vortexing, to reach a final 70% ethanol concentration. Cells were fixed for 1 h at −20 °C. Next, cells were washed with wash buffer 1 (WB1;

20 mM HEPES pH 7.5 (Gibco, 15630-056), 150 mM NaCl, 66.6 μg ml$^{-1}$ Spermidine (Sigma, S2626), 1 × cOmplete protease inhibitor cocktail (Roche, 11697498001), 0.05% Tween20 (Sigma, P9416), 2 mM EDTA). Cells were stored in WB1 supplemented with 10% dimethylsulfoxide (Calbiochem, 317275) and frozen at −80 °C.

### Antibodies

For antibodies, see Supplementary Table 1.

### ABBC and SBC adapters

**ABBC adapter.** Double-stranded ABBC antibody-adapters were conjugated to the antibody via SPAAC click reaction[28–30] (see the section 'Antibody–DNA conjugation'). The top strand was produced as an HPLC-purified oligo with a 5′ Azide modification (IDT, /5AzideN/); the bottom strand was produced as standard-desalted oligo. The NdeI-compatible adapter (TTAA motif, MseI-digested genome) has a 55-nucleotide (nt) linker, a NotI recognition site, a 6-nt ABBC barcode and an NdeI recognition site (5′–3′). In the BglII-compatible adapter (GATC motif, MboI-digested genome), the NdeI recognition site is replaced by a BglII recognition site, of which the adenine on the bottom strand was methylated to block MboI digestion (IDT, /iN6Me-dA/). The oligo is produced as HPLC-purified. For sequences, see Supplementary Table 3. Top and bottom oligos were annealed at a 1:1 ratio at 10 μM final in 1 × annealing buffer (10 mM Tris-Cl, pH 7.4, 1 mM EDTA and 100 mM NaCl) in 0.5-ml DNA LoBind tubes (Eppendorf, 0030108400) by incubating in a PCR machine at 95 °C for 5 min, followed by gradual cooling with 0.5 °C per 15 s to 4 °C final.

**SBC adapters.** SBC adapters are forked double-stranded DNA adapters, which can ligate to the ABBC adapters. The bottom adapter has a 5′ Phosphorylation modification (IDT, /5Phos/) and 4-nt GGCC (5′ to 3′) overhang to facilitate ligation to NotI-digested DNA. Top and bottom oligos were produced as standard-desalted. The adapters contain a 6-nt noncomplementary fork, T7 promoter, 5′ Illumina adapter (from Illumina TruSeq Small RNA kit) and a split 2 × 3-nt unique molecular identifier (UMI) interspaced with a split 2 × 4-nt SBC barcode (5′ to 3′). For sequences, see Supplementary Table 4. Annealing is done as for ABBC adapters, but at 40 μM final in a 96-well plate.

### Antibody–DNA conjugation

**Secondary antibody–DNA conjugates.** Secondary antibody–DNA conjugations were performed as described by Harada et al.[25,28], with minor modifications. Briefly, secondary IgG (Jackson ImmunoResearch, see Supplementary Table 1) was buffer-exchanged to 100 mM NaHCO$_3$ (pH 8.3) using Zeba Spin Desalting columns (40K MWCO, 0.5 ml, ThermoFisher, 87767). First, 100 μg of antibody in 100 μl of 100 mM NaHCO$_3$ (pH 8.3) was conjugated with dibenzocyclooctyne (DBCO)-PEG4-NHS ester (Sigma, 764019) by adding 0.25 μl of DBCO-PEG4-NHS (dissolved at 25 mM in dimethylsulfoxide, 10:1 molar ratio to antibody) and incubating for 1 h at room temperature on a rotor at 8 r.p.m. Sample was passed through a Zeba Spin Desalting column and buffer-exchanged to PBS. DBCO-PEG4-conjugated antibodies were concentrated using an Amicon Ultra-0.5 NMWL 10-kDa centrifugal filter (Merck Milipore, UFC501024) and measured on a NanoDrop 2000, before diluting to 1 μg μl$^{-1}$ in PBS. Conjugation of antibody with the ABBC adapter was performed at 1:2 molar ratio by mixing 75 μl of DBCO-PEG4-conjugated antibody (75 μg) with 100 μl of double-stranded ABBC adapter (10 μM; see the section 'ABBC and SBC adapters'). Samples were incubated at 4 °C for 1 week on a rotor at 8 r.p.m. Subsequent clean-up of the antibody–DNA conjugate was performed as described by Harada et al.[25,28], with an average yield of 20–30 μg. Antibody–DNA conjugate concentration was measured with Qubit Protein Assay (Invitrogen, Q33211). Sample quality and conjugation efficiency were assessed using standard agarose gel electrophoresis or Native PAGE with TBE 4–12% gradient gels (Invitrogen, EC62352BOX), stained with SYBR Gold Nucleic Gel stain (Invitrogen, S11494). PAGE gels

were imaged using the Amersham Typhoon laser-scanner platform (Cytiva). Antibody–DNA conjugates were stored at 4 °C.

**Primary antibody–DNA conjugates.** Primary antibody–DNA conjugations were performed as described in the previous section, 'Secondary antibody conjugates', with minor modifications. Primary antibodies were first cleaned using the Abcam Antibody Purification Kit (Protein A) (Abcam, ab102784) following manufacturer's instructions (performing overnight incubation at 4 °C in the spin cartridge on a rotor at 8 r.p.m.). All elution phases were taken. Purified antibodies were concentrated using an Amicon Ultra-0.5 NMWL 10-kDa centrifugal filter, after which 350 µl of 100 mM $NaHCO_3$ was added and concentrated again to exchange buffers. Concentrated antibody was measured on the Nanodrop 2000. Subsequent steps were performed as described from the DBCO-PEG4-NHS incubation onwards.

## Cell collection and fixation

**K562 cells, mESCs and early NPCs.** Cells were collected (~$10 \times 10^6$ cells) and washed with PBS. All centrifugation steps were at 200$g$ for 4 min at 4 °C. Cells were fixed in 1% formaldehyde (Sigma, F8875) in PBS for 5 min, before adding 125 mM final concentration of glycine (Sigma, 50046) and placing on ice. Samples were kept cold for all subsequent steps and incubations performed on a tube roller. Cells were washed three times with PBS before resuspension in WB1 (20 mM HEPES pH 7.5 (Gibco, 15630-056), 150 mM NaCl, 66.6 µg ml⁻¹ Spermidine (Sigma, S2626), 1 × cOmplete protease inhibitor cocktail (Roche, 11697498001), 0.05% saponin (Sigma, 47036), 2 mM EDTA) and transferred to a protein LoBind Eppendorf tube (Eppendorf, EP0030108116-100EA). Cells were permeabilized for 30 min at 4 °C. BSA (Sigma, A2153) was added to 5 mg ml⁻¹ final concentration and incubated for 60 min at 4 °C. Permeabilized nuclei were used for antibody incubation. Note that formaldehyde fixation (using saponin-containing wash buffers) can be replaced with ethanol fixation (using Tween20-containing wash buffers) to preserve the cellular membrane and enable immunostainings for cell-surface markers. See the section 'Mouse BM isolation and ethanol fixation'.

**Mouse BM cells.** For ethanol-fixed mouse BM cells, in all wash buffers 0.05% saponin was replaced with 0.05% Tween20. BM cells (see the section 'Mouse BM isolation and ethanol fixation') were thawed on ice and washed twice in WB1 (20 mM HEPES pH 7.5, 150 mM NaCl, 66.6 µg ml⁻¹ Spermidine, 1 × cOmplete protease inhibitor cocktail, 0.05% Tween20 (Sigma, P9416), 2 mM EDTA) before antibody incubation.

## Antibody incubations

All centrifugation steps were at 200$g$ for 4 min at 4 °C and incubations were performed on a tube roller. See Supplementary Table 1 for antibodies and concentrations.

**Primary antibody–DNA conjugates.** Permeabilized nuclei (or cells) were counted on a TC20 Automated Cell Counter (BioRad, 1450102). Nuclei were diluted to ~$2.5 \times 10^6$ cells per ml in WB1, and 100 µl (~250,000 nuclei) was used for each primary antibody incubation. Primary antibody conjugated to an ABBC adapter (see the 'Antibody–DNA conjugation' section) was added and incubated overnight at 4 °C. Next, nuclei were washed two times with wash buffer 2 (WB2; 20 mM HEPES pH 7.5, 150 mM NaCl, 66.6 µg ml⁻¹ Spermidine, 1 × cOmplete protease inhibitor cocktail, 0.05% saponin) and resuspended in 200 µl of WB2 containing Hoechst 34580 (Sigma, 63493) at 1 µg ml⁻¹. Nuclei were incubated for 1 h at 4 °C. Finally, nuclei were washed two times with WB2 and resuspended in 500 µl of WB2 before FACS.

**Secondary antibody–DNA conjugates.** Permeabilized nuclei (or cells) were counted on a TC20 Automated Cell Counter. Nuclei were diluted to ~$2.5 \times 10^6$ cells per ml in WB1, and 200 µl (~500,000 nuclei)

was used for each primary antibody incubation. Primary antibody (unconjugated) was added and nuclei were incubated overnight at 4 °C. A control sample without primary antibody was taken along. Next, nuclei were washed two times with WB2 and resuspended in 200 µl of WB2 containing Hoechst 34580 at 1 µg ml⁻¹. Secondary antibody conjugated to an ABBC adapter (see the 'Antibody–DNA conjugation' section) was added (2 µg ml⁻¹) and incubated for 1 h at 4 °C. Finally, nuclei were washed two times with WB2 and resuspended in 500 µl of WB2 before FACS.

**BM antibody–fluorophore conjugate incubations.** Hoechst staining was omitted for BM cells. Following primary antibody–DNA conjugate incubation, BM cells were washed once with WB2 (0.05% Tween20 instead of saponin) and resuspended in 400 µl of WB2 containing 5% Blocking Rat Serum (Sigma, R9759) per $1 \times 10^6$ cells. Cells were incubated with a set of commercial antibody–fluorophore conjugates against BM surface markers. Incubations were performed for 30 min at 4 °C. Samples were kept dark from this point onwards. Finally, cells were washed once with WB2 and resuspended in 1 ml of WB2 before FACS.

## FACS

Nuclei (or cells) were pipetted through a Cell Strainer Snap Cap into a Falcon 5-ml Round Bottom Polypropylene Test Tube (Fisher Scientific, 10314791) before sorting on a BD Influx, BD FACSJazz or Beckman Coulter Cytoflex SRT cell sorter. Nuclei were sorted in G1/S cell-cycle phase, based on Hoechst. For BM cells, gates were set for each of the selected cell types using individually stained samples (Extended Data Fig. 8a) and all cell-cycle phases were included. For 1,000-cell samples, nuclei were sorted into a PCR tube strip containing 5 µl of 1 × CutSmart buffer (NEB, B7204S) per well, volume after sorting ~7.5 µl per tube. For samples with 100 cells or fewer, nuclei or cells were sorted into 384-well PCR plates (BioRad, HSP3831) containing 200 nl of 1 × CutSmart buffer and 5 µl of mineral oil (Sigma, M8410) per well. Plates were sealed with aluminum covers (Greiner, 676090).

## MAbID procedure

**Manual preparation of MAbID samples.** Samples containing 1,000 nuclei were processed in PCR tube strips. Samples were spun in a table-top rotor between incubation steps. Then, 2.5 µl of Digestion-1 mix (MseI (12.5 U, NEB, R0525M) and/or MboI (12.5 U, NEB, R0147M) in 1 × CutSmart buffer) was added to a total volume of 10 µl per tube, including 7.5-µl sorting volume. Samples were incubated in a PCR machine for 3 h at 37 °C. Next, 5 µl of rSAP mix (rSAP (1 U, NEB, M0371L) in 1 × CutSmart buffer (for MseI/NdeI digestions) or 1 × NEBuffer 3.1 (NEB, B7203S) (for MboI/BglII digestions)) was added to a total volume of 15 µl per tube. Samples were incubated for 30 min at 37 °C and 3 min at 65 °C before transfer to ice. Next, 5 µl of Digestion-2 mix (NdeI (5 U, NEB, R0111L) and/or BglII (5 U, NEB, R0144L) in 1 × CutSmart buffer (for MseI/NdeI digestions) or 1 × NEBuffer 3.1 (for MboI/BglII digestions)) was added to a total volume of 20 µl per tube. Samples were incubated for 1 h at 37 °C. Then, 6 µl of Ligation-1 mix (3.75 U T4 DNA ligase (Roche, 10799009001), 33.3 mM dithiothreitol (Invitrogen, 707265), 3.33 mM ATP (NEB, P0756L) in 1 × Ligase Buffer (Roche, 10799009001)) was added to a total volume of 26 µl per tube. Samples were incubated for 16 h at 16 °C. Next, 4 µl of Lysis mix (Proteinase K (5.05 mg ml⁻¹, Roche, 3115879001), IGEPAL CA-630 (5.05%, Sigma, I8896) in 1 × CutSmart buffer) was added to a total volume of 30 µl per tube. Samples were incubated for 4 h at 56 °C, 6 h at 65 °C and 20 min at 80 °C. Then, 10 µl of Digestion-3 mix (5 U NotI-HF (NEB, R3189L) in 1 × CutSmart buffer) was added to a total volume of 40 µl. Samples were incubated for 3 h at 37 °C. Next, 2.5 µl of SBC adapter (550 nM; see the section 'ABBC and SBC adapters') was added to a concentration of ~25 nM during ligation. Then, 12.5 µl of Ligation-2 mix (6.25 U T4 DNA ligase, 34 mM dithiothreitol, 3.4 mM ATP in 1 × Ligase Buffer) was added to a final volume of 55 µl. Samples were incubated for 12 h at 16 °C and 10 min at 65 °C.

**Robotic preparation of scMAbID samples.** First, 384-well PCR plates with sorted nuclei or cells were processed using a Nanodrop II robot at 82.7 kPa (12 psi) pressure (BioNex) for adding all mixes. Indicated volumes are per well. Increasing the reaction volumes to a pipetable range (for example, two- or threefold) to circumvent using liquid-handling robots is not anticipated to influence scMAbID performance. Between handling, plates were spun for 2 min at 1,000g at 4 °C. Then, 200 nl of Digestion-1 mix (MseI (0.5 U) and/or MboI (0.5 U) in 1× CutSmart buffer) was added to a total volume of 400 nl per well. Plates were incubated in a PCR machine for 3 h at 37 °C. Next, 200 nl of rSAP mix (rSAP (0.04 U) in 1× CutSmart buffer (for MseI/NdeI digestions) or 1× NEBuffer 3.1 (for MboI/BglII digestions)) was added to a total volume of 600 nl per well. Plates were incubated for 30 minutes at 37 °C and 3 min at 65 °C before transfer to ice. Then, 200 nl of Digestion-2 mix (NdeI (0.2 U) and/or BglII (0.2 U) in 1× CutSmart buffer (for MseI/NdeI digestions) or 1× NEBuffer 3.1 (for MboI/BglII digestions)) was added to a total volume of 800 nl per well. Plates were incubated for 1 h at 37 °C. Next, 240 nl of Ligation-1 mix (0.15 U T4 DNA ligase, 33.3 mM dithiothreitol, 3.33 mM ATP in 1× Ligase Buffer) was added to a total volume of 1,040 nl per well. Plates were incubated for 16 h at 16 °C. Then, 160 nl of Lysis mix (Proteinase K (5.05 mg ml$^{-1}$), IGEPAL CA-630 (5.05%) in 1× CutSmart buffer) was added to a total volume of 1,200 nl per well. Plates were incubated for 4 h at 56 °C, 6 h at 65 °C and 20 min at 80 °C. Next, 400 nl of Digestion-3 mix (0.2 U NotI-HF in 1× CutSmart buffer) was added to a total volume of 1,600 nl per well. Plates were incubated for 3 h at 37 °C. Then, 150 nl of SBC adapter (110 nM, see the section 'ABBC and SBC adapters') was added using a Mosquito HTS robot (TTP Labtech) to a concentration of ~7.5 nM during ligation. Next, 450 nl of Ligation-2 mix (0.25 U T4 DNA ligase, 37.8 mM dithiothreitol, 3.78 mM ATP in 1× Ligase Buffer) was added to a final volume of 2,200 nl. Plates were incubated for 12 h at 16 °C and 10 min at 65 °C.

**Library preparation**
Samples were pooled, either 2–4 1,000-nuclei samples or one 384-well plate, for combined in vitro transcription (IVT). For 384-well plates, mineral oil was removed by spinning the sample for 2 min at 2,000g and transferring the liquid phase to a clean tube, which was repeated three times. After pooling, samples were incubated for 10 min with 1.0 volume of CleanNGS magnetic beads (CleanNA, CPCR-0050), diluted 1:4 (1,000-nuclei samples) to 1:10 (384-well plate) in bead binding buffer (20% PEG 8000, 2.5 M NaCl, 10 mM Tris–HCl, 1 mM EDTA, 0.05% Tween20, pH 8.0 at 25 °C). Samples were placed on a magnetic rack (DynaMag-2, ThermoFisher, 12321D) to wash beads two times with 80% ethanol before allowing beads to dry before resuspending in 8 µl of water. IVT was performed by adding 12 µl of IVT mix from the MEGAScript T7 kit (Invitrogen, AM1334) for 14 h at 37 °C. Library preparation was subsequently performed as described previously[13,58], using 5 µl of amplified RNA (aRNA) and 8–11 PCR cycles, depending on aRNA yield. Purified aRNA from different IVT reactions was pooled before proceeding with complementary DNA synthesis. Libraries were run on an Illumina NextSeq500 platform (high output 1 × 75 base pairs (bp)) or an Illumina NextSeq2000 platform (high output 1 × 100 bp or 2 × 100 bp).

**Raw data processing**
Reads of the raw sequencing output conform to a MAbID-specific layout of 5′-[3-nt UMI][4-nt SBC part 1][3-nt UMI][4-nt SBC part 2] AGGGCCGC[8-nt ABBC][genomic sequence]-3′. Raw R1 reads were demultiplexed on the expected barcode-sequences using CutAdapt 3.0 (ref. 59), with the following custom settings. First, we allow only matches with at least 29-nt overlap and keep only reads directly starting with the adapter (that is, an anchored 5′ adapter). The maximum error rate setting of 2 retains reads with (1) two mismatches in the specified adapter sequence (ignoring UMI) and (2) a 1-nt insertion or deletion (indel) at the read start due to digestion-ligation or sequencing(-library) errors.

Demultiplexed reads are parsed through a custom script to classify reads on correct adapter-structures on a seven-tiered range. Reads in class 1 adhere perfectly to the barcode-expectations, while class 7 reads only contain the AGGGCCGC-sequence at the expected location. Reads typically fall into class 1 (average for Fig. 1: 95%). This classification allows fine control over which reads are retained. For this manuscript, we allow only classes 1 and 2 (1-nt indel in the first UMI) to ensure the highest possible quality. Finally, the script creates a fastq.gz-file, adding the UMI-sequences to the read-ID for downstream processing and removing of adapter sequence.

**Sequence alignments**
Demultiplexed and filtered reads were processed similarly to Rooijers et al.[13,58], with the additional flexibility to set the selected restriction site motif. Briefly, reads are aligned using Bowtie v.2.4.1 (ref. 60) in unpaired mode, using default end-to-end parameters. We used the UCSC hg19 reference genome for K562 samples and the NCBI mm10 reference genome for mESC/early NPC/BM samples (references were downloaded from https://benlangmead.github.io/aws-indexes). Alignments are sorted and filtered (mapping quality lower than 10) with samtools. Moreover, reads not mapping at the expected ligation site (5′ for the MboI GATC motif or 5′ + 1 for the MseI TTAA motif) were discarded with a custom script adapted from Rooijers et al.[13,58].

For reads originating from mixed-species single-cell samples (for example, K562 cells and mESCs/early NPCs/BM cells), a new hybrid reference genome was built by concatenating hg19 and mm10. Aligned reads were subsequently mapped to the individual references for further downstream processing by Bowtie using the --very-sensitive -N 1 parameters. Mouse allele-specific reads were assigned by mapping mm10 reads to 129/Sv and Cast/Eij reference genomes. We designated reads to one of the genotypes if it mapped better (that is, lower edit-distance or higher alignment score) to one of the references.

**Public data**
For the K562 analyses, we downloaded the ChromHMM[61,62] calls and several ChIP–seq datasets from the ENCODE portal[63,64] with the following identifiers: ENCFF001SWK, ENCFF002CKI, ENCFF002CKJ, ENCFF002CKK, ENCFF002CKN, ENCFF002CKY, ENCFF002CUS, ENCFF002CTX, ENCFF002CUU, ENCFF002CUN, ENCFF010PHG, ENCFF312LYO, ENCFF444SGK, ENCFF689TMV, ENCFF745HXR, ENCFF827GEM, ENCFF834YLI. K562 RNA-seq (ENCFF401KET) and ATAC-seq datasets (ENCFF055NNT) were also used. For comparisons with CUT&Tag, we referenced publicly available data from Kaya-Okur et al.[27,65] and Janssens et al.[66]: GSM4842201, GSM3536514, GSM3536515, GSM3536516, GSM3536518, GSM3536522, GSM4308161. K562 LAD-annotations were downloaded from the 4D Nucleome project[67] (4DNFIX4BXSIM) and converted to hg19-coordinates with the LiftOver utility of UCSC (https://genome.ucsc.edu/cgi-bin/hgLiftOver), while the LAD-annotations of Peric-Hupkes et al.[68,69] were used for the analyses on mouse (mm10) datasets.

For benchmarking, data from MulTI-Tag[26] and NTT-seq[24] were downloaded from Zenodo. Signac-objects[24,70] were loaded in R to obtain counts per epitope and cell. A summary file of counts per cell and epitope of Multi-CUT&Tag[21] was obtained from Yeung et al.[71]. Cell-by-count matrices for nano-CUT&Tag[23] were downloaded (https://cells.ucsc.edu/?ds=mouse-epi-juv-brain). As a reference dataset for defining BM marker genes, we used H3K4me3 sortChIC[45] data (GSM5018603).

**General filtering**
Aligned reads are UMI-flattened and counted per restriction site, similar to scDam&T-seq[13,58]. We allowed up to 1,000 UMIs per site for MAbID and up to 2 UMIs per site for scMAbID. UMI counts were binned and stored into singleCellExperiment-containers[72]. Counts in bins overlapping regions of known problematic nature (that is, blacklist-regions[73])

or low mappability are set to zero. scMAbID samples for K562 cells, mESCs and early NPCs were filtered for a minimum of 800 UMIs per cell and 64 UMIs per epitope in each cell (unless otherwise indicated). BM scMAbID samples were filtered for a minimum of 256 UMIs per cell (unless otherwise indicated).

Since the majority of the analyses were performed in R, we created an R package (mabidR) to load, normalize and analyze the datasets. Technical and biological replicates were merged after verification that separate datasets were of high quality and in agreement. Genome browser tracks for bulk MAbID data represent positive $\log_2$(observed/expected) values ($\log_2$(O/E)) of merged replicate datasets.

### Accessibility normalization

Normalization of the data was performed by calculating reads per kilobase million values (RPKM) for both samples and control (see equation 1) and calculating the fold change over control with a pseudocount value of 1 (see equation 2).

$$RPKM_{bin,sample} = \frac{TPM_{bin,sample}}{\frac{1}{1,000} \text{ bin size}};$$

$$\text{where } TPM_{bin,sample} = \frac{UMI_{bin,sample}}{\frac{1}{1,000,000} \sum_{i=bin}^{n} UMI_{i,sample}}$$

(1)

$$logcounts_{bin,sample} = normalized_{bin,sample}$$

$$= \log_2\left(\frac{\varphi + RPKM_{bin,sample}}{\varphi + RPKM_{bin,control}}\right); \text{where } \varphi = 1$$

(2)

Raw counts of control and H3K27me3 data were compared with TTAA motif coverage and public ATAC-seq signal at 5-kb resolution. Pearson's $r$ correlation coefficients were calculated between ATAC-seq and H3K27me3 MAbID signal (raw and normalized). Furthermore, we aligned control, Lamin B1 and H3K4me3 MAbID signals over LADs (10-kb resolution) and active TSS regions (5-kb resolution) to ascertain that the normalization was neither too lenient nor too harsh.

### Correlation analyses

Pearson's $r$ correlation coefficients between bulk normalized MAbID, ENCODE ChIP–seq (signal $P$ value), CUT&Tag (counts) and DamID (Dam-only normalized) datasets were calculated at 5-kb resolution. Negative values (due to either biological or technical reasons) were omitted. Bins with low variability ($\mu \pm 2\sigma$) or in blacklisted regions were omitted.

### Peak calling

Peak calling was performed at 5-kb resolution using hidden Markov models. We modeled the hidden states with a Gaussian distribution family: ($\mu = [0,1]$, $\sigma = [0,1]$) for the raw data and ($\mu = [-1,0,1]$, $\sigma = [1,1,1]$) for the normalized data using depmixS4 (ref. 74). Viterbi-decoded global state sequences were used to segment the genome. The two-state model was used to segment ChIP–seq data to check for robustness and accuracy.

### FRiP and SEiP

To evaluate the sensitivity and specificity of MAbID, we calculated the FRiP. The input regions (peaks) were derived from the 18-state K562 ChromHMM model[61,62] and public LAD calls[67]. All peaks underwent filtering to exclude small peaks (~5 kb, active TSS > 1.5 kb), and states were merged in the cases of Enhancers (Enh*), Polycomb domains (ReprPC*), Active TSS (TssA & TssFlnk*) and Transcription (Tx*). We allowed for at most 100,000 entries.

Considering that MAbID data are typically normalized, we devised the SEiP metric which incorporates both positive and negative non-integer values. Unlike FRiP, SEiP employs the average signal over peaks instead of the read-sum. To obtain the expected background

distributions, we employed a 1,500-fold randomization strategy using randomizeRegions from the regioneR package[75]. This background distribution was then used to scale the observed SEiP scores:

$$SEiP_{scaled} = \frac{SEiP_{observed} - \mu_{permuted}}{\sigma_{permuted}}$$

To compare the resulting metrics, we referenced public datasets of ChIP–seq, CUT&Tag and DamID at a resolution of 5 kb.

### Signal enrichment

Enrichment computations were performed using computeMatrix from Deeptools v.3.5.1 (ref. 76). Polycomb-group domains were generated by merging 200-bp regions of the ChromHMM states 16–17, allowing a gap of 10 kb and filtering the resulting regions on a minimal size of 100 kb. Expression-based stratifications of gene bodies were made by splitting RNA-seq TPM-values on [0, 33.4, 66.7, 100] percentiles, resulting in low/mid/high categories, respectively.

### Dimensionality reduction analyses

The input for the UMAP analyses was $\log_2$(O/E) for the bulk and ISP approaches, and $\log_1 p$(UMI counts) for single-cell samples, $Z$-score normalized before principle component analysis (PCA). To limit method-specific accessibility biases dominating dimensionality reductions, one public dataset (DamID or ChIP–seq) was included per chromatin type. The cross-epitope K562 UMAP contained epitope samples with more than 150 UMIs, belonging to a cell with more than 800 UMI counts. Only bins with counts for more than ten cells were used. For the mouse version, we kept the top 300 highest-depth samples for each epitope per cell type.

Pan-epitope mouse UMAPs were generated as described in Zhu et al.[16]: for each 100-kb [bin × cell] UMI-matrix per epitope, we computed Jaccard-distances ($D_{cell\,i,cell\,j} = 1 - Jaccard_{cell\,i,cell\,j}$). We rescaled each D-matrix to have values between 0 and 1 and summed these, whereafter PCA and UMAP were performed as above.

BM scMAbID datasets were filtered to include cells with at least 256 counts across epitopes. Pan-epitope ISP UMAPs were generated by first summing the per-cell counts per cell type and plate at 20-kb resolution and concatenating the vectors. We normalized for sequencing depth per plate with Seurat's SampleUMI function[77]. PCA was performed on the log-transformed normalized data, followed by UMAP on principle components 1–18.

BM scMAbID diffusion embedding was performed using Destiny[44] on the epitope-concatenated 1-Mb count matrix. Bins with fewer than 20 counts were removed to limit computational time. PCA was performed on the scaled log-transformed and depth-normalized matrix, followed by Destiny on principle components 1–4.

### Benchmarking scMAbID

To assess the signal across different single-cell methods, the FRiP was calculated on applicable states of the 18-state K562 ChromHMM dataset[62]. LAD-annotations[67] were included as a high-quality constitutive heterochromatin annotation. Counts in peaks were tallied using Signac's[70] FeatureMatrix().

### Information gain

Information gain was calculated by subtracting the weighted entropies of each cluster from the complete entropy. Entropy is defined as $-1 \times \sum f \log_2(f)$, where $f$ is the vector of cluster frequencies.

### BM marker gene identification and analysis

As a reference for defining BM marker genes, publicly available H3K4me3 single-cell sortChIC[45] data were downloaded. Counts per million (CPM) were calculated for each cell type and each promoter region (2 kb upstream to 500 bp downstream of TSS). Per cell type, the

enrichment score was calculated per gene against the average CPM of the other cell types:

$$\text{enrichment} = \log_2\left(\frac{\text{CPM}_{\text{gene}_i,\text{this cell type}}}{\text{CPM}_{\text{gene}_i,\text{other cell types}}}\right)$$

Gene sets were generated and ordered along this score to select top X genes per set. Gene Ontology pathway analysis was performed using Limma[78] and topGO[79].

The combined counts of H3K4me1 and H3K27ac were loaded into Signac[70] and we calculated gene X cell count matrices with GeneActivity(). After depth-normalization with LogNormalize(), marker genes were identified using FindAllMarkers() with default parameters aside from min.pct = 0.01. Genes were filtered on P values smaller than 0.001.

### Reporting summary

Further information on research design is available in the Nature Portfolio Reporting Summary linked to this article.

### Data availability

All relevant data supporting the findings of this study are available within the article and its supplementary information files. All raw sequencing MAbID data and processed files are available on the Gene Expression Omnibus (GEO) database under accession code GSE218476. Any other datasets mentioned in the manuscript were obtained and generated using the computational protocols described in Methods. Public ChIP–seq, RNA-seq and ATAC-seq data used in this study can be found on the ENCODE[63,64] portal with the following identifiers: ChIP–seq: ENCFF001SWK, ENCFF002CKI, ENCFF002CKJ, ENCFF002CKK, ENCFF002CKN, ENCFF002CKY, ENCFF002CUS, ENCFF002CTX, ENCFF002CUU, ENCFF002CUN, ENCFF010PHG, ENCFF312LYO, ENCFF444SGK, ENCFF689TMV, ENCFF745HXR, ENCFF827GEM, ENCFF834YLI; RNA-seq: ENCFF401KET; ATAC-seq: ENCFF055NNT. Public CUT&Tag[27,65,66], MulTI-Tag[26] and NTT-seq[24] datasets used in this study can be found on the GEO database with the following accession codes: GSM4842201, GSM3536514, GSM3536515, GSM3536516, GSM3536518, GSM3536522, GSM4308161. Public 4D Nucleome project[67] data used in this study can be downloaded with the following identifier: 4DNFIX4BXSIM. Public sortChIC[45] data used in this study can be found on the GEO database with the following accession code: GSM5018603. Source data are provided with this paper.

### Code availability

All relevant code supporting the findings of this study is available on https://github.com/KindLab/MAbID, including the mabidR R package.

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

### Acknowledgements

We thank all members of the Kind laboratory for their comments throughout the project and their critical reading of the manuscript. We thank K. Mulder for his input on generating antibody–DNA conjugates and P. Zeller for comments and suggestions during the development of the MAbID technique. This work was funded by an ERC Consolidator grant (no. EU ERC CoG-101002885 FateID) and an NWO-ENW VIDI grant (no. 161.339). The Oncode Institute is partially funded by the KWF Dutch Cancer Society. The laboratory of H.K. is supported by MEXT/JSPS KAKENHI (grant nos. JP18H05527 and JP21H04764), the Japan Science and Technology Agency (grant no. JPMJCR16G1) and the Japan Agency for Medical Research and Development (grant no. JP22ama121020). In addition, we thank the Hubrecht Sorting Facility, especially R. van der Linden, and the Utrecht Sequencing Facility (USEQ) for providing sequencing services and data. USEQ is subsidized by the University Medical Center Utrecht and the Netherlands X-omics Initiative (NWO project no. 184.034.019).

### Author contributions

S.J.A.L., K.L.d.L. and J.K. designed the method. S.J.A.L. developed the method and performed all experiments. R.E.v.B. isolated mouse bone marrow and assisted with BM cell-surface marker staining optimizations. R.H.v.d.W., T.K. and S.J.A.L. performed preliminary computational analyses. R.H.v.d.W. performed all formal computational analyses and designed software. S.J.A.L. and R.H.v.d.W. validated and curated the data. H.K. provided resources.

R.H.v.d.W. and S.J.A.L. performed data visualization. S.J.A.L. and J.K. wrote the original draft of the manuscript, which R.H.v.d.W. edited. J.K. conceived and supervised the study, was responsible for project administration and acquired funding.

## Competing interests

The authors declare the existence of a competing interest. J.K. and S.J.A.L. are inventors on a patent application (PCT/NL2022/050635, applicant: Koninklijke Nederlandse Akademie Van Wetenschappen) related to the MAbID technology. The other authors declare no competing interests.

## Additional information

**Extended data** is available for this paper at https://doi.org/10.1038/s41592-023-02090-9.

**Correspondence and requests for materials** should be addressed to Jop Kind.

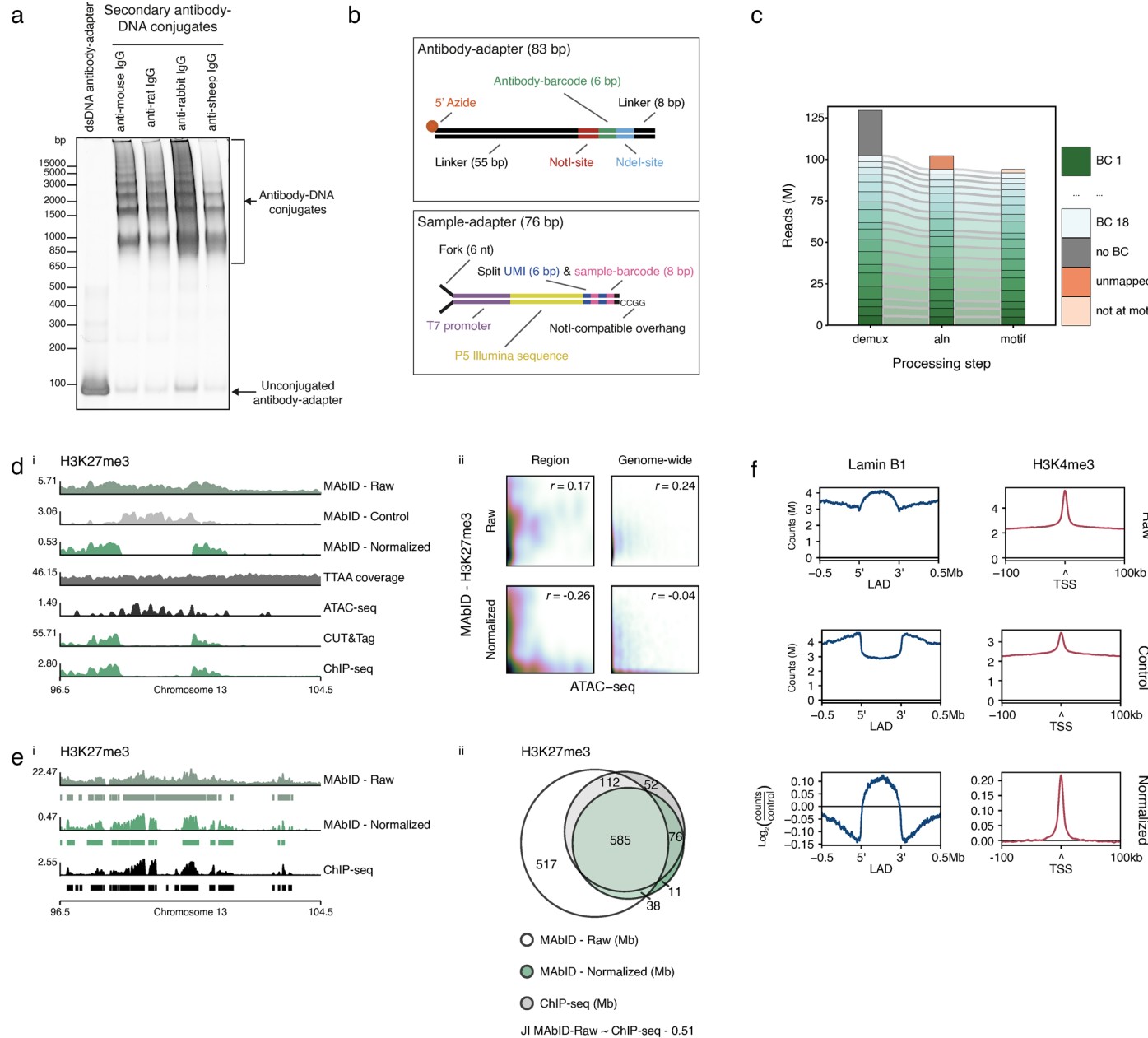

**Extended Data Fig. 1 | Overview of the MAbID method.** a) Representative gel electrophoresis analysis of secondary antibody-DNA conjugates targeting primary IgGs of different species-of-origin. Conjugates were separated on a native polyacrylamide gel with a 4-12% gradient. Unconjugated antibody-adapter was loaded as control. Antibody-DNA conjugate preparations were repeated at least 5 times with similar results. b) Designs of the antibody-adapter and sample-adapter. Top strand of the antibody-adapter has a 5' azide modification ($N_3$) for coupling to the antibody. Fork in the double-stranded sample-adapter was created by adding 6 nt non-complementary sequences on each strand. UMI, Unique Molecular Identifier; nt, nucleotide; bp, basepair. c) Barplot of read counts (M, million) retained per computational processing step. Different segments represent separate samples used (BC 1-18), identified by the combined presence of the sample (SBC) and antibody-barcode (ABBC) within the read. Demux, demultiplexing of reads on combined barcodes; aln, alignment of reads to the genome; motif, reads mapping to the TTAA sequence motif. d) i; Genome browser tracks comparing K562 MAbID samples of H3K27me3

(Raw and Normalized) and the control with TTAA motif coverage, ATAC-seq, CUT&Tag and ChIP-seq. 'Control' represents a combination of depth-normalized samples in which primary antibody was omitted, 'Raw' is the depth-normalized H3K27me3 signal, 'Normalized' is the raw sample normalized over the control. Y-axis reflects positive $\log_2$(counts/control) for MAbID and fold change (IP/input) for ChIP-seq and CUT&Tag. ii; Comparison of MAbID and ATAC-seq per bin of 'Raw' and 'Normalized' data, for the region in (i) and genome-wide. Pearson's $r$ correlation coefficients between MAbID and ATAC-seq data indicated. e) i; Genome browser tracks of H3K27me3 Raw and Normalized MAbID data with ChIP-seq. Called domains are indicated below. ii; Overlap (in Mb) between H3K27me3 domains called on MAbID Raw, MAbID Normalized and ChIP-seq data. JI, Jaccard Index. f) Average signal enrichment of MAbID samples (Raw, Control or Normalized) of Lamin B1 or H3K4me3 using secondary antibody-DNA conjugates, around LADs or TSS of active genes. Y-axis reflects counts/million (for control and raw) or $\log_2$(counts/control) (for normalized).

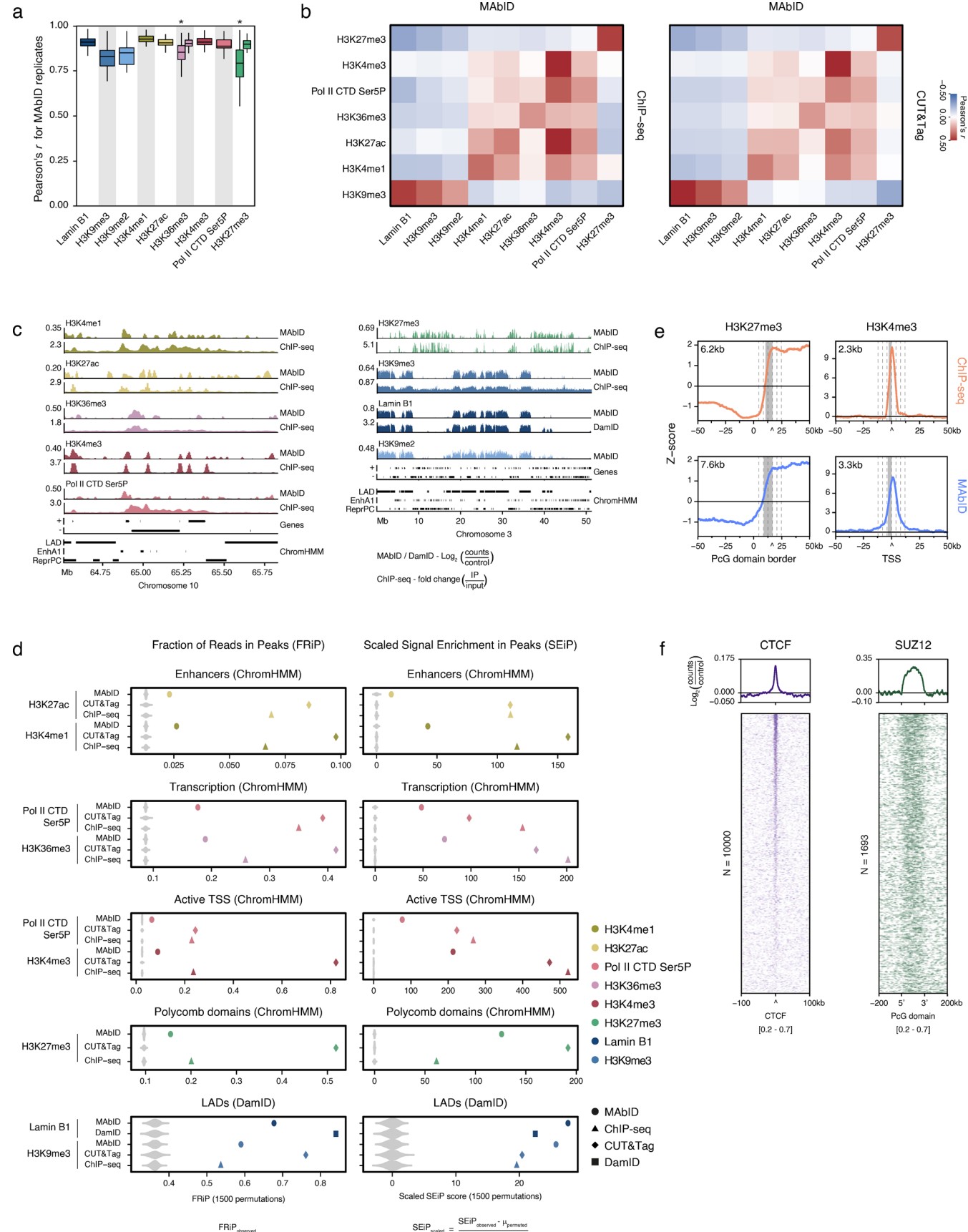

**Extended Data Fig. 2 | See next page for caption.**

**Extended Data Fig. 2 | Benchmarking MAbID to state-of-the-art methods.**
a) Boxplots showing the Pearson's *r* correlation coefficients between corresponding chromosomes of MAbID replicates. Asterisks (*) denote the correlation coefficients between MAbID samples using different primary antibodies against the same epitope. Boxplot represents the median (crossing line), minima, maxima, and interquartile range (bounds of box). n = 23 chromosomes. b) Heatmap of genome-wide Pearson's *r* correlation coefficients of MAbID with ChIP-seq (left) or CUT&Tag (right). c) Genome browser tracks of MAbID with ChIP-seq or DamID for active chromatin types on a narrow genomic scale (left) or inactive chromatin types on a broad genomic scale (right). Genes (+, forward; -, reverse) and ENCODE/ChromHMM domain calls (LAD, Lamina-associated domain; EnhA1, Active enhancer 1; ReprPC, Repressed PolyComb) are indicated. Y-axis reflects positive $\log_2$(counts/control) values for MAbID and DamID and fold change (IP/input) for ChIP-seq. d) FRiP (Fraction of Reads in Peaks) and scaled SEiP (Signal Enrichment in Peaks) scores comparing MAbID

samples with corresponding ChIP-seq, CUT&Tag and DamID samples. Scores are calculated over different ChromHMM states (Enhancers, Transcription (state 10, TranscriptionElongation), Active TSS, Polycomb domains) and LADs. e) Average signal enrichment of MAbID or ChIP-seq for H3K27me3 or H3K4me3 around Polycomb-group domain borders (PcG, ChromHMM, left) or TSS of active genes (right) respectively. Y-axis reflects Z-score normalized values of $\log_2$(counts/control) for MAbID and fold change (IP/input) for ChIP-seq. Top of the signal is called at the curve's inflection point, indicated by ^. Gray box highlights the 50% decay distance, noted in the top left corner. Dashed lines reflect linear steps of 4 kb distance from the top. f) MAbID signal enrichment of CTCF around ChIP-seq peaks (ENCODE) and SUZ12 around Polycomb-group domains (ChromHMM). Top - average enrichment, bottom - signal per genomic region (sorted on MAbID signal). N represents the number of genomic regions included per heatmap and the data range is indicated underneath.

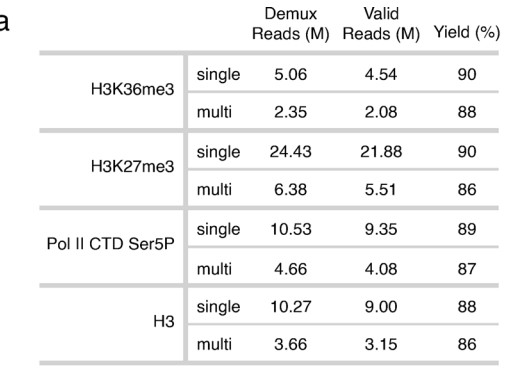

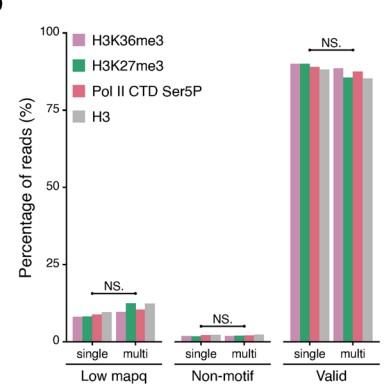

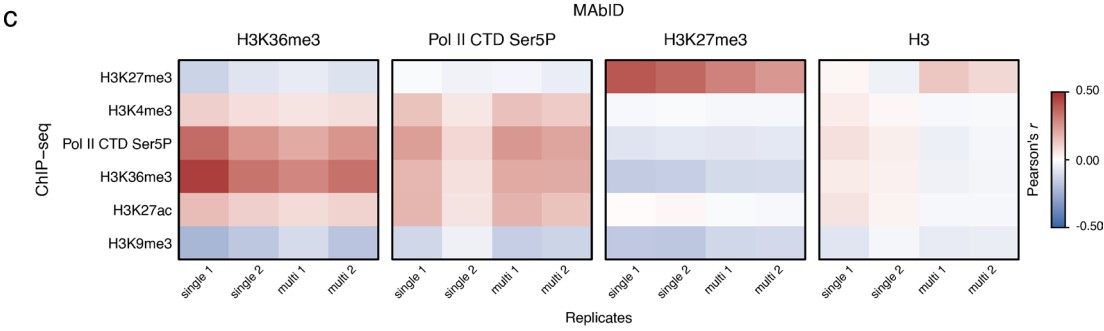

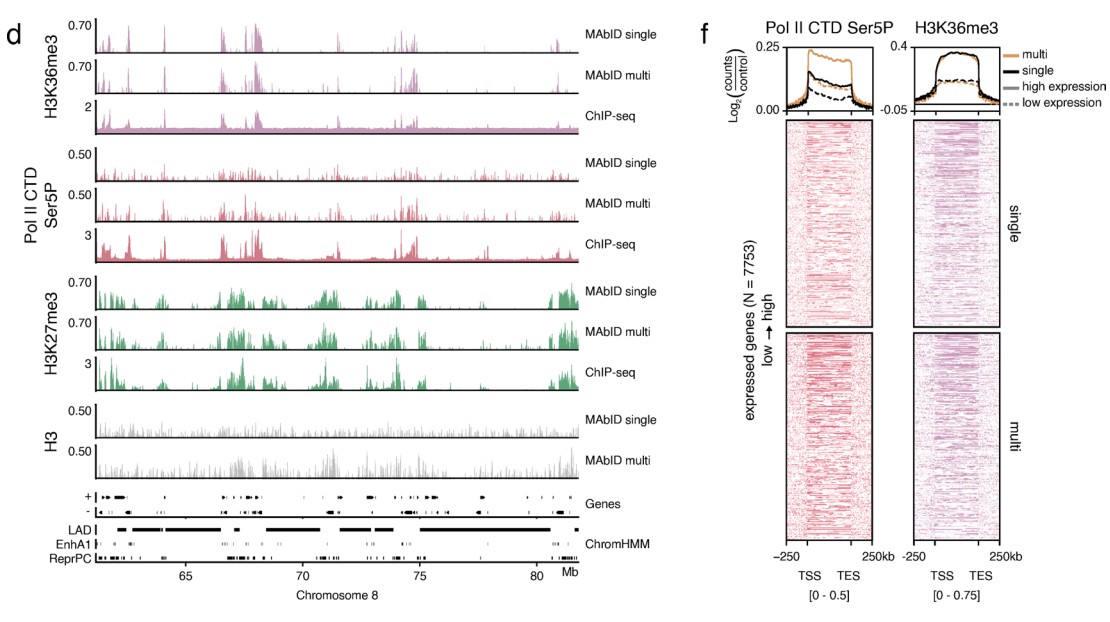

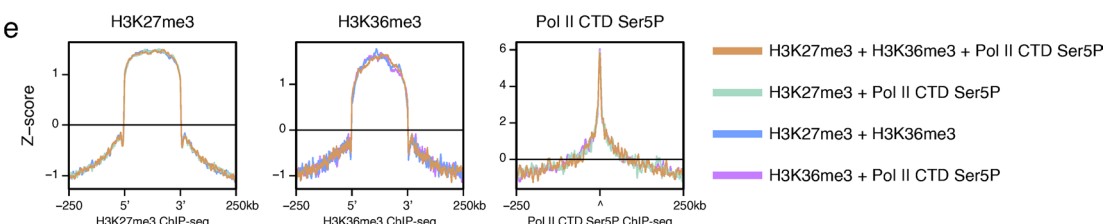

**Extended Data Fig. 3 | See next page for caption.**

**Extended Data Fig. 3 | Individual and multiplexed MAbID samples have similar data quality.** a) Table of number of demultiplexed reads (demux), valid reads (aligned at TTAA motif) and the resulting yield (% of valid in demux) for individual (single) or combined (multi) samples. M; million. b) Barplot showing percentage of reads lost or retained in the different computational processing steps comparing individual (single) or combined (multi) measurements. Low mapq, low mapping quality; Non-motif, not aligned at TTAA sequence motif; Valid, reads passing quality thresholds aligning at a TTAA sequence motif; NS, non-significant difference based on two-sided Wilcoxon signed-rank test. c) Heatmap showing the genome-wide Pearson's *r* correlation coefficient of MAbID replicates of individual (single) or combined (multi) measurements with ChIP-seq. d) Genome browser tracks comparing MAbID samples of individual (single) or combined (multi) measurements with ChIP-seq. Genes (+, forward; -, reverse) and ENCODE/ChromHMM domain calls (LAD, Lamina-associated domain;

EnhA1, Active enhancer 1; ReprPC, Repressed PolyComb) are indicated. Y-axis reflects positive $\log_2$(counts/control) values for MAbID and fold change (IP/input) for ChIP-seq. e) Average MAbID signal enrichment of H3K27me3, H3K36me3 and Pol II CTD Ser5P around the same domains/peaks called on ChIP-seq data, comparing MAbID samples generated with different combinations of primary antibodies and corresponding secondary antibody-DNA conjugates. Y-axis reflects Z-score normalized values of $\log_2$(counts/control). f) MAbID signal enrichment of individual (single) or combined (multi) measurements of Pol II CTD Ser5P and H3K36me3 around active genes. Genes were stratified on expression level by setting the top 50% as the "high"-group and the bottom 50% as the "low"-group. Top - average enrichment per group, bottom - signal per set of genes, ordered from high to low. Heatmaps are split for the individual (top) or combined (bottom) staining. N = 7553 genes included per heatmap, the data range is indicated underneath.

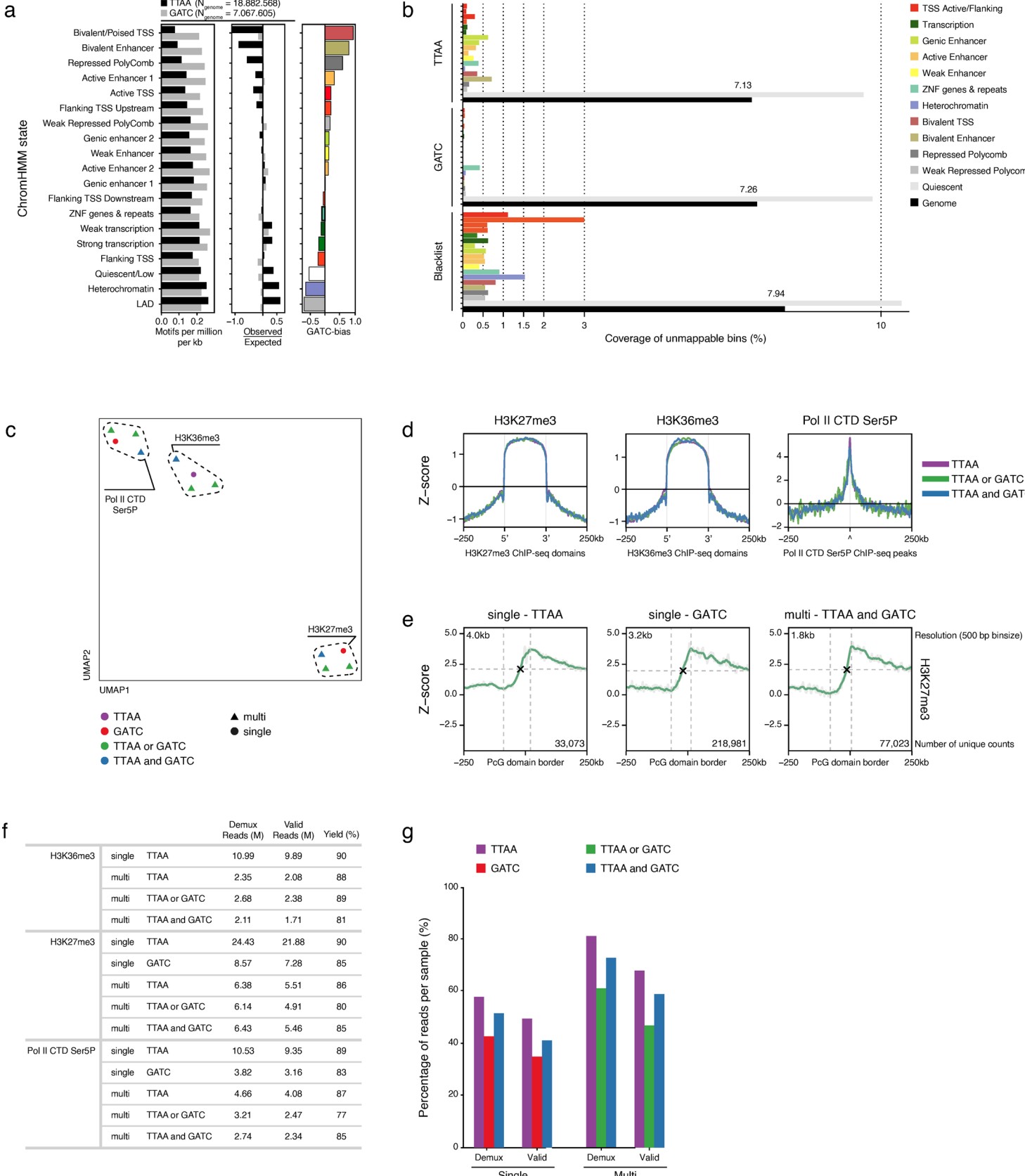

**Extended Data Fig. 4 | See next page for caption.**

**Extended Data Fig. 4 | MAbID is customizable to the genomic context of the epitope of interest.** a) Barplots showing the number of TTAA or GATC sequence motifs in the human genome distributed over ChromHMM states. Observed/Expected (O/E) is the $\log_2$ transformation of the number of motifs observed compared to the expected number based on the proportion of the genome per state. GATC-bias shows the difference between the O/Es for the GATC versus the TTAA motif. b) Barplot showing the genomic coverage (%) of unmappable bins across the different ChromHMM states and the whole genome (black, percentages indicated) for the TTAA motif, GATC motif or blacklisted regions[72,73]. c) UMAP embedding of MAbID replicates of individual (single) or combined (multi) measurements using different antibody-adapter types. Colouring on the targeted sequence motif, used primary antibodies are encircled. TTAA, NdeI-compatible antibody-adapter; GATC, BglII-compatible antibody-adapter; TTAA or GATC, NdeI- or BglII compatible antibody-adapter; TTAA and GATC, both types of antibody-adapter – each category indicating the antibody-adapter per epitope. d) Average MAbID signal enrichment of H3K27me3, H3K36me3 and Pol

II CTD Ser5P around the same domains/peaks called on ChIP-seq data, comparing MAbID samples from combined or individual measurements using different types of antibody-adapters. Y-axis reflects Z-score normalized $\log_2$(counts/control). e) Average signal enrichment of H3K27me3 MAbID samples around Polycomb-group domain borders (PcG, ChromHMM) for individual (TTAA; GATC) or combined (TTAA and GATC) measurements. Y-axis reflects Z-score normalized values of $\log_2$(counts/control). Vertical dashed lines - top and bottom of the signal; x - 50% decay distance, noted in the top left corner (kb). Bottom right values reflect number of unique counts per sample. f) Table of demultiplexed reads (demux), valid reads (aligned at TTAA or GATC motif) and the resulting yield (% of valid in demux) for samples using different types of antibody-adapters in individual or combined samples. M; million. g) Barplot showing percentage of demultiplexed (demux) and valid (aligned at motif) reads per total number of sequenced reads per sample, comparing different antibody-adapter types and individual versus combined measurements.

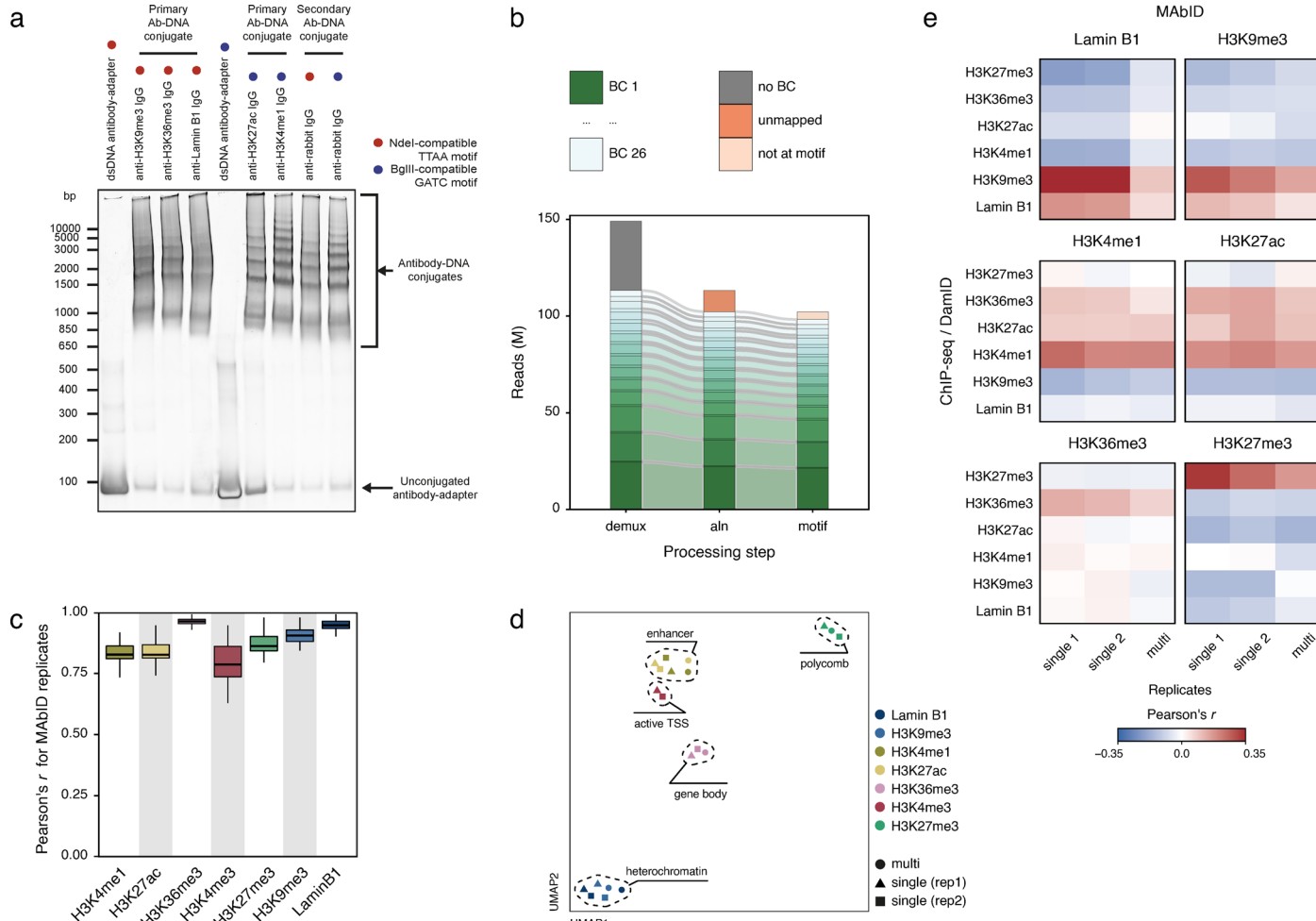

**Extended Data Fig. 5 | MAbID with primary antibody-DNA conjugates generates specific genomic profiles in individual and combined measurements.** a) Representative gel electrophoresis analysis of primary antibody-DNA conjugates targeting different epitopes. Conjugates were separated on a native polyacrylamide gel with a 4-12% gradient. Unconjugated antibody-adapter was loaded as control. Red and blue dots indicate the type of antibody-adapter used. Antibody-DNA conjugate preparations were repeated at least 5 times with similar results. b) Barplot of read counts (M, million) retained per computational processing step. Different segments represent separate samples used (BC 1-26), identified by the combined presence of the sample (SBC) and antibody-barcode (ABBC) within the read. Demux, demultiplexing of reads on combined barcodes; aln, alignment of reads to the genome; motif,

reads mapping to the TTAA or GATC sequence motif. c) Boxplots showing the Pearson's *r* correlation coefficient between corresponding chromosomes of MAbID replicates using primary antibody-DNA conjugates. Boxplot represents the median (crossing line), minima, maxima, and interquartile range (bounds of box). n = 23 chromosomes. d) UMAP of MAbID samples from combined (multi) or individual (single, two replicates) measurements with primary antibody-DNA conjugates. Colouring is based on the epitope, chromatin types are encircled. e) Heatmap showing the genome-wide Pearson's *r* correlation coefficient of MAbID replicates using primary antibody-DNA conjugates for individual (single) or combined (multi) measurements with different ChIP-seq or DamID (for Lamin B1) samples.

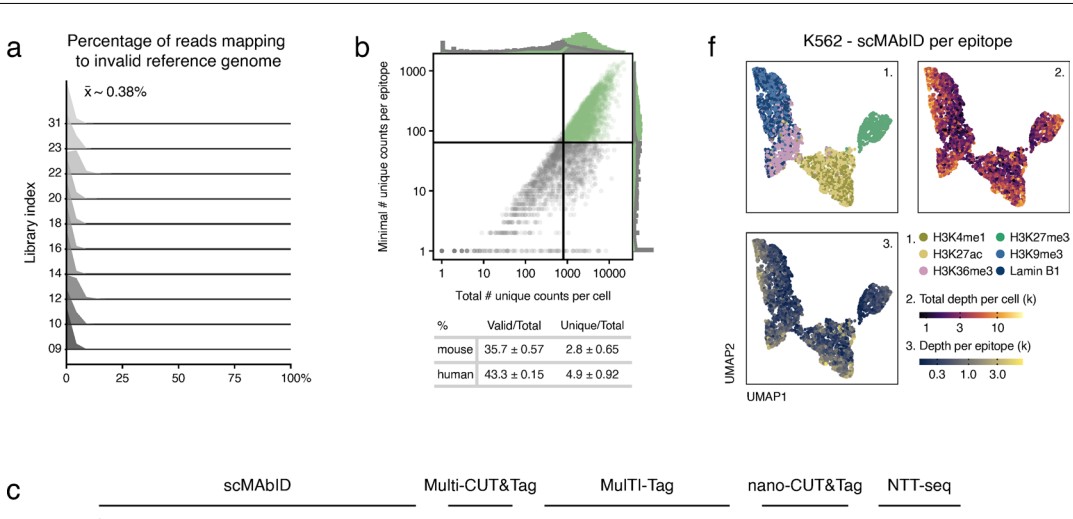

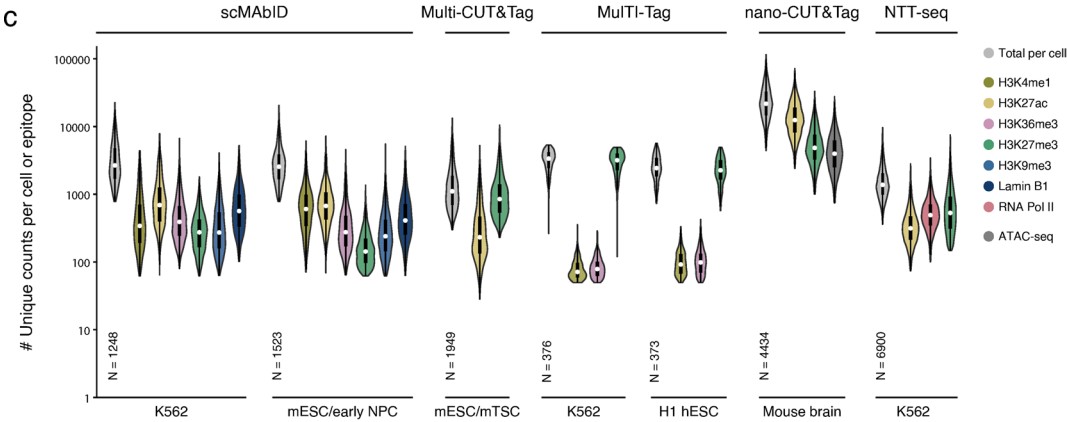

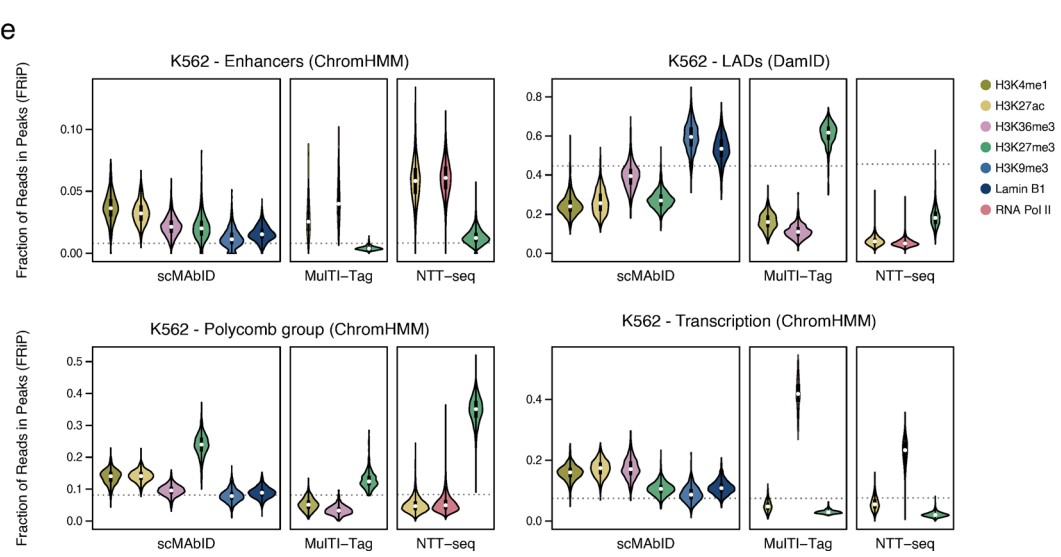

Extended Data Fig. 6 | See next page for caption.

**Extended Data Fig. 6 | scMAbID data of human and mouse single cells with six multiplexed measurements.** a) Percentages of reads per well mapping to the invalid reference genome per library index (mean = 0.38%). Calculations are based on wells containing cells of only one origin. b) Quality thresholds of scMAbID. Dot plot for K562, mESC and early NPC cells shows total number of unique counts per cell versus minimal number of unique counts per epitope per cell, with density plots indicating number of samples. Samples passing quality thresholds are highlighted in green. Table shows the yield (% of reads from total) for valid (at motif) or unique (based on UMI) counts for each species. c) Violin plots comparing the number of counts/reads per cell and per epitope/modality between scMAbID (K562, n = 1248; mESC/early NPC, n = 1523), Multi-CUT&Tag[21] (mESC/mTSC, n = 1949), MulTI-Tag[26] (K562, n = 376; H1 hESC, n = 373), nano-CUT&Tag[23] (mouse brain, n = 4434) and NTT-seq[24] (K562, n = 6900). Boxplots inside violin-plots show the minima, maxima, interquartile range (box bounds), and median (white dot). N, number of cells. d) Table comparing metrics of

scMAbID, Multi-CUT&Tag[21], MulTI-Tag[26], nano-CUT&Tag[23] and NTT-seq[24]. Input cells staining – Number of cells used for antibody incubations; Input cells loaded – number of cells sorted or loaded on a 10x Chromium/ICELL8 chip; Recovered cells – number of cells reported after quality filtering; % Recovery – percentage of 'Recovered cells' from 'Input cells loaded'; Reads per cell – total reads/counts/fragments per cell; Reads per epitope – range of reads/counts/ fragments per epitope per cell. Metrics are based on the reported values of each manuscript as well as the calculated read counts from the public datasets. e) FRiP scores per single K562 cell for scMAbID (n = 1248), MulTI-Tag[26] (n = 376) and NTT-seq[24] (n = 6900). FRiP scores are calculated across different ChromHMM and LAD (DamID) domains. f) UMAP of K562 scMAbID samples. Each dot represents one epitope measurement. Colouring is based on 1) the epitope, 2) total depth per cell or 3) depth per epitope. Only cells passing the threshold of 150 unique counts per epitope were included (n = 6729). k, thousands.

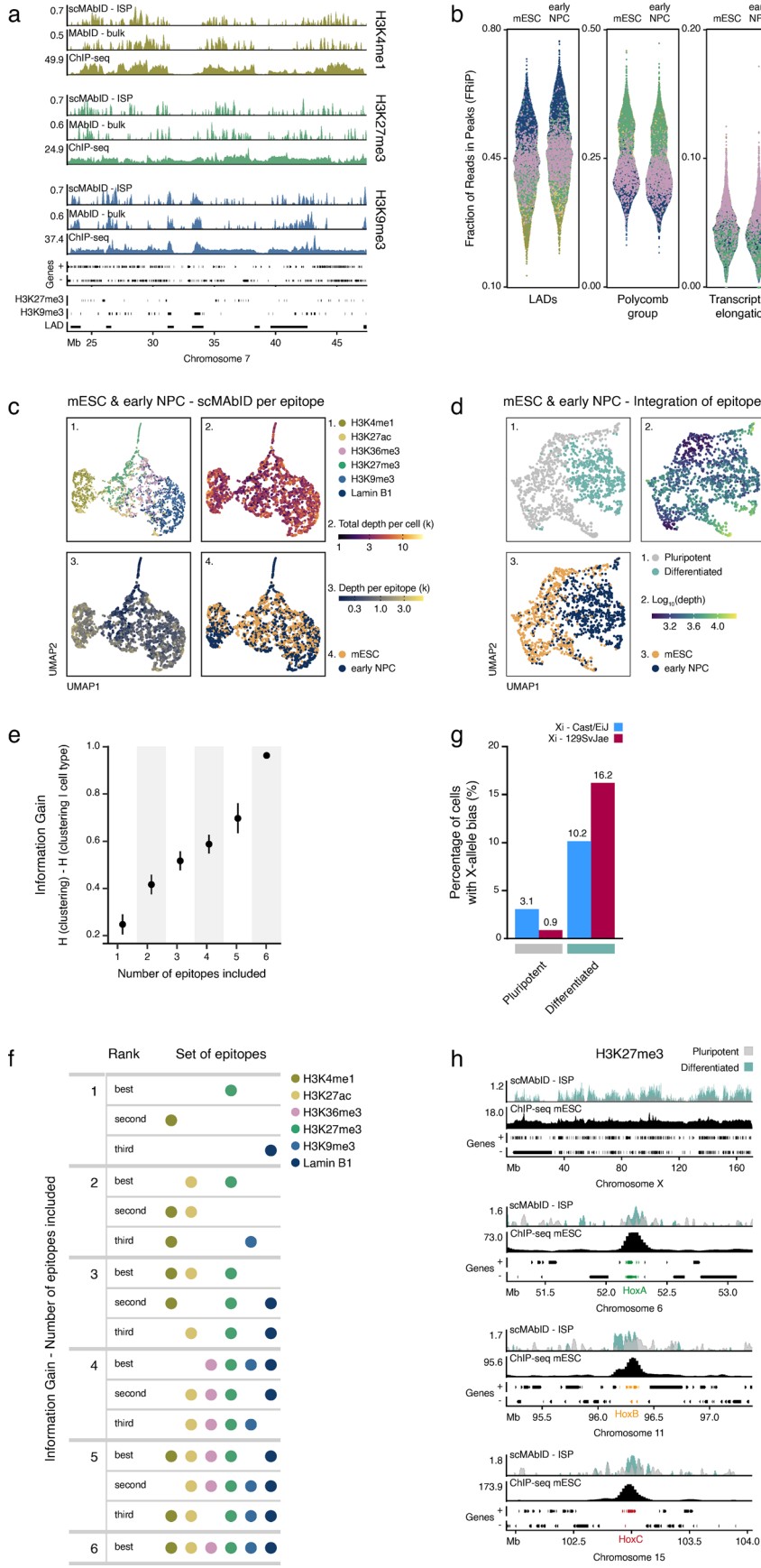

**Extended Data Fig. 7 | See next page for caption.**

**Extended Data Fig. 7 | Applying scMAbID to a mouse *in vitro* neural differentiation system.** ISP, *in silico* population. a) Genome browser tracks comparing mESC scMAbID ISP (n = 674) samples of H3K4me1, H3K27me3 and H3K9me3 with bulk MAbID and ChIP-seq. Genes and ENCODE domain calls are indicated. Y-axis reflects positive $\log_2$(counts/control) for MAbID and fold change (IP/input) for ChIP-seq. b) FRiP scores of each scMAbID epitope measurement per mESC and early NPC cell. FRiP scores are calculated across different ChromHMM and LAD domains. Plotting order from back to front – H3K4me1, H3K27ac, H3K9me3, Lamin B1, H3K27me3, H3K36me3. c) UMAP of mESC and early NPC scMAbID samples. Per cell type, the top 300 highest depth measurements per epitope are included (mESC, n = 1800; early NPC, n = 1800). Colouring is based on 1) epitope, 2) total depth per cell, 3) depth per epitope or 4) cell type of origin. k, thousands. d) UMAP of integrated mouse scMAbID samples (mESC, n = 674; early NPC, n = 849). Each dot represents one cell, colouring based on 1) Leiden algorithm cluster assignment, 2) total depth per cell

($\log_{10}$ transformed) or 3) cell type of origin. e) Information Gain (IG) for different numbers of epitopes included for data integration, calculated by comparing each resulting clustering to the gold standard (cells assigned as mESC$_{\text{pluripotent}}$ and early NPC$_{\text{differentiated}}$ using six integrated epitopes). Whiskers denote Standard Error of the Mean. Unique epitope-combinations, n = 6, 15, 20, 15, 6 and 1 respectively. f) Table showing sets of epitopes with the three highest IG values of (e) per number of included epitopes, colouring based on epitopes. g) Barplot with percentage of cells that have undergone XCI in the pluripotent or differentiated clusters, coloured on assigned inactive X-chromosome allele (Xi). Percentage values are indicated per bar (%). h) Genome browser tracks comparing scMAbID ISP H3K27me3 samples of pluripotent (n = 912) and differentiated (n = 611) clusters, along with mESC H3K27me3 ChIP-seq. Genomic regions are the X-chromosome and regions around Hox clusters A, B and C, highlighted with green, orange and red respectively. Y-axis reflects positive $\log_2$(counts/control) for MAbID and fold change (IP/input) for ChIP-seq.

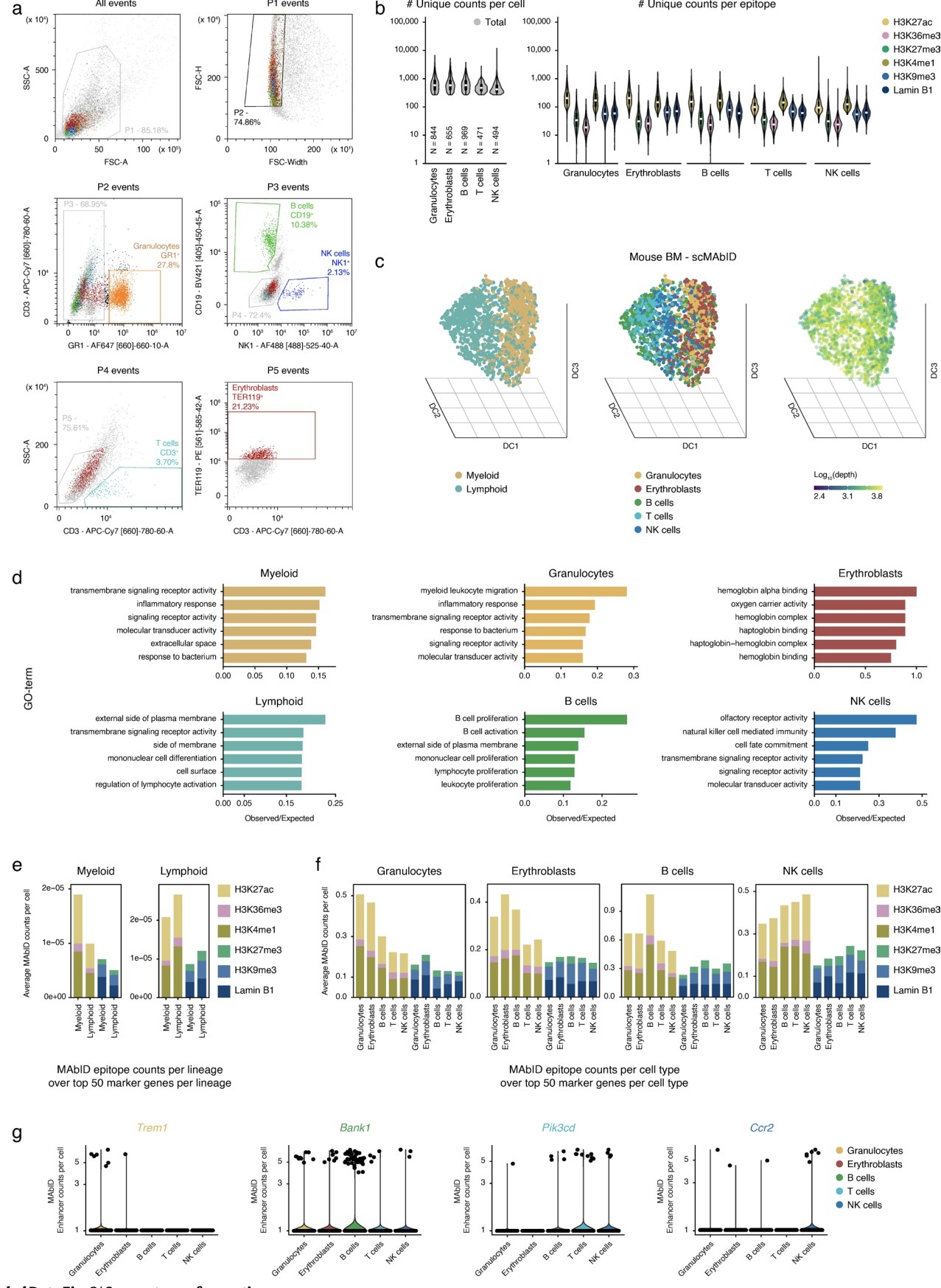

**Extended Data Fig. 8 | See next page for caption.**

**Extended Data Fig. 8 | scMAbID data of primary mouse bone marrow cells with combined measurement of six epitopes.** a) FACS gates to sort five specific cell types. Cells were generally gated on FSC and SSC values (gates P1 to P2) to obtain high quality single-cell samples, before passing through several gates to select each cell type. Title of each plot indicates the gate through which events passed previously. Percentages indicate events falling into the indicated gate compared to total events per plot. Granulocytes were selected as GR1$^+$, B cells as GR$^-$ CD19$^+$ NK1$^-$, NK cells as GR$^-$ CD19$^-$ NK1$^+$, T cells as GR$^-$ CD19$^-$ NK1$^-$ CD3$^+$ and Erythroblasts were more leniently selected as GR$^-$ CD19$^-$ NK1$^-$ CD3$^-$ TER119$^{high}$. b) Violin plots of total number of unique scMAbID counts per cell and per epitope for all cell types (Granulocytes, n = 844 cells; Erythroblasts, n = 655; B cells, n = 969; T cells, n = 471, NK cells, n = 494) after filtering on quality metrics. Boxplots inside violin-plots show the minima, maxima, interquartile range (box bounds), and median (white dot). N, number of cells. c) 3D Diffusion maps of single-cell scMAbID samples, with integrated epitope-measurements for each cell (n = 3433). Colouring is based on the lineage (left), cell type (middle) or depth (right, log$_{10}$ of the total depth per cell). DC, Diffusion component. d) GO-term analysis of differentially expressed marker genes for each lineage and cell type, based on publicly available sortChIC[45] data. The six terms with highest observed/expected are shown. e) Bargraph of the number of scMAbID counts per cell over the top 50 marker genes per lineage for each epitope. Active chromatin types and inactive chromatin types are stacked as individual bars for scMAbID counts of the myeloid- or lymphoid-lineage cells (x-axis). The title of each panel reflects the lineage-specific marker gene set. f) Bargraph as (e), per cell type instead of lineage. g) Count violins showing the number of combined scMAbID H3K4me1 and H3K27ac counts per cell over the *Trem1*[46,47], *Bank1*[48,49], *Pik3cd*[50,51] and *Ccr2*[52,53] genes.

# Reporting Summary

## Statistics

For all statistical analyses, confirm that the following items are present in the figure legend, table legend, main text, or Methods section.

| n/a | Confirmed | |
|---|---|---|
| ☐ | ☒ | The exact sample size (*n*) for each experimental group/condition, given as a discrete number and unit of measurement |
| ☐ | ☒ | A statement on whether measurements were taken from distinct samples or whether the same sample was measured repeatedly |
| ☐ | ☒ | The statistical test(s) used AND whether they are one- or two-sided *Only common tests should be described solely by name; describe more complex techniques in the Methods section.* |
| ☒ | ☐ | A description of all covariates tested |
| ☒ | ☐ | A description of any assumptions or corrections, such as tests of normality and adjustment for multiple comparisons |
| ☐ | ☒ | A full description of the statistical parameters including central tendency (e.g. means) or other basic estimates (e.g. regression coefficient) AND variation (e.g. standard deviation) or associated estimates of uncertainty (e.g. confidence intervals) |
| ☐ | ☒ | For null hypothesis testing, the test statistic (e.g. *F*, *t*, *r*) with confidence intervals, effect sizes, degrees of freedom and *P* value noted *Give P values as exact values whenever suitable.* |
| ☒ | ☐ | For Bayesian analysis, information on the choice of priors and Markov chain Monte Carlo settings |
| ☒ | ☐ | For hierarchical and complex designs, identification of the appropriate level for tests and full reporting of outcomes |
| ☐ | ☒ | Estimates of effect sizes (e.g. Cohen's *d*, Pearson's *r*), indicating how they were calculated |

*Our web collection on statistics for biologists contains articles on many of the points above.*

## Software and code

Policy information about availability of computer code

| Data collection | BD FACS Sortware (1.2.0.142)<br>CytExpert SRT (1.1)<br>Cutadapt (3.0)<br>Bowtie2 (2.2.9)<br>samtools (1.10)<br>scDamAndTools (1.0) |
|---|---|
| Data analysis | R (4.1)<br>scDamAndTools (1.0)<br>MAbIDR R package (V1.0)<br>Signac (1.9)<br>Destiny (3.14)<br>Seurat (4.9.9.9044)<br>https://github.com/KindLab/MAbID |

For manuscripts utilizing custom algorithms or software that are central to the research but not yet described in published literature, software must be made available to editors and reviewers. We strongly encourage code deposition in a community repository (e.g. GitHub). See the Nature Portfolio guidelines for submitting code & software for further information.

## Data

Policy information about availability of data

All manuscripts must include a data availability statement. This statement should provide the following information, where applicable:

- Accession codes, unique identifiers, or web links for publicly available datasets
- A description of any restrictions on data availability
- For clinical datasets or third party data, please ensure that the statement adheres to our policy

All relevant data supporting the findings of this study are available within the article and its supplementary information files. All raw sequencing data and processed files are made available on the GEO database under accession GSE218476. Any other datasets mentioned in the manuscript were generated using the computational protocols described in the methods.

Public ENCODE datasets: ENCFF001SWK, ENCFF002CKI, ENCFF002CKJ, ENCFF002CKK, ENCFF002CKN, ENCFF002CKY, ENCFF002CUS, ENCFF002CTX, ENCFF002CUU, ENCFF002CUN, ENCFF010PHG, ENCFF312LYO, ENCFF444SGK, ENCFF689TMV, ENCFF745HXR, ENCFF827GEM, ENCFF834YLI, ENCFF401KET, ENCFF055NNT
Public CUT&Tag, MulTI-Tag, and NTT-seq datasets: GSM4842201, GSM3536514, GSM3536515, GSM3536516, GSM3536518, GSM3536522, GSM4308161
Public 4D nucleome dataset: 4DNFIX4BXSIM
Public sortChIC dataset: GSM5018603

## Human research participants

Policy information about studies involving human research participants and Sex and Gender in Research.

| | |
|---|---|
| Reporting on sex and gender | This study did not include human research participants. |
| Population characteristics | This study did not include human research participants. |
| Recruitment | This study did not include human research participants. |
| Ethics oversight | This study did not include human research participants. |

Note that full information on the approval of the study protocol must also be provided in the manuscript.

# Field-specific reporting

Please select the one below that is the best fit for your research. If you are not sure, read the appropriate sections before making your selection.

☒ Life sciences      ☐ Behavioural & social sciences      ☐ Ecological, evolutionary & environmental sciences

For a reference copy of the document with all sections, see nature.com/documents/nr-reporting-summary-flat.pdf

# Life sciences study design

All studies must disclose on these points even when the disclosure is negative.

| | |
|---|---|
| Sample size | Two replicates were used for bulk MAbID experiments. These were collected as biological replicates, in which cells from established cell lines were grown independently for each replicate. Since these replicates show a high concordance (based on Pearson's r statistical test), no further replicates were included. For single-cell MAbID experiments, single biological samples (no biological replicates) were collected to obtain all single-cell measurements as well as the bulk MAbID reference samples. Over 1424 to 1956 single-cell samples were collected per sample type for the in vitro neural differentiation system, as well as 4862 single cell samples for the mouse bone marrow sample (all numbers reported are before quality filtering). Per cell type, the individual single-cell measurements can function as technical replicates, and a minimum of 800 cells per cell type was collected. |
| Data exclusions | Several antibodies were tested during the development of the method, to establish their functionality within the MAbID protocol. Some of the antibodies tested did not provide sufficient data quality in terms of signal specificity, which was determined upon comparison with publicly available data. This was especially relevant using primary antibody-DNA conjugates, since the quality of the antibody can decrease during the conjugation procedure. These samples were therefore excluded from further analysis and from the dataset presented in this manuscript. |
| Replication | Two biological replicates were used for each bulk MAbID experiment. The concordance between these replicates was analyzed and confirmed using Pearson's r statistical testing and UMAP vizualizations. For single-cell MAbID experiments, no biological replicates were included, but individual cell measurements per cell type can in principle be considered as technical replicates. Per cell type, the individual single-cell measurements can function as technical replicates, and a minimum of 800 cells per cell type was collected. |
| Randomization | Biological replicates were collected in seperate experiments, so these reside in different sample groups. All individual epitope-measurements (using different antibodies) per bulk MAbID experiment were collected in one experiment and combined in one library before sequencing. For single-cell MAbID experiments of the in vitro differentiation dataset, samples and cells were annotated based on the time point of harvest (either mESC or early NPC). K562 cells were taken along as a separate group. All samples were treated equally but kept in separate containers |

per cell type. For the single-cell MAbID experiments of the mouse bone marrow dataset, samples and cells were annotated based on the FACS information - cells were gated based on cell-surface markers to select for specific cell types and enrich for these. K562 cells were taken along as a separate group. K562 cells and mouse bone marrow samples were treated equally but kept in separate containers.

Blinding

Since all samples (both different replicates as well as different epitope-measurements) were treated equally during sample collection and subsequent data analysis, it was not required to include blinding of the investigators. Biological replicates were statistcially compared to confirm concordance between replicates and samples were extensively compared to publicly available data to confirm the validity of the data.

# Reporting for specific materials, systems and methods

We require information from authors about some types of materials, experimental systems and methods used in many studies. Here, indicate whether each material, system or method listed is relevant to your study. If you are not sure if a list item applies to your research, read the appropriate section before selecting a response.

## Materials & experimental systems

| n/a | Involved in the study |
|---|---|
| ☐ | ☒ Antibodies |
| ☐ | ☒ Eukaryotic cell lines |
| ☒ | ☐ Palaeontology and archaeology |
| ☐ | ☒ Animals and other organisms |
| ☒ | ☐ Clinical data |
| ☒ | ☐ Dual use research of concern |

## Methods

| n/a | Involved in the study |
|---|---|
| ☒ | ☐ ChIP-seq |
| ☐ | ☒ Flow cytometry |
| ☒ | ☐ MRI-based neuroimaging |

## Antibodies

Antibodies used

See Supplementary Table 1 for more extensive information on used antibodies. Short description on all antibodies is included below:
- Anti-Lamin B1 antibody - Nuclear Envelope Marker, Abcam, ab16048, Lot numbers: GR3398319-7, GR3369248-1;
- Histone H3K9me2 antibody (pAb), Active Motif, 39041, Lot numbers: 39239, 34718002;
- H3K9me3 Recombinant Rabbit Monoclonal Antibody (RM389), Invitrogen, MA5-33395, Lot numbers: WI3388337, WH3388337;
- Tri-methyl-histone-H3 (Lys27) Rabbit mAb, Cell Signaling Technologies, 9733S, Lot numbers: 16,19;
- Recombinant Anti-Histone H3 (tri methyl K27) antibody [EPR18607] - BSA and Azide free, Abcam, ab222481, Lot numbers: GR3256223-6, GR3256223-1;
- Anti-Trimethyl-Histone H3 (Lys36) antibody, clone RM155, RevMab, 31-1051-00, Lot numbers: T-04-02948;
- Histone H3 trimethyl K36 antibody, In-house by Hiroshi Kimura, CM333;
- H3K4me3 Monoclonal Antibody (G.532.8), Invitrogen, MA5-11199, Lot numbers: WG3341041, WH334779;
- Histone H3 (mono methyl K4) antibody, Abcam, ab8895, Lot numbers:GR3402097-1;
- Recombinant Anti-Histone H3 (acetyl K27) antibody [EP16602], Abcam, ab177178, Lot numbers: GR3202987-6, GR3202987-19;
- Recombinant Anti-RNA polymerase II CTD repeat YSPTSPS (phospho S5) antibody [3E8], Abcam, ab252852, Lot numbers: GR3302510-1, GR33352497-1;
- CTCF Antibody, Diagenode, C15410210, Lot numbers: A2354-0010;
- SUZ12 (D39F6) XP® Rabbit mAb, Cell Signalling Technologies, 3737S, Lot numbers: 8;
- Histone H3 Antibody, Novus Biologicals, NB100-747, Lot numbers: p60919;
- AffiniPure Goat Anti-Rabbit IgG (H+L), Jackson ImmunoResearch, 111-005-144, Lot numbers: 147466;
- AffiniPure Donkey Anti-Rabbit IgG (H+L), Jackson ImmunoResearch, 711-005-152, Lot numbers: 156033;
- AffiniPure Donkey Anti-Mouse IgG (H+L), Jackson ImmunoResearch, 715-005-150, Lot numbers: 155934;
- AffiniPure Donkey Anti-Rat IgG (H+L), Jackson ImmunoResearch, 712-005-150, Lot numbers: 154663;
- AffiniPure Donkey Anti-Sheep IgG (H+L) , Jackson ImmunoResearch, 713-005-147, Lot numbers: 150929;
- Alexa Fluor® 647 anti-mouse Ly-6G/Ly-6C (Gr-1) Antibody, Biolegend, 108418, Lot numbers: B287274;
- Brilliant Violet 421™ anti-mouse CD19 Antibody, Biolegend, 115549, Lot numbers: B328655;
- PE anti-mouse TER-119/Erythroid Cells Antibody, Biolegend, 116208, Lot numbers: B311713;
- APC/Cyanine7 anti-mouse CD3 Antibody, Biolegend, 100222, Lot numbers: B324939;
- Alexa Fluor® 488 anti-mouse NK-1.1 Antibody, Biolegend, 108718, Lot numbers: B316543;
- All antibody-DNA conjugates (as well as antibody-DBCO-PEG4 intermediates, both primary and secondary) were derived from the previously stated antibodies by following the conjugation procedure described in the methods section.

Validation

All antibodies (except for the Histone H3 trimethyl K36 antibody, in-house by Hiroshi Kimura, CM333) are commercially available and have been verified for specificity by the supplier as described on the specification sheets. Antibodies were further selected based on their compatibility with ChIP-seq, CUT&RUN or CUT&Tag methodology, as indicated by the supplier. The histone H3 trimethyl K36 antibody, in-house by Hiroshi Kimura, CM333, was validated by Western Blotting, Immunofluorescence staining and peptide ELISA. Antibodies were validated within the MAbID method by testing at least two concentrations and comparing the resulting data to publicly available datasets. Only antibodies with specific signal enrichment at the expected genomic locations were included and subsequently used at the concentration that resulted in the highest signal-to-noise ratio. Commercial antibodies for BM cell-surface marker stainings were optimized and validated by performing indivual stainings at different concentrations.

# Eukaryotic cell lines

Policy information about cell lines and Sex and Gender in Research

| | |
|---|---|
| Cell line source(s) | The K562 cells were a gift from the van Steensel lab at the Netherlands Cancer Institute in Amsterdam, The Netherlands. K562 cell line is generated from human female lymphoblast cells isolated from the bone marrow of a 53-year-old chronic myelogenous leukemia patient. mESCs (F1ES) are mouse embryonic stem cells resulting from a hybrid cross between Cast/EiJ (paternal) x 129SvJae (maternal) mice. These cells are female and were gifted by the Joost Gribnau lab from the Erasmus Medical Centre at Rotterdam, The Netherlands. |
| Authentication | Cell lines were not authenticated, but genomic data obtained from these cells matches the expected reference genome, including known varations in karyotype. mESC cells were grown as standard in feeder conditions to maintain pluripotency, which was monitered visually during cell culture. |
| Mycoplasma contamination | Cell lines were regulary tested for mycoplamsa contamination (every 3 months) and have always tested negative, using both PCR based and ELISA-based approaches. |
| Commonly misidentified lines (See ICLAC register) | No commonly misidentified lines were used in this study. |

# Animals and other research organisms

Policy information about studies involving animals; ARRIVE guidelines recommended for reporting animal research, and Sex and Gender in Research

| | |
|---|---|
| Laboratory animals | For mouse bone marrow isolations, 4 female mice (littermates) with the C57BL/6NCrl genotype were used, all approximately 9 weeks old. The light/dark cycles constitutes of 14 hours light (starting at 5:00) and 10 hours dark (starting at 19:00). The temperature is kept between 20 and 24 degrees Celsius and the humidity is kept between 40 and 65%. |
| Wild animals | The study did not involve wild animals. |
| Reporting on sex | All mice used were female, in order to prevent biases related to sex. |
| Field-collected samples | The study did not involve samples collected from the field. |
| Ethics oversight | All mice used in this study were bred and maintained in the Hubrecht Institute Animal Facility. Experimental procedures were approved by the Animal Experimentation Committee of the Royal Netherlands Academy of Arts and Sciences and performed according to the guidelines. |

Note that full information on the approval of the study protocol must also be provided in the manuscript.

# Flow Cytometry

## Plots

Confirm that:

☒ The axis labels state the marker and fluorochrome used (e.g. CD4-FITC).

☒ The axis scales are clearly visible. Include numbers along axes only for bottom left plot of group (a 'group' is an analysis of identical markers).

☒ All plots are contour plots with outliers or pseudocolor plots.

☒ A numerical value for number of cells or percentage (with statistics) is provided.

## Methodology

| | |
|---|---|
| Sample preparation | To isolate mouse bone marrow cells (BM), the tibia and femur bones from the hindlegs were removed. The top of the bone was removed and the marrow was flushed out using a syringe with HBSS buffer (Gibco, 14025092). Cells were isolated from the marrow by pipetting up and down several times and poured through a 70 µm cell strainer (Greiner, 542070) before diluting in 25 mL HBSS buffer. Cells were centrifuged for 10 minutes at 300 g at 4 °C. Supernatant was removed, cells were resuspended in 10 mL PBS and centrifuged at 500g for 5 min. After removing the supernatant, cells were counted and fixated (ethanol fixation) before freezing until further use - see methods for further details. Before sorting, BM cells were thawed on ice and washed twice in Wash buffer 1 (20mM HEPES pH 7.5,, 150 mM NaCl, 66.6 µg/mL Spermidine, 1X cOmplete™ protease inhibitor cocktail, 0.05% Tween20 (Sigma, P9416), 2mM EDTA) before antibody incubation. Samples were incubated with primary antibody-DNA conjugates overnight and several washing steps (with Wash Buffer 2, 20mM HEPES pH 7.5, 150 mM NaCl, 66.6 µg/mL Spermidine, 1X cOmplete™ protease inhibitor cocktail, 0.05% Tween20) were done afterwards to remove unbound antibody. Directly following primary antibody-DNA conjugate incubation, BM cells were washed once with Wash Buffer 2 and resuspended in 400 µL Wash Buffer 2 containing 5% Blocking Rat Serum (Sigma, R9759) per 1 million cells. Cells were incubated with commercial antibody-fluorophore conjugates against specific BM surface markers of Granulocytes, B cells, T cells, Erythroblasts and NK cells (see Supplementary Table 1 for antibodies and concentrations). Incubations were performed for 30 minutes at 4 °C on a tube roller. Samples were kept in the dark from this point onwards. Finally, cells were |

washed once with Wash Buffer 2 and resuspended in 1 mL Wash Buffer 2 before proceeding to FACS sorting.

| | |
|---|---|
| Instrument | A Beckman Coulter CytoFLEX SRT Benchtop Cell Sorter was used for all mouse bone marrow sorts. For other FACS sorts of K562, mESC or early NPCs cells, BD FACSJazz™ Cell Sorter and BD Influx™ Cell Sorter machines were used. |
| Software | Beckman Coulter software was used for all data acquirements.<br>BD FACS Sortware (1.2.0.142)<br>CytExpert SRT (1.1) |
| Cell population abundance | Single-cell samples were sorted and directly processed in the scMAbID protocol. Proper enrichment and purity of cell types was confirmed by assessing the quality metrics of the scMAbID dataset. |
| Gating strategy | The gating strategy is shown in Extended Data Figure 8a. Cells were first gated generally on FSC and SSC values (gates P1 to P2) to obtain high quality single-cell samples. From P2, Granulocytes were gated based on the GR1 - AF647 [660]-660-10-A value. The remaining cells were gated through P3. From P3, B cells were gated based on the CD19 - BV421 [405]-450-45-A value and NK cells were gated on the NK1 - AF488 [488]-525-40-A value. Remaining cells were gated in P4. From P4, T cells were selected gated on the CD3 - APC-Cy7 [660]-780-60-A value. Remaining cells were gated in P5. FInally, from P5 Erythroblasts were gated based on the TER119 - PE [561]-585-42-A. Boundaries in all cases were selected to obtain the cells with the highest values, ideally with some seperation from the non-gated cells. |

☒ Tick this box to confirm that a figure exemplifying the gating strategy is provided in the Supplementary Information.

