## [Peer Review file · Nature Methods]

Peer Review Information

Manuscript Title: Combinatorial single-cell profiling of major chromatin types with MAbID

Corresponding author name(s): Jop Kind

Editorial Notes:

Reviewer Comments & Decisions:

Decision Letter, initial version:
--

6th Mar 2023

Dear Professor Kind,

Your Article, "Combinatorial single-cell profiling of all major chromatin types with MAbID", has now been seen by 3 reviewers. As you will see from their comments below, although the reviewers find your work of considerable potential interest, they have raised a number of concerns. We are interested in the possibility of publishing your paper in Nature Methods, but would like to consider your response to these concerns before we reach a final decision on publication.

We therefore invite you to revise your manuscript. The required data and revisions include:

- a strong response to the issues raised by reviewers
- experimental benchmarking against the tagmentation- and pA-based methods MULTI-Tag and multi-CUT&Tag, showing strong performance of MAbID
- a clear comparison of the advantages of the new method with the recently developed NTT-seq or nano-CUT&Tag
- validations and discussions on the limitations of MAbID

[REDACTED]

We hope to receive your revised paper within 20 weeks. If you cannot send it within this time, please let us know. In this event, we will still be happy to reconsider your paper at a later date so long as nothing similar has been accepted for publication at Nature Methods or published elsewhere.

OPEN SCIENCE REQUIREMENTS

REPORTING SUMMARY AND EDITORIAL POLICY CHECKLISTS

DATA AVAILABILITY

We strongly encourage you to deposit all new data associated with the paper in a persistent repository where they can be freely and enduringly accessed. We recommend submitting the data to discipline-specific and community-recognized repositories; a list of repositories is provided here:

<http://www.nature.com/sdata/policies/repositories>

All novel DNA and RNA sequencing data, protein sequences, genetic polymorphisms, linked genotype and phenotype data, gene expression data, macromolecular structures, and proteomics data must be deposited in a publicly accessible database, and accession codes and associated hyperlinks must be provided in the "Data Availability" section.

Please include a "Data availability" subsection in the Online Methods. This section should inform readers about the availability of the data used to support the conclusions of your study, including accession codes to public repositories, references to source data that may be published alongside the paper, unique identifiers such as URLs to data repository entries, or data set DOIs, and any other statement about data availability. At a minimum, you should include the following statement: "The data that support the findings of this study are available from the corresponding author upon request", describing which data is available upon request and mentioning any restrictions on availability. If DOIs are provided, please include these in the Reference list (authors, title, publisher (repository name), identifier, year). For more guidance on how to write this section please see:

<http://www.nature.com/authors/policies/data/data-availability-statements-data-citations.pdf>

CODE AVAILABILITY

Please include a "Code Availability" subsection in the Online Methods which details how your custom code is made available. Only in rare cases (where code is not central to the main conclusions of the paper) is the statement "available upon request" allowed (and reasons should be specified).

We request that you deposit code in a DOI-minting repository such as Zenodo, Gigantum or Code

Ocean and cite the DOI in the Reference list. We also request that you use code versioning and provide a license.

For more information on our code sharing policy and requirements, please see:
<https://www.nature.com/nature-research/editorial-policies/reporting-standards#availability-of-computer-code>

MATERIALS AVAILABILITY

SUPPLEMENTARY PROTOCOL

To help facilitate reproducibility and uptake of your method, we ask you to prepare a step-by-step Supplementary Protocol for the method described in this paper. We [encourage authors to share their step-by-step experimental protocols](https://www.nature.com/nature-research/editorial-policies/reporting-standards#protocols) on a protocol sharing platform of their choice and report the protocol DOI in the reference list. Nature Portfolio 's Protocol Exchange is a free-to-use and open resource for protocols; protocols deposited in Protocol Exchange are citable and can be linked from the published article. More details can found at www.nature.com/protocolexchange/about.

ORCID

Sincerely,

Hui Hua, Ph.D.
Associate Editor
Nature Methods

Reviewers' Comments:

Reviewer #1:

Remarks to the Author:

The authors describe a novel approach to single-cell chromatin profiling using antibody-conjugated barcoded adapters inserted by ligation rather than tagmentation, followed by T7-promoter-driven cDNA synthesis and sequencing. They demonstrate that they can obtain high-quality single-cell data using this approach, and show that by conjugating either secondary antibodies using different host specificities or primary antibodies, they can simultaneously profile up to 6 different epitopes in a single experiment. As previous methods for doing the same thing using tagmentation described no more than 3 epitopes, MAbID would seem to have set a higher bar for single-cell multi-omics. However, whether the authors' approach can compete with current tagmentation-based methods already used in many laboratories is questionable for the following reasons:

1) One tagmentation-based method (Gopalan et al. Mol Cell 2021, PMID:34637755) also performs simultaneous antibody binding and another (Meers et al. Nat Biotech 2022) uses antibody conjugation, and so in principle tagmentation-based methods could be adapted for 6 or more epitopes. However, it is unclear just how much more regulatory information per cell can be obtained using abundant histone modification or other epitopes. The authors point out that the Gopalan paper showed that simultaneous binding of two epitopes in the same cell can be done by tagmentation, and this should be possible with minor modifications of MAbID. But by the same reasoning, so can a tagmentation-based approach be used for 6 or more modifications.

2) In both the Introduction and Discussion, it is stated that the major drawback of tagmentation-based methods to do what MAbID does is open chromatin tagmentation, referring to a biorxiv preprint from 2021 (reference 28) as evidence. However the results of this unrefereed preprint have been contradicted by another group (<https://www.biorxiv.org/content/10.1101/2021.08.14.456176v1>) that points out flaws in their analysis. Also, the notion that tagmentation-based methods are not suitable for constitutive heterochromatin is contradicted by evidence from many groups that have reported excellent results using CUT&Tag with H3K9me3 (e.g. PMID: 35028637 and many preprints on biorxiv including <https://doi.org/10.1101/2022.04.07.486777>, <https://doi.org/10.1101/2022.04.28.489907>, <https://doi.org/10.1101/2022.03.31.486614> and <https://doi.org/10.1101/2023.01.22.525083>). While an accessibility artifact was a problem in publications describing the ChIL-seq, CoBATCH and ACT-seq methods, the vast majority of the ~1000 publications and preprints describing antibody-directed in situ tagmentation profiling, including those cited above, follow the CUT&Tag protocol of Reference 27, which is not significantly affected by this artifact.

3) While the authors highlight a perceived limitation of the widely used methods that they are trying to replace, they do not list limitations of MAbID. One is that there are large numbers of individual steps that need to be carried out for each experiment. Even assuming that purified conjugated

adapters are available, there is 1) cell harvesting, 2) nuclei isolation, 3) fixation, 4) permeabilization, 5) antibody (primary or primary then secondary) 6) two washes, 7) FACS sorting, 8) restriction digestion 1, 9) restriction digestion 2, 10) ligation 1, 11) lysis/digestion, 12) restriction digestion 3, 13) ligation 2, 14) pooling, 15) magnetic bead clean-up, 16) in vitro transcription (a multi-step process to prepare cDNA), 17) PCR, and 18) sequencing. Figure 1a summarizes this workflow in just 6 steps, but as described in the Methods section, the authors used a Bionex Nanodrop II liquid handling robot. In contrast, single-cell tagmentation-based methods have many fewer individual steps, which is what has made ATAC-seq so popular. It seems unlikely that without a major advantage over current tagmentation-based chromatin profiling, users are unlikely to try MAbID.

4) The authors report 674-1248 high-quality cells per experiment with a median of 2281-2842 reads per cell, then on line 260 state: "These numbers are in a similar range to those reported by other recent methods that measure two or three histone PTMs simultaneously" referring to the Gopalan and Meers papers mentioned above. However, Meers reported similar read/cell values for >20,000 cells. Also, a single-cell antibody-directed tagmentation-based method called "Tip-seq" (Bartlett et al. JCB 2021 PMID: 34783858) showed that an order-of-magnitude increase in read/cell values (median ~25,000) could be obtained using in vitro transcription, which is what MAbID uses to obtain more reads/cell. This implies that the basic MAbID ligation-based method is inherently much less efficient than current tagmentation-based methods.

Reviewer #2:

Remarks to the Author:

In this manuscript, Lochs et al. developed a novel multimodal single-cell chromatin profiling method (MAbID) based on oligo-conjugated antibodies and combinatorial ligation. They first tested the technique in a series of well-designed quality control experiments in K562 cells and then validated the method for profiling a more dynamic system by applying this technology to an in vitro differentiation mouse model. The method solves some critical limitations of the conventional single-cell chromatin profiling methods (e.g., limited multiplexity, and stringent workflows) and could be broadly applied for analysis of diverse in vivo systems. I am generally satisfied with the quality of the manuscript's data and excited about this technique's potential applications. There are several comments and questions that should be addressed before the publication of this method:

1) In figure S1f, H3K27ac peaks show a positive correlation with H3K27me3 peaks- this is unexpected due to the mutually exclusive nature of these histone marks. Is it possible that open chromatin regions are preferentially digested by the restriction enzyme, resulting in increased non-specific ligation with the K27me3 antibody? How do the authors take into account possible bias introduced by the non-homogeneous digestion of the chromatin? What is the coverage of the H3K27me3 signal on open chromatin regions compared to CUT&Tag? To evaluate the specificity of the method, the authors should also compare genome-wide the coverage of H3K27me3 and H3K27ac and demonstrate that they retain mutual exclusivity in the multiplexed experiments.

2) In Figure 2D, 2E and S2E, the overall signal for S5P-PolII seems significantly higher in the multiplexed experiment compared to the single mapping experiment. How do the authors explain this difference? Is this due to miscalling of H3K36me3 signal, cross reactivity or non-specific ligation? Can the authors comment on these results?

3) Some of the datasets produced have very low fragment counts per cell, always less than 1,000 per individual marker. Is there any reason or expectation as to why these numbers are so low? They do compare to multiCUT&Tag and multi-Tag, but recently nanobody-based methods, such as NTT-seq or nanoCUT&Tag have been shown to outperform pA-based methods, how does MAbID perform compared to those protocols? How does MAbID perform compared to single-cell or bulk CUT&Tag? I believe that more extensive benchmarking against available chromatin profiling technologies is needed.

4) The authors should report the fraction of reads in peaks as well as the number of peaks called for each prep and, where possible, compare them with ChIP-seq or CUT&Tag data. I am guessing that for some, the peak calling power is very low based on the sparsity of the data. This should be noted, and as an alternative, the fraction of reads in peaks for ENCODE peaks can be reported instead.

5) In Fig. 4, the authors systematically characterized 6 chromatin markers that are dynamically altered during the differentiation of mouse embryonic stem cells to neural progenitors, which is exciting. However, the depth of the analysis is limited and does not reveal biological insight related to cell differentiation. For example, with this novel approach, can we identify any new genetic or epigenetic signatures that are missed in conventional scRNA-seq or scATAC-seq analysis? Which genes are dynamically regulated during this process? Are these genes previously reported to be essential for differentiation? This dataset deserves a deeper analysis.

6) One of the major advantages of MAbID compared to CUT&Tag-based methods is the lack of highly stringent washes used to avoid Tn5 binding on open chromatin regions. This opens the exciting possibility to profile chromatin interacting proteins beyond histone PTMs at the single-cell level. How does the protocol perform with transcription factors or chromatin remodeling complexes?

7) Single-cell chromatin multiplexed mapping, particularly with this high number of features profiled simultaneously, could potentially be applied in countless different contexts and research fields; this is very exciting but the lack of evidence showing that this method works in primary cells is a major concern. The authors should validate their method in a more complex system, such as mouse brain cells or peripheral blood samples to demonstrate they have the power and the resolution to at least identify the major cell subtypes in these tissues.

8) Due to the potentially broad application of this new technique, it would be helpful if the authors upload detailed step-by-step experimental protocol (e.g., in protocol.io).

Reviewer #3:

Remarks to the Author:

The Authors present a novel method for the multiplexed epigenetic profiling (MAbID) of low-input samples. As stated in the manuscript, further need of methodology development and refinement is still lacking in this field, as the golden standard (ChIP-Seq) requires large amounts of input material, and one of its most promising alternatives (CUT&Tag) has been shown to have a clear bias to map open chromatin regions.

The methodological design is based on the conjugation of oligonucleotide adapters to antibodies, and

their ligation to compatible restriction enzyme cutting sites. Thus, the genomic regions where the target epitope is bound to can be labeled for its genomic profiling by sequencing.

The authors first show a proof-of-principle MAbID experiment in K562 cells with a single epitope profiling, demonstrating its reproducibility for several histone marks and two chromatin-binding proteins. Notably, MAbID detects the genomic profiles of histone marks and proteins at sites of heterochromatin with good correlation with ChIP-Seq data, clearly overcoming the before-mentioned bias of the Tn5-based techniques. Next, the authors test the capacity of multiplexing the assay, whether by combining primary antibodies of different sources with adapter-conjugated secondary antibodies; or by combining multiple adapter-conjugated primary antibodies. This second strategy allows for easy multiplexing of the assay with up to 6 different antibodies. Remarkably, one of the main strengths of the proposed method is the unaffected performance, in terms of detection and resolution, when increasing the number of epitopes to be analyzed simultaneously.

The authors also demonstrate the utility of scMAbID for the multiplex epigenetic profiling in a biologically relevant model, such as differentiation of mouse ESCs to neural progenitors, being able to discriminate between different populations on the base of the single-cell signature generated by the integration of multiple epigenetic marks; and allowing the single cell monitoring of the epigenetic inactivation of the X chromosome along this differentiation process.

In summary, MAbID and scMAbID represent a clear progress and a new promising tool for the epigenetic profiling of multiple chromatin-binding proteins, with fair detection and reproducibility in both hetero and euchromatic regions, low needs of input material, and its capacity of being multiplexed without losing performance power.

Nevertheless, the manuscript in its current form would benefit from improved clarity, before it is ready for publication. Below I list a few points to clarify:

1. MAbID is based on the ligation of adapters into restriction enzyme target sites through specific DNA sequences. Indeed, the authors clearly show that the resulting reads mainly map (>95%) to the target DNA motifs (TTAA and GATC for each of the used systems). While this is a solid validation of the methodological design, this sequence-specificity may interfere with detection of protein binding sites. As the authors explain, the simplicity of these DNA binding motifs makes them highly abundant in the human genome (18 million TTAA and 7 million GATC motifs) but it also poses a clear bias, since one would not be able to detect protein binding sites in genomic regions without these concrete motifs nearby. Could this explain the fact that the MAbID peaks are broader in comparison to ChIP-Seq? One of the proposed advantages of MAbID is the fact that it overcomes the preference of Tn5-based methods towards euchromatin. For consistency, the Authors should remark and discuss this possible limitation of their own technology and its effects on the results, discuss further analyses to assess and measure its impact, and propose a possible mitigation strategy.

2. The authors state that MAbID allows the "profiling of an increasingly complex set of histone PTMs and chromatin-binding proteins." However, the only non-histone chromatin-binding proteins tested and presented in the manuscript are Lamin B1 and RNA polymerase II. Increasing the examples of non-histone protein profiling would increase the impact of the manuscript. Have the Authors tested the method with some other chromatin-binding proteins characterized by a more labile interaction with DNA, which often pose problems in ChIP-Seq? It would be interesting to see profiles of some chromatin modifiers and remodelers, like members of SWI/SNF, Polycomb or Thrtorax complexes. This would add a major strength of the proposed methodology, in comparison with the golden standard in the field.

3. The Authors suggest in line 361 of the manuscript that they detected a higher MAbID background signal in regions of open chromatin. Although they clarify and show that it can be corrected with a negative control, like a ChIP-Seq input, they do not show data regarding this bias towards regions of euchromatin, which they attribute to "locally increased efficiency of either the digestion or ligation

steps". If they have a specific analysis supporting this statement and measurement of the degree of the impact of this bias, they should include it in the manuscript.

4. The Authors should compare the results of MAbID with multiplex CUT&Tag data. Apparently, one of the main advantages of their technology is the proper detection of binding sites in heterochromatin sites, as opposed to Tn5-methods. Comparing and measuring this difference will improve the robustness of their results and their claims.

5. At some instances the Authors use thought short-cuts assuming that things are obvious to everyone but I would encourage the authors to provide more explanations for why they make certain choices in their method and approach. I list here such instances:

a. "We initially employed secondary antibody-DNA conjugates to more accurately compare the quality of multiple genomic profiles in parallel."

b. "We therefore excluded H3 from subsequent analysis and only added it to function as a crowding reagent during following experiments." I do not understand why it would be needed to add it as a crowding reagent and how it would work as such.

c. "The conjugation procedure was slightly modified to account for differences in buffer compositions". Which buffer do the Authors refer to?

6. The correlations the Authors report, ranging from 0.24 to 0.47 seem low. Could the Authors comment on this?

7. Can the Authors add information in the main text whether the ChIP-seq data they used were publicly-available or whether they were generated for the purpose of this study.

8. It would be useful if the Authors provided some considerations on the number of cells they use in their bulk assay (250,000). Could the Authors comment on the upper limit?

9. Why do the Authors work with isolated nuclei as opposed to entire cells. Can the Authors comment on this? Can one work with entire cells as well?

10. Is there any risk coming from the way the conjugation of the adapters to the antibodies is done whereby multiple adapters could get conjugated to the same antibody? Could the Authors comment on that?

11. Can the Authors comment in the main text on how the tailoring of the enzymes for a given epitope quantitatively affects the resolution of the assay? How strong is this effect?

12. Related to this, the Authors comments on the fact that with this approach (tailoring by selecting the proper restriction site) they expect an increase in the resolution and yet they conclude this paragraph by saying "Both the resolution and distribution of MAbID signal were surprisingly independent of the choice of antibody-adapter and the complexity of the data was similar between all approaches (Extended Data Fig. 3c-d)." What am I missing?

13. I wonder whether the assay could be simplified by having the adapter conjugated to the Antibodies already with a sticky end and phosphorylated so that no additional step of enzyme digestion would be needed. In general it would be great to add more explanation for why the method requires two different enzymes (I assume it has to do with the need for only one of the two sites being

in a phosphorylated state but it would be good to have this reasoning explicitly stated in the main text).

Author Rebuttal to Initial comments

Response to reviewers

We thank the reviewers for their constructive feedback and we highly appreciate the time and effort they invested in the careful reading of our manuscript.

On the next pages, please find our point-by-point response to the reviewer comments in blue, references to manuscript figures and line numbers in red and manuscript citations in *italic*. In the manuscript text, we highlighted all modified sections and sentences.

We added additional Extended Data Figures based on the reviewer comments. Please find below the updated panel numbers for clarity:

- Original Extended Data Figure 1 → New Extended Data Figure 1 & 2:
 - 1c to 1f
 - 1d to 1c
 - 1e to 2a
 - 1f to 2b
 - 1g to 2c
 - 1h to 2e

- Original Extended Data Figure 2 → New Extended Data Figure 3
 - Panel numbers same as original

- Original Extended Data Figure 3 → New Extended Data Figure 4 & 5
 - 3b to 4c
 - 3c to 4d
 - 3d to 4f
 - 3e to 5a
 - 3f to 5b
 - 3g to 5c
 - 3h to 5d
 - 3i to 5e

- Original Extended Data Figure 4 → New Extended Data Figure 6 & 7
 - 4d to 6f
 - 4e to 7a
 - 4f to 7b
 - 4g to 7c
 - 4h to 7d
 - 4i to 7e
 - 4j to 7f
 - 4k to 7g
 - 4l to 7h

- Figure 5 and Extended Data Figure 8 are newly included

Reviewers' Comments

Reviewer #1:

The authors describe a novel approach to single-cell chromatin profiling using antibody-conjugated barcoded adapters inserted by ligation rather than tagmentation, followed by T7-promoter-driven cDNA synthesis and sequencing. They demonstrate that they can obtain high-quality single-cell data using this approach, and show that by conjugating either secondary antibodies using different host specificities or primary antibodies, they can simultaneously profile up to 6 different epitopes in a single experiment. As previous methods for doing the same thing using tagmentation described no more than 3 epitopes, MAbID would seem to have set a higher bar for single-cell multi-omics. However, whether the authors' approach can compete with current tagmentation-based methods already used in many laboratories is questionable for the following reasons:

We would like to thank the reviewer for taking the time to read our manuscript and provide very valuable feedback. Our intention with MAbID was to develop and present an approach that yields highly multiplexed (single-cell) genomic profiles. We are equally excited about the powerful developments of the Tn5-based multifactorial methods and hope that by adding MAbID to the existing toolbox, this will contribute to new synergistic developments in the single-cell multi-omics field. Based on the reviewer comments, we realized that we may have surpassed our intentions by contrasting MAbID too strongly to other existing methods. To leverage this, we have now made several modifications to the text (see below) and also more clearly highlighted the limitations of MAbID.

Comment #1.1

One tagmentation-based method (Gopalan et al. Mol Cell 2021, PMID:34637755) also performs simultaneous antibody binding and another (Meers et al. Nat Biotech 2022) uses antibody conjugation, and so in principle tagmentation-based methods could be adapted for 6 or more epitopes. However, it is unclear just how much more regulatory information per cell can be obtained using abundant histone modification or other epitopes. The authors point out that the Gopalan paper showed that simultaneous binding of two epitopes in the same cell can be done by tagmentation, and this should be possible with minor modifications of MAbID. But by the same reasoning, so can a tagmentation-based approach be used for 6 or more modifications.

We fully agree with the reviewer on this point. Although MAbID is the first method to simultaneously measure six epitopes in the same cell, we acknowledge that there is no theoretical limitation for the tagmentation-based methods to increase the number of measurements from the current three epitopes to six or more. We made this more explicit in the discussion section:

Lines ~492-494: *"To our knowledge, MAbID is the first method to profile a combination of more than three epitopes, even though there is no theoretical or technical limitation towards combining more measurements for ~~similar~~ Tn5-based multifactorial approaches."*

We would like to note that our analysis implies that one can obtain more information per cell by combining additional epitope measurements. We employed the Information Gain metric (IG) to assess the contribution of each modality to the separation of the pluripotent and differentiated clusters in our neural differentiation system (Extended Data Fig. 7e-f). The IG improves with the inclusion of additional modalities (Extended Data Fig. 7e). We do observe that the increase in IG starts to plateau, which indicates that with we start approaching the maximum information content with this number of measurements. Additionally, we find that especially the enhancer and Polycomb-associated epitopes contribute to the increase in IG (Extended Data Fig. 7f). This is expected since these epitopes are frequently employed to predict cell type and lineage.

We agree that measuring combined localizations or binding-events on a single DNA-molecule - as in Gopalan et al.¹ - can provide insights into the interplay between epitopes and their combined regulatory function. We therefore included this as a potential improvement or outlook in the discussion section. We realize that we perhaps wrongfully suggested that adding this measurement to the technology is an easy feat. We modified this sentence in the discussion to clarify this further:

Lines ~515-517: *"Such a strategy is currently not integrated in the MAbID procedure, but could be accommodated within the method ~~with minor adaptations~~ by including a PCR-based amplification along with some additional adaptations."*

Comment #1.2

In both the Introduction and Discussion, it is stated that the major drawback of tagmentation-based methods to do what MAbID does is open chromatin tagmentation, referring to a biorxiv preprint from 2021 (reference 28) as evidence. However the results of this unrefereed preprint have been contradicted by another group (<https://www.biorxiv.org/content/10.1101/2021.08.14.456176v1>) that points out flaws in their analysis. Also, the notion that tagmentation-based methods are not suitable for constitutive heterochromatin is contradicted by evidence from many groups that have reported excellent results using CUT&Tag with H3K9me3 (e.g. PMID: 35028637 and many preprints on biorxiv including <https://doi.org/10.1101/2022.04.07.486777>, <https://doi.org/10.1101/2022.04.28.489907>, <https://doi.org/10.1101/2022.03.31.486614> and <https://doi.org/10.1101/2023.01.22.525083>). While an accessibility artifact was a problem in publications describing the ChIL-seq, CoBATCH and ACT-seq methods, the vast majority of the ~1000 publications and preprints describing antibody-directed in situ tagmentation profiling, including those cited above, follow the CUT&Tag protocol of Reference 27, which is not significantly affected by this artifact.

We appreciate the reviewer pointing this out to us. Accordingly, we have removed this reference² (#28 in the original manuscript) from the main text. We have now modified the corresponding text in the introduction and discussion to clarify this.

Introduction, lines ~55-63: *"The recent multifactorial methodologies almost exclusively rely on antibody detection followed by Tn5-mediated tagmentation, which is commonly used in state-of-the-art genomic profiling techniques²⁷. The advantage of this approach is that Tn5 is very efficient and yields specific data at a high resolution. However, ~~its propensity towards integrating into open chromatin regions^{28,29} may introduce accessibility biases and limits measuring modalities that reside in constitutive heterochromatin.~~ the currently available multifactorial methods have thus far only been implemented to profile up to three epitopes per cell, all of which were either residing in active chromatin regions or in facultative heterochromatin²¹⁻²⁶. Thus, it remains unresolved whether combined profiles of an increasingly complex set of epitopes can be obtained from single cells, especially including those enriched in constitutive and inaccessible heterochromatin types."*

AND

Discussion, lines ~463-471: *"Several other methods have recently been developed to generate combined measurements of multiple histone PTMs in single cells, by employing the Tn5 transposase to integrate barcodes into the genome at sites of antibody binding^{21,23-26}. ~~Even though~~ Tn5 is highly efficient and has been shown to generate high quality single-cell profiles across different chromatin types^{16,54}. However, it remains unclear how its intrinsic affinity for open chromatin⁵⁵ will affect combined measurements, especially when including epitopes enriched in constitutive heterochromatin. ~~challenging to profile histone PTMs enriched in constitutive heterochromatin due to affinity of Tn5 for open chromatin regions²⁸. Instead of employing Tn5, MAbID uses restriction-digestion and ligation steps to effectively integrate barcodes into the genome. We successfully performed multifactorial profiling of two ~~profiled several~~ epitopes located at inaccessible chromatin, Lamin B1 and H3K9me3, in combination with epitopes residing in active chromatin."~~*

Comment #1.3

While the authors highlight a perceived limitation of the widely used methods that they are trying to replace, they do not list limitations of MAbID. One is that there are large numbers of individual steps that need to be carried out for each experiment. Even assuming that purified conjugated adapters are available, there is 1) cell harvesting, 2) nuclei isolation, 3) fixation, 4) permeabilization, 5) antibody (primary or primary then secondary) 6) two washes, 7) FACS sorting, 8) restriction digestion 1, 9) restriction digestion 2, 10) ligation 1, 11) lysis/digestion, 12) restriction digestion 3, 13) ligation 2, 14) pooling, 15) magnetic bead clean-up, 16) in vitro transcription (a multi-step process to prepare cDNA), 17) PCR, and 18) sequencing. Figure 1a summarizes this workflow in just 6 steps, but as described in the Methods section, the authors used a BioNex Nanodrop II liquid handling robot. In contrast, single-cell

tagmentation-based methods have many fewer individual steps, which is what has made ATAC-seq so popular. It seems unlikely that without a major advantage over current tagmentation-based chromatin profiling, users are unlikely to try MAbID.

Based on the comments by the reviewer, we realize and acknowledge that we should have been more careful in positioning MAbID as an orthogonal method to generate multiplexed genomic profiles, in addition to the existing technologies. To make this point more explicit, we have modified the final sentence of the discussion:

Lines ~525-528: “We anticipate that ~~innovations such as MAbID and other methods~~ MAbID, as an orthogonal method to the existing tagmentation-based approaches, will enable researches will contribute to the advancement of the single-cell multi-omics field to study the combined epigenetic landscapes of complex biological systems in integrated experiments.”

The MAbID protocol indeed contains several experimental steps, which we explained in detail in the results and method sections. However, almost all of these steps are very short and ‘standard’ for most equivalent technologies. Any skilled molecular biologist can successfully perform this protocol. To facilitate the use of the method, we will also provide a publicly accessible step-by-step protocol of both the experimental and computational methods (please also see **comment #2.8** in response to reviewer 2).

The reviewer raises a valid point concerning the liquid-handling robots, which are not readily available in every laboratory or require a substantial infrastructural investment. We make use of robotics to i) increase throughput, ii) reduce sampling handling and iii) reduce overall processing time. However, we anticipate that increasing the reaction volumes to bypass the use of robotics will not influence the performance of scMAbID. We made a specific note on this in the methods section (**lines ~1294-1297**). More importantly, the implementation of a combinatorial-indexing strategy (as mentioned in the discussion, **lines ~507-509**) will circumvent the use of liquid-handling robots altogether and increase throughput.

We now also mention several current limitations of MAbID in the discussion and list multiple suggestions for future improvements. Steps to improve are i) the reduction of background signal related to chromatin accessibility (**lines 472-479**), ii) increasing resolution by including more restriction-digestion or sequence-independent enzymes per reaction (**lines 480-487**), iii) increasing throughput by adopting combinatorial-indexing strategies (**lines ~508-509**) and iv) increasing the low number of unique counts by improving ligation efficiencies (**lines ~517-520**).

Comment #1.4

The authors report 674-1248 high-quality cells per experiment with a median of 2281-2842 reads per cell, then on line 260 state: "These numbers are in a similar range to those reported by other recent methods that measure two or three histone PTMs simultaneously" referring to the Gopalan and Meers papers mentioned above. However, Meers reported similar read/cell values for >20,000 cells. Also, a single-cell antibody-directed tagmentation-based method called "Tip-seq" (Bartlett et al. JCB 2021 PMID: 34783858) showed that an order-of-magnitude increase in read/cell values (median ~25,000) could be obtained using *in vitro* transcription, which is what MAbID uses to obtain more reads/cell. This implies that the basic MAbID ligation-based method is inherently much less efficient than current tagmentation-based methods.

In the sentence on line 260 of the first submission, we refer to the unique read counts being in a similar range compared to other methods. We acknowledge that the benchmarking to other methods was insufficient. Therefore, we have updated and additionally included panels on read counts per cell and FRiP score comparisons in the manuscript (Extended Data Fig. 6c and 6e). Please see comments #2.3 and #2.4 for more details on these specific quality metrics. We also included information on additional features of the multifactorial approaches, which is provided in Extended Data Fig. 6d.

As the reviewer correctly points out, the reported cell numbers for scMAbID are considerably lower compared to most other multifactorial methods (Extended Data Fig. 6d). Since MAbID is a new method, we first focused our efforts on optimizing the approach. A next step in the development of the technology will focus on increasing throughput by implementing a combinatorial-indexing strategy. Nevertheless, the current plate-based approach can be very valuable for specific applications and experimental settings. These include testing multiple conditions in parallel or the selection of specific – rare - cell populations based on FACS parameters. The newly added Figure 5 illustrates such an approach to select for five populations of blood cells isolated from mouse bone marrow (see comment #2.7). We have included a section in the text to illustrate this further:

Lines 299-303: "While the other approaches depend on droplet-based cell barcoding, scMAbID is the only plate-based protocol, resulting in a lower throughput (Extended Data Fig. 6d). However, the recovery of cells after sequencing is equal to the other multifactorial methods and the combination of FACS sorting with plate-based sample processing provides the opportunity to select for specific cells from whole populations, thereby circumventing the need to sequence all cells (Extended Data Fig. 6d)."

Based on our personal experience with linear amplification by *in vitro* transcription, we do not expect an increased yield in read counts. For example, scDamID³ (PCR amplification) has similar unique read counts compared to scDam&T-seq⁴ (linear amplification) in the same clonal cell line. However, linear amplification has clear advantages over PCR - we observe a more even distribution in reads and fewer "jackpot" amplification events. The main improvement to increase unique read counts for scMAbID will

likely involve increasing restriction-digestion and ligation efficiencies. Additionally, we could include a selective enrichment step for reads containing the antibody-adapter via biotin-streptavidin pulldowns. This will increase the yield of reads with the correct read structures. We have included these options in the discussion:

Lines ~517-520: *“Increasing the efficiency in recovering ligation events can be an additional optimization to the single-cell protocol, potentially by including a selective enrichment for reads containing the antibody-adapter via biotin-streptavidin pulldowns.”*

Reviewer #2:

In this manuscript, Lochs et al. developed a novel multimodal single-cell chromatin profiling method (MABID) based on oligo-conjugated antibodies and combinatorial ligation. They first tested the technique in a series of well-designed quality control experiments in K562 cells and then validated the method for profiling a more dynamic system by applying this technology to an in vitro differentiation mouse model. The method solves some critical limitations of the conventional single-cell chromatin profiling methods (e.g., limited multiplexity, and stringent workflows) and could be broadly applied for analysis of diverse in vivo systems. I am generally satisfied with the quality of the manuscript's data and excited about this technique's potential applications. There are several comments and questions that should be addressed before the publication of this method:

We are very happy to hear that the reviewer is enthusiastic about the method and is excited about potential applications of MABID. We thank the reviewer for the constructive feedback and the excellent recommendations to improve the manuscript and the approach further.

Comment #2.1

In figure S1f, H3K27ac peaks show a positive correlation with H3K27me3 peaks- this is unexpected due to the mutually exclusive nature of these histone marks. Is it possible that open chromatin regions are preferentially digested by the restriction enzyme, resulting in increased non-specific ligation with the K27me3 antibody? How do the authors take into account possible bias introduced by the non-homogeneous digestion of the chromatin? What is the coverage of the H3K27me3 signal on open chromatin regions compared to CUT&Tag? To evaluate the specificity of the method, the authors should also compare genome-wide the coverage of H3K27me3 and H3K27ac and demonstrate that they retain mutual exclusivity in the multiplexed experiments.

We thank the reviewer for pointing this out. We first looked into the origin of this slight positive correlation between H3K27me3 and H3K27ac (old Extended Data Fig. S1f → new Extended data Fig. 2b). Extended Data Fig. 2b represents a genome-wide correlation heatmap without additional filtering on low-variable regions. We found that these regions contributed most to the positive correlation between H3K27me3 and H3K27ac. We corrected for this by omitting these regions from the analysis (along with commonly black-listed regions⁵ that had already been omitted) and recalculated the correlation heatmaps. In addition, to provide more extensive benchmarking of MABID, we included CUT&Tag datasets as a reference (Extended Data Fig. 2b).

Based on the reviewer comment, we realized that we should have provided more insight into the normalization procedure of MABID. In order to normalize MABID data, we take along a control sample in which the primary antibody incubation is omitted. This control has a specific signal distribution, which

generally resembles chromatin accessibility. We also observe a background chromatin accessibility component in the MAbID samples prior to normalization. To illustrate this, we showed the distributions of raw, control and normalized MAbID data over TSS sites and LADs (old Extended Data Fig. 1c → new Extended Data Fig. 1f).

To provide more insight into this aspect of MAbID, we have included two extra panels. These depict genomic tracks of the raw, control and normalized MAbID data for H3K27me3, in comparison to genome-wide TTA site coverage, ATAC-seq, H3K27me3 CUT&Tag and ChIP-seq data (Extended Data Fig. 1d-i). We chose to show ATAC-seq here (instead of H3K27ac), because it clearly illustrates the accessibility patterns in the MAbID control data. These panels emphasize the excellent concordance between H3K27me3 MAbID, ChIP-seq and CUT&Tag data after normalization. Without normalization, the H3K27me3 profiles show a modest genome-wide positive correlation with ATAC-seq; a signal that is removed from the data upon normalization (Extended data Fig. 1d-ii).

To illustrate this further, we also performed domain calling on raw and normalized H3K27me3 MAbID data and overlaid these domains with those called on ENCODE ChIP-seq data (Extended Data Fig. 1e). This analysis illustrates that upon normalization of the H3K27me3 MAbID data, the overlap in domains between the methods increases substantially. This shows that the normalization strategy effectively removes the background component for MAbID data, both for H3K27me3 as for the other chromatin types (Extended Data Fig. 1f). To further clarify this in the manuscript, we updated the main text:

Lines ~112-120: “These experiments were performed in biological replicates using one primary antibody against an epitope of interest per sample, which was targeted by the secondary antibody-DNA conjugate in a subsequent incubation. In parallel, a control sample was generated in which the primary antibody was omitted during the first incubation step, to serve as an input (mock IP) dataset for normalization. The control signal is largely unbiased by TTA sequence motif coverage and instead mirrors chromatin accessibility, when compared to publicly available ATAC-seq data (Extended Data Fig. 1d). Normalization reduces the correlation with ATAC-seq data and strongly increases the overlap between MAbID and ChIP-seq domains, confirming that this approach effectively removes the background component from MAbID data (Extended Data Fig. 1d-f).”

The cross-reactivity between samples in multiplexed MAbID experiments will be discussed in **comment #2.2**.

Comment #2.2

In Figure 2D, 2E and S2E, the overall signal for S5P-PolIII seems significantly higher in the multiplexed experiment compared to the single mapping experiment. How do the authors explain this difference? Is

this due to miscalling of H3K36me3 signal, cross reactivity or non-specific ligation? Can the authors comment on these results?

This is an important observation that we wished to address experimentally. The signal for the RNA Pol II CTD Ser5P measurements is indeed considerably higher in all the combined measurements compared to the individual measurements. We reason that this could be due to an improvement of signal-to-noise ratios, related to cumulative “blocking” of accessible chromatin sites by all four antibody combinations. This may be most apparent for RNA Pol II CTD Ser5P due to the slightly poorer affinity/specificity of this antibody compared to H3K36me3 and H3K27me3, or related to properties of the RNA Polymerase complex that are different from histone PTMs (i.e. residence time).

The observation that MAbID signal increases in quality upon multiplexing is favorable for the technology. However, to ensure that the increased signal is not related to cross-reactivity, we performed multiplexed MAbID experiments with different combinations of antibodies. In each combination, we omitted one of the primary antibodies along with its corresponding species-specific secondary antibody-DNA conjugate from the incubation. These results confirm that there is no difference in RNA Pol II CTD Ser5P signal distributions for combinations including or excluding H3K36me3. Furthermore, Extended Data Fig. 3f also shows that the RNA Pol II CTD Ser5P signal over genes is higher at the TSS for both the individual and the combined measurements, which is not the case for the H3K36me3 signal. In summary, these results verify that the increased RNA Pol II CTD Ser5P signal in combined measurements can't be attributed to cross-reactivity or miscalling of H3K36me3 signal.

Lines ~187-193: “The unaltered correlation coefficients of the other epitopes suggests that the other secondary antibody-DNA conjugates are specific for their respective target species (Extended Data Fig. 3c). To fully rule out the possibility of cross-reactivity between the other secondary antibody-DNA conjugates, we performed MAbID experiments comparing the combination of all antibodies with those in which one of the respective three was excluded. The signal enrichments are highly similar between the different combinations, verifying that the signal is indeed specific to the corresponding IgG (Extended Data Fig. 3e).”

Comment #2.3

Some of the datasets produced have very low fragment counts per cell, always less than 1,000 per individual marker. Is there any reason or expectation as to why these numbers are so low? They do compare to Multi-CUT&Tag and Multi-Tag, but recently nanobody-based methods, such as NTT-seq or nano-CUT&Tag have been shown to outperform pA-based methods, how does MAbID perform compared to those protocols? How does MAbID perform compared to single-cell or bulk CUT&Tag? I believe that more extensive benchmarking against available chromatin profiling technologies is needed.

We thank the reviewer for raising this point and we acknowledge that the manuscript can benefit from more extensive benchmarking to the available bulk (in particular CUT&Tag) and single-cell multifactorial methods. We have therefore added the following points to the manuscript:

- **Extended Data Fig. 2b** – We extended our analysis of the correlation between MAbID and CHIP-seq by including publicly available bulk CUT&Tag datasets. Correlations between MAbID and CHIP-seq are highly comparable to those between MAbID and CUT&Tag. We updated **lines ~124-126** in the main text accordingly.
- **Extended Data Fig. 2d** – We calculated FRiP (Fraction of Reads in Peaks) scores over ChromHMM domains and LAD regions for MAbID as well as public CHIP-seq and CUT&Tag datasets. FRiP scores are calculated based on raw reads, while for MAbID data we generally perform a normalization step during the overall data processing. To make a more comprehensive comparison between MAbID and the reference datasets, we also implemented an analogous Signal Enrichment in Peaks (SEiP) score that can incorporate positive and negative non-integer values. We explain the calculation of this SEiP score, which is based on random permutations of genomic windows, in the methods section, **lines ~1436-1452**, and added the following section to the main text:

Lines ~130-135: *“To further explore the on-target specificity of MAbID, we calculated the enrichment of MAbID signal over relevant genomic regions from publicly available datasets, such as CHIP-seq peaks, ChromHMM domains or LAD regions. For raw MAbID data, the FRiP (Fraction of Reads in Peaks) scores are lower with respect to other approaches, as a result of the background component in the raw dataset (Extended Data Fig. 2d). However, when comparing the signal enrichment of normalized data (Signal Enrichment in Peaks, SEiP score), MAbID performance is in a similar range (Extended Data Fig. 2d).”*

- **Extended Data Fig. 6c** – When we assembled the original manuscript, only Multi-CUT&Tag¹ and Multi-Tag⁶ were published, so we had only included these as reference datasets for our original calculations of read/fragments counts per cell and epitope. We now downloaded public data for all currently available multifactorial approaches, including nano-CUT&Tag⁷ and NTT-seq⁸, and recalculated these metrics for all methods. scMAbID read counts per epitope are within a similar range compared to most of the other multifactorial approaches. Reads per epitope are just slightly lower when compared to Multi-CUT&Tag and Multi-Tag, which is at least partly caused by the higher number of epitope measurements per cell for scMAbID (**Extended Data Fig. 6c-d**). The notable exception is nano-CUT&Tag, which clearly outperforms all multifactorial methods in both total read numbers per cell as well as per epitope, which we highlighted in the main text (see below). However, a clear disadvantage of nano-CUT&Tag is the limited potential to increase

epitope measurements, since it is (at least in the current form) dependent on using primary antibodies from different species, to match the species-specific nanobody-Tn5 fusions.

Lines ~293-298: *“The median number of unique counts per cell after filtering was 2715 for K562, 2281 for mESC and 2842 for early NPCs, with per epitope a median of unique counts ranging from 119 to 706 in each cell (Fig. 4b). These numbers are in a similar range to those reported by other recent methods that measure two or three histone PTMs simultaneously^{21,23,24,26} (Extended Data Fig. 6c-d). nano-CUT&Tag²³ is the only notable exception, as this method significantly outperforms all others in terms of read counts (Extended Data Fig. 6c-d).”*

- **Extended Data Fig. 6d** – Besides the read counts in **Extended Data Fig. 6c**, we also included a comparison of i) the number of epitopes measured, ii) the input numbers of cells for both the antibody staining as well as sequencing and iii) the recovery of cells passing quality thresholds after sequencing. Please refer to **comment #1.4** for our considerations on the reported number of cells.
- **Extended Data Fig. 6e** – We calculated FRiP scores for the scMAbID data (also see **comment #2.4**). We only included Multi-Tag⁶ and NTT-seq⁸ as reference datasets, since these were based on human K562 cells and we could thus perform identical calculations for all of the methods. We updated the main text as follows:

Lines 312-323: *“To further assess the specificity of the data at single-cell resolution, we calculated FRiP scores for each epitope in single cells using ChromHMM domains as a reference. High FRiP scores are observed for epitope measurements at the corresponding domain, while these are considerably lower at unrelated chromatin types (Fig. 4e). We directly compared scMAbID FRiP scores to those calculated on publicly available Multi-Tag⁶ and NTT-seq²⁴ datasets with multifactorial measurements of 3 epitopes in K562 cells (Extended Data Fig. 6e). NTT-seq moderately outperforms scMAbID, especially on the active chromatin types, but overall FRiP scores are on a comparable scale (Extended Data Fig. 6e). scMAbID FRiP scores for H3K9me3 and Lamin B1 in LAD regions are markedly higher than expected for a random distribution (Extended Data Fig. 6e). However, since these epitopes were not measured with any of the other approaches, direct comparisons in constitutive heterochromatin were unattainable. Collectively, these results corroborate that scMAbID generates specific measurements of a combination of six epitopes in single cells.”*

Comment #2.4

The authors should report the fraction of reads in peaks as well as the number of peaks called for each prep and, where possible, compare them with CHIP-seq or CUT&Tag data. I am guessing that for some,

the peak calling power is very low based on the sparsity of the data. This should be noted, and as an alternative, the fraction of reads in peaks for ENCODE peaks can be reported instead.

These are good suggestions. We have included FRiP scores for the MAbID data and compared these to public CHIP-seq and CUT&Tag datasets (Extended Data Fig. 2d, see comment #2.3). Furthermore, we called domains on the H3K27me3 MAbID data and compared these to CHIP-seq data (see comment #2.1). We observe a good overlap between the domains called on normalized MAbID data and those called on CHIP-seq data (Extended Data Fig. 1e).

For scMAbID data, the sparsity of the data is indeed preventing us from calling reliable domains or peaks, as the reviewer rightfully notes. We have therefore calculated FRiP scores for both MAbID and other multifactorial methods (Multi-Tag⁶ and NTT-seq⁸) using K562 data (Extended Data Fig. 6e, see comment #2.3). NTT-seq moderately outperforms scMAbID, especially on the active chromatin types, but we note that scMAbID FRiP scores are overall on a comparable scale and are equivalent to Multi-Tag scores (Extended Data Fig. 6e). Direct comparisons of the FRiP scores in constitutive heterochromatin were unattainable since the other approaches have not included these chromatin types, but we observe a higher than randomly expected score for Lamin B1 and H3K9me3 in scMAbID data.

Comment #2.5

In Fig. 4, the authors systematically characterized 6 chromatin markers that are dynamically altered during the differentiation of mouse embryonic stem cells to neural progenitors, which is exciting. However, the depth of the analysis is limited and does not reveal biological insight related to cell differentiation. For example, with this novel approach, can we identify any new genetic or epigenetic signatures that are missed in conventional scRNA-seq or scATAC-seq analysis? Which genes are dynamically regulated during this process? Are these genes previously reported to be essential for differentiation? This dataset deserves a deeper analysis.

We had focused on this particular mESC to early NPC (day 5) trajectory with the intention to capture the process of X-chromosome inactivation and to explore the potential to leverage the hybrid-genome of this line to determine single-cell Xi-status based on allelic H3K27me3 enrichment (Fig. 4i-j). The allelic resolution of the data enabled us to directly investigate the associated changes of other epigenetic marks on the inactivated X-chromosome (Fig. 4j and Extended Data Fig. 7h). Additionally, we observe that enhancer- and Polycomb-associated epitopes are main predictors in identifying cell types, which highlights that these potentially drive the changes in cellular state during differentiation (Extended Data Fig. 7e-f). For gene-centric analyses, this differentiation trajectory is somewhat less suitable because we harvested the cells after only five days into the differentiation towards NPCs (takes 17 days in our protocol) and as a result the cell states are still relatively fluid and quite mixed. At day 5, we observe that

~30% of early NPC cells is still in a pluripotent state, which complicates performing meaningful gene-centric analyses (Fig. 4g-h).

For this part of the analysis, we decided to focus on the new data of joint measurements of six chromatin types in primary mouse bone marrow cells in Figure 5, which contains more discrete cell types and well-annotated marker-gene expression. Please see comment #2.7 for our discussion of these results.

Comment #2.6

One of the major advantages of MAbID compared to CUT&Tag-based methods is the lack of highly stringent washes used to avoid Tn5 binding on open chromatin regions. This opens the exciting possibility to profile chromatin interacting proteins beyond histone PTMs at the single-cell level. How does the protocol perform with transcription factors or chromatin remodeling complexes?

It would indeed be very exciting if MAbID is broadly applicable to many chromatin-binding proteins, especially those that more transiently associate with the DNA. We had so far only tested two chromatin-binding proteins, namely Lamin B1 and RNA Polymerase II, which were both included in the original manuscript. To further assess whether MAbID also works for transcription factors and other types of chromatin-binding proteins, we tested a panel of four other antibodies that were i) available to us, ii) of which we knew the expected binding pattern and iii) of which there was publicly available reference data – CTCF (transcription factor), SUZ12 (subunit of Polycomb Repressive Complex 2, PRC2), Rad21 (Cohesion subunit) and SETDB1 (Histone methyltransferase of H3K9me3). All of these epitopes show signal enrichments over the corresponding reference domain (see panel below and Extended Data Fig. 2f).

Legend – MAbID signal enrichment of CTCF over CHIP-seq peaks (ENCODE, ± 100 kb), SUZ12 over Polycomb-group domains (ChromHMM, ± 200 kb), Rad21 over architectural peaks (ENCODE of Rad21 and CTCF, ± 100 kb) and SETDB1 over LADs (4DNucleome, ± 500 kb). Top line plot shows the average enrichment of signal, bottom heatmap shows signal per genomic region (sorted on MAbID signal). The number (N) of genomic regions included per heatmap is indicated. The heatmap data range is indicated underneath.

However, we reason that both Rad21 and SETDB1 would benefit from additional optimizations in antibody concentration, which could improve the signal-to-noise ratio and enhance the enrichment. We therefore only included CTCF and SUZ12 in the revised manuscript. With these results, we are confident that, besides histone PTMs, MAbID can be applied to chromatin-binding proteins that transiently associate with the DNA. We updated the manuscript with the following text:

Lines ~146-150: *“Finally, we investigated the potential of MAbID to profile other general chromatin-binding proteins. We focused on two proteins with a well-characterized genomic binding profile - CTCF, a zinc-finger transcription factor, and SUZ12, a subunit of the Polycomb Repressive Complex 2. Both proteins*

display the expected enrichment of signal over the corresponding publicly available ChIP-seq peaks or ChromHMM domains (Extended Data Fig. 2f)."

Comment #2.7

Single-cell chromatin multiplexed mapping, particularly with this high number of features profiled simultaneously, could potentially be applied in countless different contexts and research fields; this is very exciting but the lack of evidence showing that this method works in primary cells is a major concern. The authors should validate their method in a more complex system, such as mouse brain cells or peripheral blood samples to demonstrate they have the power and the resolution to at least identify the major cell subtypes in these tissues.

We thank the reviewer for emphasizing the significance and potential of performing MAbID in more complex primary samples. Therefore, we decided to implement MAbID on cells isolated from mouse bone marrow (BM) and at the same time, we modified the protocol slightly to also leverage the use of FACS-sorting to select for cells of interest by implementation of fluorescent antibodies against cell-surface receptors (Fig. 5a and Extended Data Fig 8a).

We generated a dataset in which we performed combined genomic profiling of six epitopes using primary antibody-DNA conjugates on isolated mouse bone marrow, which we added to the manuscript in a new main and Extended Data figure (Fig. 5 and Extended Data Fig. 8). We chose this tissue based on the suggestion of the reviewer as well as the availability of the material. There are excellent commercially available fluorescent antibodies against well-known BM cell-surface receptors. This enabled us to FACS sort five distinct cell types, two of which originate in the myeloid lineage and 3 originating from the lymphoid lineage (Fig. 5a and Extended Data Fig. 8a). To perform immuno-staining against cell-surface receptors, we slightly modified the MAbID protocol and used ethanol fixation instead of formaldehyde. This largely preserves the integrity of the cellular membrane. We note that this procedure results in reduced unique read counts, which requires further optimizations in the future (Extended Data Fig. 8b). Nevertheless, the MAbID data retains specificity and we think this additional application of MAbID has a lot of potential.

We integrated all epitope counts, as we did previously for the neural differentiation trajectory, and examined whether this would allow us to distinguish the cell types. For the combined scMAbID *in silico* populations (ISP), we see an excellent separation on both lineage and cell type (Fig. 5b), indicating that the single-cell samples hold cell type-specific information. The scMAbID single-cell samples also show a noticeable separation on cell type, but mainly cluster by lineage (Fig. 5c and Extended Data Fig. 8c). We reason that the relatively low count numbers contribute to these results, as the large fraction of zero-value genomic bins causes a significant similarity between all cells, making it challenging to identify unique features within a specific cell type. Combining single-cell samples together in an ISP population effectively

increases the count number per sample and thereby indeed generates enough power to identify cell types based on the multifactorial chromatin profiles.

Besides investigating the identification of cell types, we also examined the potential of scMABID to discern differential gene expression programs. To this end, we generated a highly compatible reference dataset with the most differentially expressed genes for each lineage and cell type, based on public sortChIC⁹ data in which the same FACS gating strategy was applied. We observe a strong lineage-specific enrichment in enhancer counts (combining H3K4me1 and H3K27ac) over the corresponding marker gene sets (Fig. 5d and Extended Data Fig. 8e). Moreover, we identify similar patterns in the different cell types when comparing these to the top 50 cell type-specific marker genes (Fig. 5e and Extended Data Fig. 8f). Finally, we investigated whether scMABID data could be used unbiasedly to identify differentially expressed genes using the multifactorial genomic profiles. By calculating the number of scMABID enhancer counts over each gene in single cells and comparing the differential signal enrichment between cell types, we could identify a small set of significant genes (Supplementary Table 4). Based on literature comparisons, most of these are known to be expressed in the corresponding BM cell types and many overlap with the genes identified in the reference dataset (Extended Data Fig. 8g). Despite the small fraction of cells in which these genes are detected, related to the relatively low count numbers, these results validate that scMABID can be applied in a complex primary tissue and can distinguish cell type-specific gene expression programs.

Comment #2.8

Due to the potentially broad application of this new technique, it would be helpful if the authors upload detailed step-by-step experimental protocol (e.g., in protocol io).

This is a very good and important suggestion that we will for sure follow up on. We will provide a publicly accessible step-by-step protocol for both the experimental and computational methods, similar to what we have provided in the past for our scDamID³ and scDam&T-seq⁴ methods. Currently, the original manuscript is already available on bioRxiv, (<https://doi.org/10.1101/2023.01.18.524584>), including a detailed description of the method. Furthermore, we created an R-package (mabidR) to facilitate processing of the MABID data, which is publicly available on Github along with the other relevant code (<https://github.com/KindLab/MABID>).

Reviewer #3:

The Authors present a novel method for the multiplexed epigenetic profiling (MAbID) of low-input samples. As stated in the manuscript, further need of methodology development and refinement is still lacking in this field, as the golden standard (ChIP-seq) requires large amounts of input material, and one of its most promising alternatives (CUT&Tag) has been shown to have a clear bias to map open chromatin regions. The methodological design is based on the conjugation of oligonucleotide adapters to antibodies, and their ligation to compatible restriction enzyme cutting sites. Thus, the genomic regions where the target epitope is bound to can be labeled for its genomic profiling by sequencing.

The authors first show a proof-of-principle MAbID experiment in K562 cells with a single epitope profiling, demonstrating its reproducibility for several histone marks and two chromatin-binding proteins. Notably, MAbID detects the genomic profiles of histone marks and proteins at sites of heterochromatin with good correlation with ChIP-seq data, clearly overcoming the before-mentioned bias of the Tn5-based techniques. Next, the authors test the capacity of multiplexing the assay, whether by combining primary antibodies of different sources with adapter-conjugated secondary antibodies; or by combining multiple adapter-conjugated primary antibodies. This second strategy allows for easy multiplexing of the assay with up to 6 different antibodies. Remarkably, one of the main strengths of the proposed method is the unaffected performance, in terms of detection and resolution, when increasing the number of epitopes to be analyzed simultaneously.

The authors also demonstrate the utility of scMAbID for the multiplex epigenetic profiling in a biologically relevant model, such as differentiation of mouse ESCs to neural progenitors, being able to discriminate between different populations on the base of the single-cell signature generated by the integration of multiple epigenetic marks; and allowing the single cell monitoring of the epigenetic inactivation of the X chromosome along this differentiation process.

In summary, MAbID and scMAbID represent a clear progress and a new promising tool for the epigenetic profiling of multiple chromatin-binding proteins, with fair detection and reproducibility in both hetero and euchromatic regions, low needs of input material, and its capacity of being multiplexed without losing performance power.

Nevertheless, the manuscript in its current form would benefit from improved clarity, before it is ready for publication. Below I list a few points to clarify:

We thank the reviewer for the in-depth and detailed evaluation of the manuscript. We are equally enthusiastic about the unaffected performance upon multiplexing measurements and are optimistic that this will allow profiling of increasingly complex sets of epitopes. We very much appreciate the valid points regarding the limitations of MAbID as well as valuable suggestions to strengthen the method.

Comment # 3.1

MABID is based on the ligation of adapters into restriction enzyme target sites through specific DNA sequences. Indeed, the authors clearly show that the resulting reads mainly map (>95%) to the target DNA motifs (TTAA and GATC for each of the used systems). While this is a solid validation of the methodological design, this sequence-specificity may interfere with detection of protein binding sites. As the authors explain, the simplicity of these DNA binding motifs makes them highly abundant in the human genome (18 million TTAA and 7 million GATC motifs) but it also poses a clear bias, since one would not be able to detect protein binding sites in genomic regions without these concrete motifs nearby. Could this explain the fact that the MABID peaks are broader in comparison to ChIP-seq? One of the proposed advantages of MABID is the fact that it overcomes the preference of Tn5-based methods towards euchromatin. For consistency, the Authors should remark and discuss this possible limitation of their own technology and its effects on the results, discuss further analyses to assess and measure its impact, and propose a possible mitigation strategy.

We fully agree with the reviewer that targeting of specific sequence motifs can result in biases and could potentially lead to decreased detection rates in certain types of chromatin. Our tailored design to match the epitope of interest, by implementing another type of antibody-adapter targeting GATC sequence motifs, was included to (at least in part) overcome this limitation. To explore the occurrence of genomic biases in the MABID approach, we assessed whether certain chromatin types have a lower potential to be detected by MABID due to a decreased coverage in sequence motifs. We determined this by calculating the amount of 'mappable' bins in each ChromHMM state for both TTAA and GATC motifs, as well as common blacklisted regions⁵ (Extended Data Fig. 4b). A mappable genomic bin is defined as 'having the potential to uniquely align a read at the respective sequence motif'¹⁰, thereby also taking into account repetitive sequences.

We do not observe a strong overrepresentation of unmappable bins in certain chromatin types for the TTAA and GATC sequence motifs, besides the 'Quiescent' state, which contains a large number of repetitive sequences (Extended Data Fig. 4b). 'Enhancer' states are slightly less mappable for the TTAA motif, but since we use the GATC (BglII-compatible) adapter for the H3K4me1 and H3K27ac hPTMs, this should not influence the detection of these epitopes in the combined measurements. 'ZNF genes & repeats' are somewhat less mappable for the GATC motif, again most likely due to repetitive sequences. Overall, the mappable genomic coverage for both motifs is higher than for the blacklisted regions, indicating that MABID does not have an overall lower detection rate than is common for general sequencing approaches.

Lines ~209-213: "We selected the combination of MboI & BglII to target GATC motifs, because of i) the high efficiency of MboI to digest cross-linked chromatin^{35,36}, ii) the different genomic distribution of the GATC motif compared to the TTAA motif (Extended Data Fig. 4a), iii) the high fraction of mappable genome

across all chromatin types (Extended Data Fig. 4b) and iv) the compatibility of the nucleotide overhangs that remain after Mbol and BglII digestion.”

It is not trivial to determine whether the limitation by sequence motifs in the MAbID approach is the cause for the somewhat lower resolution compared to ChIP-seq, as the reviewer suggests (Extended Data Fig. 2e). This is a possibility, but could also be caused by the differences in signal values (as a result of the normalization strategy implemented in MAbID), or by the lower cell numbers that are used to obtain the data (1000 cell for MAbID compared to millions of cells for ChIP-seq). We recognize that this potential bias in sequence motifs was not sufficiently discussed within the manuscript text and thank the reviewer for pointing this out. The following section was thus added to the discussion:

Lines 480-487: “Since MAbID employs restriction-ligation steps, its signal distribution is restricted to certain sequence motifs (initially only TTAA). We have already expanded the approach with an additional motif (GATC) by adopting another set of restriction enzymes. These four-basepair motifs are highly abundant in the genome and we observe an equal representation of chromatin types when using these sequences to uniquely align reads. However, we are unable to rule out that small genomic fragments go undetected due to local biases in sequence, potentially influencing the experimental resolution of MAbID. To circumvent this, more sets of restriction enzymes could be included or the method could be adopted to use genome-digestion enzymes that are unbiased towards sequence, such as MNase⁵⁷.”

Comment #3.2

The authors state that MAbID allows the “profiling of an increasingly complex set of histone PTMs and chromatin-binding proteins.” However, the only non-histone chromatin-binding proteins tested and presented in the manuscript are Lamin B1 and RNA polymerase II. Increasing the examples of non-histone protein profiling would increase the impact of the manuscript. Have the Authors tested the method with some other chromatin-binding proteins characterized by a more labile interaction with DNA, which often pose problems in ChIP-seq? It would be interesting to see profiles of some chromatin modifiers and remodelers, like members of SWI/SNF, Polycomb or Thrtorax complexes. This would add a major strength of the proposed methodology, in comparison with the golden standard in the field.

We agree that ability to profile such transiently-interacting chromatin-binding proteins with MAbID would be a great addition to the method, so we tested several other chromatin-binding proteins based on the suggestions of the reviewers. For further explanation, please refer to our response to **comment #2.6** of reviewer 2.

Comment #3.3

The Authors suggest in line 361 of the manuscript that they detected a higher MAbID background signal in regions of open chromatin. Although they clarify and show that it can be corrected with a negative control, like a CHIP-seq input, they do not show data regarding this bias towards regions of euchromatin, which they attribute to “locally increased efficiency of either the digestion or ligation steps”. If they have a specific analysis supporting this statement and measurement of the degree of the impact of this bias, they should include it in the manuscript.

Thank you for this important comment. We have now provided more in-depth information on the distribution of the control signal and the normalization strategy (see **comment #2.1**). We added additional panels (**Extended Data Fig. 1d-f**) to give more insight into this and have included extra sentences in the result section of the main text for further clarification. We would like to refer to our response to **comment #2.1** of reviewer 2 for further explanations on the new panels. Additionally, we have updated the discussion section in the main text to provide a broader review of this potential limitation:

*Lines 472-479: “MAbID can be further improved to reduce the background signal that we observed at accessible chromatin regions (**Extended Data Fig. 1d**). This signal most likely results from the locally increased efficiency of either the digestion or ligation steps. The background can be effectively corrected for by normalization over a control sample (**Extended Data Fig. 1d-f**), analogous to normalization approaches in CHIP-seq⁵⁶. Still, further reduction of the off-target signal would be beneficial, in order to circumvent any normalization requirement. Additional technical improvements to achieve this could include optimizing blocking reagents, performing extensive antibody titrations or optimizing restriction-digestion and ligation steps of the protocol.”*

Comment #3.4

The Authors should compare the results of MAbID with multiplex CUT&Tag data. Apparently, one of the main advantages of their technology is the proper detection of binding sites in heterochromatin sites, as opposed to Tn5-methods. Comparing and measuring this difference will improve the robustness of their results and their claims.

We have provided a more extensive comparison between both MAbID and bulk CUT&Tag as well as scMAbID with the other new multifactorial approaches. These include an evaluation of FRiP scores, read/count numbers per cell as well as epitope and the required/recovered numbers of cells – please

refer to our response to **comments #2.3 and #2.4** of reviewer 2 as well as **comment #1.4** of reviewer 1 for an in-depth discussion of these results.

None of the other recent multifactorial approaches have included epitopes residing in constitutive heterochromatin in their combined measurements^{1,6-8}, and it is therefore difficult to make a direct comparison. We validated our combined scMABID measurements of H3K9me3 and Lamin B1 by comparing genomic ISP (*in silico* population) profiles to corresponding bulk ChIP-seq profiles and by calculating single-cell FRiP scores over LAD regions (Fig. 4c and 4e, Extended Data Fig. 6e and Extended Data Fig. 7a-b). These results verify that our measurements generate the expected distribution of signal.

Based on comments by reviewer 1, we also realized and acknowledged that our statements in both the introduction and discussion sections on the performance of CUT&Tag in constitutive heterochromatin were too broadly phrased. We updated these sections of the main text and provide a more detailed response to the reviewer **comment #1.2** above.

Comment #3.5

At some instances the Authors use thought short-cuts assuming that things are obvious to everyone but I would encourage the authors to provide more explanations for why they make certain choices in their method and approach. I list here such instances:

- a. "We initially employed secondary antibody-DNA conjugates to more accurately compare the quality of multiple genomic profiles in parallel."
- b. "We therefore excluded H3 from subsequent analysis and only added it to function as a crowding reagent during following experiments." I do not understand why it would be needed to add it as a crowding reagent and how it would work as such.
- c. "The conjugation procedure was slightly modified to account for differences in buffer compositions". Which buffer do the Authors refer to?

We apologize for the lack of clarity regarding these sentences, and we hope to have sufficiently amended them to make our intention clearer to the reader. In the specific instance of sentence 'b' noted by the reviewer, we also updated our terminology to make the explanation more accurate.

- a. **Lines ~107-110:** *"We initially employed secondary antibody-DNA conjugates to more accurately compare the quality of multiple genomic profiles in parallel, since different primary IgGs can be combined with the same batch of secondary antibody-DNA conjugate."*

- b. **Lines ~184-187:** *“We therefore excluded H3 from subsequent analysis and only added it during following experiments to function as a potential blocking reagent, by preventing unspecific chromatin-binding events of the other secondary antibody-DNA conjugates.”*
- c. **Lines ~244-245:** *“The conjugation procedure was slightly modified to account for differences in storage buffer compositions of the commercially acquired primary antibodies.”*

Comment #3.6

The correlations the Authors report, ranging from 0.24 to 0.47 seem low. Could the Authors comment on this?

We agree with the reviewer that the correlation scores are relatively low, yet it is difficult to provide a conclusive explanation for this. It is important to consider that the MAbID data is generated in 1000-cell samples, as opposed to millions of cells used for the ENCODE ChIP-seq or CUT&Tag data that we used in comparison. Based on visual comparisons of the genomic tracks on a local scale (Fig. 1d) and the patterns of enrichment over corresponding ChIP-seq and ChromHMM domains (Fig. 1e-f), we note a high overlap between the MAbID and reference datasets. In addition, we observe that the correlations are clearly highest with the corresponding dataset for both ChIP-seq and CUT&Tag data and anti-correlate with non-related chromatin types (Extended Data Fig. 2b). Finally, the newly added FRiP and SEiP scores corroborate the specificity of MAbID (Extended Data Fig. 2d). These results show that with the quantitative genome-wide correlations, we observe the same trends as with the more qualitative comparisons, but at a lower overall score.

Of note, we have examined our method to calculate genome-wide correlations between datasets and have updated these to provide a more accurate assessment, in response to **comment #2.1** of reviewer 2. Still, these were minor adjustments and thus did not change the overall mean Pearson’s correlation coefficients, ranging from 0.24 to 0.50 for active chromatin types and 0.24 to 0.46 for heterochromatin types (updated in **lines ~124-126**).

One of the other potential reasons for the relatively low correlation scores is the normalization method. Because of the normalization procedure in MAbID data processing, the overall signal values are in the format of $\log_2(\text{counts}/\text{control})$ for each genomic window (bin). Since we aim for a very rich control dataset to enhance the accuracy of the normalization, many bins contain high signal in the control sample and we thereby concomitantly decrease the signal amplitude after normalization. In addition to this, as we unavoidably still observe a minority of bins in the control dataset with a value of 0, we have to use a

pseudocount value of 1 to be able to perform the normalization. This essentially means the following for the calculation – normalized MAbID signal per bin = $\log_2((\text{signal-rpkm} + 1)/(\text{control-rpkm} + 1))$. This pseudocount value diminishes noise in regions of low coverage, but also results in a reduction of the $\log_2(\text{counts}/\text{control})$ values. Taken together, this means that the signal values in MAbID are non-integer values that are overall much lower than for ChIP-seq and CUT&Tag, as can be noted from the y-axis in Fig. 1d. Even though the signal distribution is as expected, this most likely reduces the overall genome-wide correlation.

Comment #3.7

Can the Authors add information in the main text whether the ChIP-seq data they used were publicly-available or whether they were generated for the purpose of this study.

This was indeed insufficiently highlighted in the main text – we thank the reviewer for bringing this to our attention. All the reference data used in the manuscript is publicly available, which we used in order to make the fairest comparison of MAbID and scMAbID to state-of-the-art approaches. We have now included this information at lines 104, 124, 131, 149, 168, 175, 201, 249-250, 261, 271, 307, 316, 336 and 426.

Comment #3.8

It would be useful if the Authors provided some considerations on the number of cells they use in their bulk assay (250,000). Could the Authors comment on the upper limit?

The starting number of cells required for both the MAbID and scMAbID protocol is largely dependent on the antibody incubation steps. To ensure low levels of background, several washing steps are done to remove all unbound antibody from the sample after incubation, during which nuclei are inevitably lost. The more starting material is used, the more efficient the recovery is, especially when the numbers are in a range where the pellet is clearly visible. So far, we have performed MAbID with starting material of 3 million to 150,000 nuclei, but we generally aim for 250,000 nuclei per sample, which works robustly in our hands. However, this number could be decreased further, and we expect that starting materials of 50,000-100,000 nuclei should be feasible without additional alterations to the protocol.

Furthermore, since MAbID employs FACS sorting to obtain purified samples, there is the possibility to add fluorescently labelled 'carrier' nuclei to the input material. These nuclei can be included during the antibody incubation steps to prevent loss of material and will be removed again during FACS sorting by setting gates to select against these fluorescent nuclei. Another option is to label different samples with distinct fluorescent dyes, to pool these during antibody incubations and subsequently FACS sort the desired number of cells for each sample. This is one of the advantages of the plate-based scMAbID protocol, as one can preselect cells based on FACS parameters, instead of having to sequence all cells in a population or sample. We specifically used this to sort discrete populations of cells from mouse bone marrow by staining for cell-surface markers (Fig. 5a and Extended Data Fig. 8a, see **comment #2.7 and #3.9**). Moreover, we have also implemented a labelling strategy in the lab by staining nuclei with commercially available CellTrace dyes, which is highly efficient and can be performed with fixed material. We have included the following sentence in the discussion to emphasize this option:

Lines 498-509: "Besides the advantage of using primary antibody-DNA conjugates, the implementation of FACS sorting within the MAbID protocol provides the opportunity to select specific cells from a larger population. This strategy can prove especially powerful when enriching for rare cell types. We have validated this by sorting five discrete cell types from mouse bone marrow, using fluorescently labelled antibodies against cell-surface markers. In combination with the plate-based scMAbID protocol, this can greatly reduce sequencing costs, by selecting cells of interest instead of sequencing entire populations. Moreover, it creates the potential to reduce the required number of cells during antibody incubations, by adding fluorescently labelled carrier cells to a low-input sample. These labelled cells will increase the efficiency of the antibody incubation steps and can be removed again during FACS sorting, thereby enabling the protocol to work on increasingly low cell numbers. On the other hand, the current plate-based protocol prevents achieving high throughput. Implementing combinatorial-indexing strategies⁵⁸ in future improvements of the scMAbID method could resolve this limitation."

Comment #3.9

Why do the Authors work with isolated nuclei as opposed to entire cells. Can the Authors comment on this? Can one work with entire cells as well?

When we started developing the MAbID approach, we build on (now published) information from the sortChIC method⁹ to optimize buffer conditions. In sortChIC, as well as CUT&RUN¹¹ and CUT&Tag¹² approaches, isolated nuclei are used to optimize entry of the antibodies, the pA-fusion protein and other reagents required to perform all enzymatic steps. We therefore used similar buffers and timings for antibody incubations.

To explore the possibility to perform MAbID in whole cells, we have slightly modified the MAbID method – we changed the formaldehyde fixation to ethanol fixation and used Tween20 instead of Saponin in all washing buffers. Excitingly, this preserved the cellular membrane, thereby offering the possibility to select cells of interest based on cell-surface markers (Fig. 5a and Extended Data Fig. 8a). We think this option further illustrates an advantage of plate-based assays and increases the potential for MAbID. We have included details on this approach in the methods section.

Lines ~1117-1123: *'After removing the supernatant, cells were counted using a TC20™ Automated Cell Counter (BioRad, 1450102) and diluted in 300 µL PBS per 1×10^6 cells. Per 300 µL PBS, 700 µL ice-cold Ethanol (100%, Boom, 84028185) was added dropwise while vortexing the suspension, to reach a 70% Ethanol final concentration. Cells were fixated for 1 hour at -20 °C. After fixation completed, cells were washed with Wash buffer 1 (20mM HEPES pH 7.5 (Gibco, 15630-056), 150 mM NaCl, 66.6 µg/mL Spermidine (Sigma, S2626), 1X cOmplete™ protease inhibitor cocktail (Roche, 11697498001), 0.05% Tween20 (Sigma, P9416), 2mM EDTA).'*

AND

Lines ~1206-1209: *"We would like to note that the formaldehyde fixation (in combination with Saponin-containing Wash buffers) can also be replaced with ethanol fixation (in combination with Tween20-containing Wash buffers) to preserve the cellular membrane and enable immunostainings for cell-surface markers. See section 'Mouse bone marrow isolation and ethanol fixation' for further details."*

Comment #3.10

Is there any risk coming from the way the conjugation of the adapters to the antibodies is done whereby multiple adapters could get conjugated to the same antibody? Could the Authors comment on that?

The antibodies are conjugated to the antibody-adapters using SPAAC click-chemistry^{13,14}, which results in a covalent bond between the antibody and the DNA adapter. This bond is irreversible under these conditions and the DNA adapter can therefore not 'jump' to another antibody after the conjugation reaction is completed. We perform the antibody conjugations for different antibodies in parallel (and obviously in separate containers) to reduce potential batch effects, but for a skilled molecular biologist maintaining high lab-standards there is no risk of any cross contaminations.

Comment #3.11 & #3.12

Can the Authors comment in the main text on how the tailoring of the enzymes for a given epitope quantitatively affects the resolution of the assay? How strong is this effect?

AND

Related to this, the Authors comments on the fact that with this approach (tailoring by selecting the proper restriction site) they expect an increase in the resolution and yet the conclude this paragraph by saying “Both the resolution and distribution of MAbID signal were surprisingly independent of the choice of antibody-adapter and the complexity of the data was similar between all approaches (Extended Data Fig. 3c-d).” What am I missing?

We thank the reviewer for bringing this to our attention. Our analyses mainly focused on the equal distribution of signal between different antibody-adapters instead of also focusing on a quantitative assessment of the difference in resolution. We have now included addition analyses to address this.

To determine whether including both motifs indeed increases the resolution of the MAbID data, we calculated the resolution of H3K27me3 MAbID data when profiled with a TTAA adapter, a GATC adapter or both antibody-adapter types (TTAA and GATC). This resulted in an increased resolution of approximately 1-2 kb for the combined approach in comparison to the individual measurements (Extended Data Fig. 4e).

Lines ~224-230: “All sample types group based on epitope and display the expected distribution of signal enrichment ~~enrichment of signal~~, regardless of the choice of recognition motif or the number of multiplexed antibodies (Extended Data Fig. 4c-d). Both the resolution and distribution of MAbID signal were surprisingly independent of the choice of antibody adapter and the complexity of the data was similar between all approaches (Extended Data Fig. 3c-d). The signal resolution increased 1-2 kb in the combined sample (TTAA and GATC) compared to the individual samples, as measured by the decay of H3K27me3 signal over the Polycomb (PcG) ChromHMM domain border (Extended Data Fig. 4e). The overall complexity of the data was independent on the choice of antibody-adapter and we obtain similar read numbers per sample for all approaches (Extended Data Fig. 4f-g).”

Comment #3.13

I wonder whether the assay could be simplified by having the adapter conjugated to the Antibodies already with a sticky end and phosphorylated so that no additional step of enzyme digestion would be needed. In general it would be great to add more explanation for why the method requires two different enzymes (I assume it has to do with the need for only one of the two sites being in a phosphorylated state but it would be good to have this reasoning explicitly stated in the main text).

This is an excellent suggestion and we would like to thank the reviewer for this. In the current MAbID protocol, the genomic DNA is dephosphorylated after digestion to prevent self-ligation of genomic fragments and promote ligation of the antibody-adapter into the genome. In case the antibody-adapter was already phosphorylated upon antibody-DNA conjugate incubation, the phospho-group would also be

removed from the antibody-adapter during this step. For this reason, the antibody-adapter is digested after the dephosphorylation to provide a compatible overhang to ligate into the genome.

We tested with H3K36me3 whether it was possible to perform genomic digestion and dephosphorylation before incubating with the antibody DNA-conjugates, in order to directly use a compatible and phosphorylated adapter and omit the antibody-adapter digestion step. We named this approach ‘pre-stain digestion’, compared to the ‘post-stain digestion’ of the current approach (see figure below, i). The complexity of the samples is similar – the percentage of valid reads is slightly higher for the post-stain digestion, but this sample also had a slightly higher sequencing depth (see figure below, ii). However, when comparing the enrichment of signal over genes, we noted that the background levels in the pre-stain digestion approach are significantly higher, while the signal over the gene itself is lower (see panel below, iii). We therefore decided that the current approach results in higher data quality and is thus preferred.

Legend – i) Cartoon showing the difference between the antibody-adapters in the ‘post-stain’ (current MAbID approach) and ‘pre-stain’ genomic digestions. In the ‘pre-stain’ condition, genomic DNA digestion occurs in bulk directly after fixation and permeabilization. After dephosphorylation of the genome (with rSAP), the nuclei are incubated with antibody-DNA conjugates. These are different from the current ‘post-stain’ adapters, in that they already have an overhang and phospho-group to facilitate ligation into the genome. Thereby, the antibody-adapter digestion step (with NdeI or BglII) can be omitted. Antibody-adapters are then ligated into the genome in bulk, before FACS sorting 1000 cell samples and performing the subsequent steps of the MAbID protocol, starting with lysis and proteinase K digestion. ii) Plot showing the yield for both approaches against the sequencing depth (in demultiplexed reads). iii) MAbID signal enrichment of H3K36me3 over genes (-/+ 250 kb), comparing the post-stain and pre-stain digestion approaches. Top line plot shows the average enrichment of signal, bottom heatmap shows signal per genomic region (sorted on gene expression). The number (N) of genes included per heatmap is indicated and the heatmap data range is indicated underneath.

Yet, we were excited to note that there is potential to perform genomic digestions and ligations in bulk, since this could benefit potential expansions of the MAbID method, especially when implementing combinatorial-indexing approaches. We agree with the reviewer that it would be useful to provide additional information on this part of the protocol in the main text, so we included a small section on this:

Lines ~90-98: “4) digestion of the genome with the MseI restriction enzyme, which recognizes TTAA sequence motifs, 5) dephosphorylation of the digested genome ~~to prevent self-ligation of genomic fragments~~, 6) digestion of the antibody-adapter with the NdeI restriction enzyme, which leaves a MseI-compatible overhang with a 5’ phosphate and 7) ligation of the antibody-adapter into the digested genome using the matching overhangs. The position of the antibody-adapter within the genome hereby becomes a proxy for the localization of the epitope of interest. Dephosphorylation of the digested genome (step 5) is included to prevent self-ligation of the genome and to thereby enhance the integration of the antibody-adapter. Performing the digestion of the antibody-adapter (step 6) after dephosphorylation ensures that the antibody-adapter itself is not affected and retains the potential to ligate.”

Other adjustments

- We modified the abstract to include a sentence on the application of MAbID on primary mouse bone marrow - **lines ~34-35**. To prevent exceeding the word count, surplus words were removed from other sentences in the abstract. – **lines ~23-37**. Likewise, we added a similar sentence to the end of the introduction – **lines ~75-77**.
- As mentioned in our response to **comments #2.3 and #2.4**, we calculated FRiP scores on both the MAbID and scMAbID data along with publicly available datasets. To this end, we made slight alterations to our original way of calculating FRiP scores on scMAbID data alone. To ensure that all FRiP score calculations are done identically throughout the manuscript, we updated **Fig. 4e** and **Extended Data Fig. 7b**.

- We have updated the approach to calculate genome-wide correlations between datasets (see **comment #2.1**) and have updated **Fig. 1b** and **Extended Data Fig. 5e** accordingly.
- We included the isolation of bone marrow cells as well as the staining with cell-surface markers in the methods section – **lines 1107-1124, 1206-1216, 1221 and 1246-1266**.
- We updated the description of the computation analyses in methods section, to provide a more detailed report and include new analyses – **lines 1370-1392, 1397-1400 and 1407-1524**.
- We updated the figure legends to the updated numbering and included the legends for the newly added panels – **lines 704-1045**, directly after the corresponding figure.

References

- 1 Gopalan, S., Wang, Y., Harper, N. W., Garber, M. & Fazio, T. G. Simultaneous profiling of multiple chromatin proteins in the same cells. *Molecular Cell* **81**, 4736-4746. e4735 (2021).
- 2 Wang, M. & Zhang, Y. Tn5 transposase-based epigenomic profiling methods are prone to open chromatin bias. *bioRxiv*, 2021.2007.2009.451758 (2021). <https://doi.org/10.1101/2021.07.09.451758>
- 3 Kind, J. *et al.* Genome-wide maps of nuclear lamina interactions in single human cells. *Cell* **163**, 134-147 (2015).
- 4 Rooijers, K. *et al.* Simultaneous quantification of protein–DNA contacts and transcriptomes in single cells. *Nature biotechnology* **37**, 766-772 (2019).
- 5 Amemiya, H. M., Kundaje, A. & Boyle, A. P. The ENCODE blacklist: identification of problematic regions of the genome. *Scientific reports* **9**, 1-5 (2019).
- 6 Meers, M. P., Llagas, G., Janssens, D. H., Codomo, C. A. & Henikoff, S. Multifactorial profiling of epigenetic landscapes at single-cell resolution using Multi-Tag. *Nature Biotechnology*, 1-9 (2022).
- 7 Bartosovic, M. & Castelo-Branco, G. Multimodal chromatin profiling using nanobody-based single-cell CUT&Tag. *Nature Biotechnology* (2022). <https://doi.org/10.1038/s41587-022-01535-4>
- 8 Stuart, T. *et al.* Nanobody-tethered transposition enables multifactorial chromatin profiling at single-cell resolution. *Nature Biotechnology* (2022). <https://doi.org/10.1038/s41587-022-01588-5>
- 9 Zeller, P. *et al.* Single-cell sortChIC identifies hierarchical chromatin dynamics during hematopoiesis. *Nature Genetics* **55**, 333-345 (2023). <https://doi.org/10.1038/s41588-022-01260-3>
- 10 Markodimitraki, C. M. *et al.* Simultaneous quantification of protein–DNA interactions and transcriptomes in single cells with scDam&T-seq. *Nature Protocols* **15**, 1922-1953 (2020).
- 11 Skene, P. J. & Henikoff, S. An efficient targeted nuclease strategy for high-resolution mapping of DNA binding sites. *Elife* **6**, e21856 (2017).
- 12 Kaya-Okur, H. S., Janssens, D. H., Henikoff, J. G., Ahmad, K. & Henikoff, S. Efficient low-cost chromatin profiling with CUT&Tag. *Nature Protocols* **15**, 3264-3283 (2020). <https://doi.org/10.1038/s41596-020-0373-x>
- 13 Agard, N. J., Prescher, J. A. & Bertozzi, C. R. A strain-promoted [3+ 2] azide–alkyne cycloaddition for covalent modification of biomolecules in living systems. *Journal of the American Chemical Society* **126**, 15046-15047 (2004).
- 14 van Buggenum, J. A. *et al.* A covalent and cleavable antibody-DNA conjugation strategy for sensitive protein detection via immuno-PCR. *Scientific reports* **6**, 1-12 (2016).

Decision Letter, first revision:

Our ref: NMETH-A51240A

31st Aug 2023

Dear Dr. Kind,

Thank you for submitting your revised manuscript "Combinatorial single-cell profiling of all major chromatin types with MAbID" (NMETH-A51240A). It has now been seen by the original referees and their comments are below. The reviewers find that the paper has improved in revision, and therefore we'll be happy in principle to publish it in Nature Methods, pending minor revisions to satisfy the referees' final requests and to comply with our editorial and formatting guidelines.

TRANSPARENT PEER REVIEW

ORCID

Sincerely,
Lei

Lei Tang, Ph.D.
Senior Editor
Nature Methods

Reviewer #1 (Remarks to the Author):

The authors have addressed my concerns that they have oversold their method relative to competing methods based on CUT&Tag. Importantly, they have removed the criticism of open chromatin bias of CUT&Tag, which was based on a thoroughly debunked unreviewed preprint, and in response to points raised by Reviewer 2 have gone on to find that their own method had an apparently more serious open chromatin bias. To deal with this, they run a negative control, using it to remove the open chromatin bias in the analysis (I wouldn't call it "normalization" which implies something very different). I appreciate the authors being upfront about this (which had misled Reviewer 3) and doing more thorough comparisons against competing methods in response to reviewer suggestions. The authors have also improved the method to the point that they can include less abundant epitopes and have adopted an ethanol fixation method that allows them to perform the method on whole cells. Taken together with the novelty of the method, and the importance of making available orthogonal methods for single-cell multi-factorial chromatin profiling, I support publication in Nature Methods.

Reviewer #2 (Remarks to the Author):

The updated analysis and dataset have successfully addressed all my comments. Overall I am satisfied with the updated manuscript. Congratulations on this exciting technique.

Reviewer #3 (Remarks to the Author):

I am very satisfied with how the authors improved the manuscript following Reviewers' comments and I now find it ready for publication.

Author Rebuttal, first revision:

Response to reviewers

We would like to thank all the reviewers for their comments and the time and effort they invested in reviewing our manuscript. Their constructive feedback and helpful suggestions were very valuable to us in revising the manuscript to its final version. Please find our point-by-point response to the reviewer comments in blue.

Reviewer #1:

The authors have addressed my concerns that they have oversold their method relative to competing methods based on CUT&Tag. Importantly, they have removed the criticism of open chromatin bias of CUT&Tag, which was based on a thoroughly debunked unreviewed preprint, and in response to points raised by Reviewer 2 have gone on to find that their own method had an apparently more serious open

chromatin bias. To deal with this, they run a negative control, using it to remove the open chromatin bias in the analysis (I wouldn't call it "normalization" which implies something very different). I appreciate the authors being upfront about this (which had misled Reviewer 3) and doing more thorough comparisons against competing methods in response to reviewer suggestions. The authors have also improved the method to the point that they can include less abundant epitopes and have adopted an ethanol fixation method that allows them to perform the method on whole cells. Taken together with the novelty of the method, and the importance of making available orthogonal methods for single-cell multi-factorial chromatin profiling, I support publication in Nature Methods.

We appreciate the reviewer pointing us towards the concerns regarding the mentioned reference, along with the valuable feedback to more carefully position MAbID as an orthogonal method to the tagmentation-based multifactorial approaches. We feel that the direct comparisons to the other multifactorial methods, which were also proposed by the other reviewers, have significantly increased the quality of the manuscript.

We fully agree with the reviewer that the additional panels on the normalization strategy provide further clarity on the MAbID procedure and protocol. We have maintained the term 'normalization' – this term is also used in ChIP-seq when correcting for background signal, where signal is normalized (or divided by) an input control in which a non-specific IgG was used. For MAbID, we have a similar approach in correcting for background, as we divide our signal over a control in which the primary antibody was omitted. We therefore anticipate that this term will be most recognizable and clear to the scientific community. Finally, we are happy that the reviewer appreciates the new additions to the method and would like to thank the reviewer for the support for publication.

Reviewer #2:

The updated analysis and dataset have successfully addressed all my comments. Overall I am satisfied with the updated manuscript. Congratulations on this exciting technique.

We thank the reviewer for the kind comment and the excellent previous suggestion to include MAbID profiling of a primary tissue. We feel that this and other suggestions greatly benefitted the manuscript.

Reviewer #3:

I am very satisfied with how the authors improved the manuscript following Reviewers' comments and I now find it ready for publication.

We are happy to hear this and would like to thank the reviewer for the extensive comments provided to revise the original manuscript. This significantly aided us in improving it

Final Decision Letter:

17th Oct 2023

Dear Dr Kind,

I am pleased to inform you that your Article, "Combinatorial single-cell profiling of major chromatin types with MAbID", has now been accepted for publication in Nature Methods. Your paper is tentatively scheduled for publication in our Dec 2023 or Jan 2024 print issue, and will be published online prior to that. The received and accepted dates will be 16th Dec 2022 and 17th Oct 2023. This note is intended to let you know what to expect from us over the next month or so, and to let you know where to address any further questions.

Over the next few weeks, your paper will be copyedited to ensure that it conforms to Nature Methods style. Once your paper is typeset, you will receive an email with a link to choose the appropriate publishing options for your paper and our Author Services team will be in touch regarding any additional information that may be required.

You will receive a link to your electronic proof via email with a request to make any corrections within 48 hours. If, when you receive your proof, you cannot meet this deadline, please inform us at rjsproduction@springernature.com immediately.

Please note that *Nature Methods* is a Transformative Journal (TJ). Authors may publish their research with us through the traditional subscription access route or make their paper immediately open access through payment of an article-processing charge (APC). Authors will not be required to make a final decision about access to their article until it has been accepted. [Find out more about Transformative Journals](https://www.springernature.com/gp/open-research/transformative-journals)

Authors may need to take specific actions to achieve [compliance with funder and institutional open access mandates](https://www.springernature.com/gp/open-research/funding/policy-compliance-faqs). If your research is supported by a funder that requires immediate open access (e.g. according to [Plan S principles](https://www.springernature.com/gp/open-research/plan-s-compliance)) then you should select the gold OA route, and we will direct you to the compliant route where possible. For authors selecting the subscription publication route, the journal's standard licensing terms will need to be accepted, including [self-archiving policies](https://www.springernature.com/gp/open-research/policies/journal-policies). Those licensing terms will supersede any other terms that the author or any third party may assert apply to any version of the manuscript.

Your paper will now be copyedited to ensure that it conforms to Nature Methods style. Once proofs are generated, they will be sent to you electronically and you will be asked to send a corrected version within 24 hours. It is extremely important that you let us know now whether you will be difficult to contact over the next month. If this is the case, we ask that you send us the contact information (email, phone and fax) of someone who will be able to check the proofs and deal with any last-minute problems.

If, when you receive your proof, you cannot meet the deadline, please inform us at rjsproduction@springernature.com immediately.

Once your manuscript is typeset and you have completed the appropriate grant of rights, you will receive a link to your electronic proof via email with a request to make any corrections within 48 hours. If, when you receive your proof, you cannot meet this deadline, please inform us at rjsproduction@springernature.com immediately.

Once your paper has been scheduled for online publication, the Nature press office will be in touch to confirm the details.

Once your paper has been scheduled for online publication, the Nature press office will be in touch to confirm the details.

Content is published online weekly on Mondays and Thursdays, and the embargo is set at 16:00 London time (GMT)/11:00 am US Eastern time (EST) on the day of publication. If you need to know the exact publication date or when the news embargo will be lifted, please contact our press office after you have submitted your proof corrections. Now is the time to inform your Public Relations or Press Office about your paper, as they might be interested in promoting its publication. This will allow them time to prepare an accurate and satisfactory press release. Include your manuscript tracking number NMETH-A51240B and the name of the journal, which they will need when they contact our office.

About one week before your paper is published online, we shall be distributing a press release to news organizations worldwide, which may include details of your work. We are happy for your institution or funding agency to prepare its own press release, but it must mention the embargo date and Nature Methods. Our Press Office will contact you closer to the time of publication, but if you or your Press Office have any inquiries in the meantime, please contact press@nature.com.

Nature Portfolio journals [encourage authors to share their step-by-step experimental protocols](https://www.nature.com/nature-research/editorial-policies/reporting-standards#protocols) on a protocol sharing platform of their choice. Nature Portfolio 's Protocol Exchange is a free-to-use and open resource for protocols; protocols deposited in Protocol Exchange are citable and can be linked from the published article. More details can found at www.nature.com/protocolexchange/about.

Best regards,
Lei

Lei Tang, Ph.D.
Senior Editor
Nature Methods